# FedDA: Faster Framework of Local Adaptive Gradient Methods via Restarted Dual Averaging

## Abstract

Federated learning (FL) is an emerging learning paradigm to tackle massively distributed data. In Federated Learning, a set of clients jointly perform a machine learning task under the coordination of a server. The FedAvg algorithm is one of the most widely used methods to solve Federated Learning problems. In FedAvg, the learning rate is a constant rather than changing adaptively. The adaptive gradient methods show superior performance over the constant learning rate schedule; however, there is still no general framework to incorporate adaptive gradient methods into the federated setting. In this paper, we propose **FedDA**, a novel framework for local adaptive gradient methods. The framework adopts a restarted dual averaging technique and is flexible with various gradient estimation methods and adaptive learning rate formulations. In particular, we analyze **FedDA-MVR**, an instantiation of our framework, and show that it achieves gradient complexity $\tilde{O}(\epsilon^{-1.5})$ and communication complexity $\tilde{O}(\epsilon^{-1})$ for finding a stationary point $\epsilon$. This matches the best known rate for first-order FL algorithms and **FedDA-MVR** is the first adaptive FL algorithm that achieves this rate. We also perform extensive numerical experiments to verify the efficacy of our method.

## 1 Introduction

Federated Learning denotes the process in which a set of distributed located clients jointly perform a machine learning task under the coordination of a central server over their privately-held data. A widely used method in FL is the FedAvg(Local-SGD) McMahan et al. (2017) algorithm. As indicated by its name, FedAvg performs (stochastic) gradient descent steps on each client and averages local states periodically. This method can be shown to converge Stich (2018); Haddadpour & Mahdavi (2019); Woodworth et al. (2020) when the distributions of the clients are homogeneous or with bounded heterogeneity. Recently, a large amount of literature has focused on accelerating FedAvg. In particular, many research works use momentum-based methods to accelerate FL, and significant progress has been made in this direction with improved gradient complexity and communication complexity Das et al. (2020); Karimireddy et al. (2019a); Khanduri et al. (2021a). However, another important category of methods: adaptive gradient methods have received much less attention, and there is still no general framework to incorporate adaptive gradient methods into the federated setting.

Adaptive gradient methods such as Adagrad Duchi et al. (2011), Adam Kingma & Ba (2014) and AMSGrad Reddi et al. (2018) are widely used in the non-distributed setting. The gradient descent method uses either a fixed learning rate or a fixed learning rate schedule. In contrast, adaptive gradient methods set the learning rate to be inversely proportional to the magnitude of the gradient; this can incorporate the local curvature structure of the problem. Adaptive gradient methods perform well in practice; meanwhile, they also enjoy useful theoretical implications that make them outperform the vanilla gradient descent method Duchi et al. (2011); Guo et al. (2021). For example, a recent study Staib et al. (2019) showed that adaptive gradients help escape saddle points. Furthermore, some studies Loshchilov & Hutter (2018); Chen et al. (2018a) showed that adaptive gradients improve the generalization performance of the model.

Adaptive gradient methods can be viewed as a type of generalized mirror descent Huang et al. (2021) methods, where the associated mirror map is defined according to adaptive learning rates. However,

Table 1: Comparisons of representative Federated Learning algorithms for finding an $\epsilon$-stationary point of Objective equation 1 *i.e.*, $\|\nabla f(x)\|^2 \leq \epsilon$ or its equivalent variants. $Gc(f, \epsilon)$ denotes the number of gradient queries *w.r.t.* $f^{(k)}(x)$ for $k \in [K]$; $Cc(f, \epsilon)$ denotes the number of communication rounds; **State** means what state the algorithm maintains locally (Primal/Dual); **Local-Adaptive** means whether the algorithm performs adaptive gradient descent locally or not; **Constrained** means whether the algorithm can solve both constrained and unconstrained problems or not. The first three algorithms are not adaptive gradient methods, and the last four methods support some form of adaptive gradients.

| Algorithm | $Gc(f, \epsilon)$ | $Cc(f, \epsilon)$ | State | Local-Adaptive | Constrained |
|---|---|---|---|---|---|
| FedAvg McMahan et al. (2017) | $O(\epsilon^{-2})$ | $O(\epsilon^{-1.5})$ | Primal/Dual | ✗ | ✗ |
| FedCM Khanduri et al. (2021a) | $\tilde{O}(\epsilon^{-1.5})$ | $\tilde{O}(\epsilon^{-1})$ | Primal/Dual | ✗ | ✗ |
| STEM Khanduri et al. (2021a) | $\tilde{O}(\epsilon^{-1.5})$ | $\tilde{O}(\epsilon^{-1})$ | Primal/Dual | ✗ | ✗ |
| FedAdam Reddi et al. (2020) | $O(\epsilon^{-2})$ | $O(\epsilon^{-1})$ | Primal | ✗ | ✗ |
| Local-AMSGrad Chen et al. (2020b) | $O(\epsilon^{-2})$ | $O(\epsilon^{-1.5})$ | Primal | ✓ | ✗ |
| MIME-MVR Karimireddy et al. (2020a) | $\tilde{O}(\epsilon^{-1.5})$ | $O(\epsilon^{-1.5})$ | Primal | ✓ | ✗ |
| **FedDA-MVR(Ours)** | $\tilde{O}(\epsilon^{-1.5})$ | $\tilde{O}(\epsilon^{-1})$ | Dual | ✓ | ✓ |

the mirror map is dynamic and changes at every training step. As a special case, the gradient descent method can be viewed as a mirror descent method with the mirror map being the $L_2$ distance function. Following the convention in the mirror descent literature, we denote the parameter space as the primal space and the gradient space as the dual space. The primal and dual space differ in adaptive gradient methods, but they coincide in the gradient descent method. We can exploit this primal-dual view to understand existing FL algorithms and design new algorithms. FedAvg actually exploits the usefulness of average dual states. In FedAvg, the gradient average approximates the true gradient evaluated at an average point of the client states, and the approximation error is upper-bounded by the client states difference; therefore, clients can perform multiple local steps without communication. Although in FedAvg, we do not differentiate the primal and dual space as they are the same, but the dual state average and primal state average are not equivalent for adaptive gradient methods.

Current federated adaptive gradient methods in the literature either only perform adaptive gradient steps on the server side, or ignore this primal-dual nuance when supporting local adaptive gradient steps. An early work is Reddi et al. (2020), the authors proposed applying adaptive gradients in the server-average step, while performing normal gradient descent updates locally. This method is simple to implement and gets better performance than FedAvg, but the adaptive information is not exploited during local updates; this weakens the usage of adaptive gradients. Recently, some work Karimireddy et al. (2020a); Chen et al. (2020b) exploited adaptive information during local update steps; however, a common characteristic of these methods is that they average the primal states (parameters) of the problem during the synchronization step. This will cause some problems. Firstly, since adaptive learning rates define the mirror map, updating adaptive learning rates locally makes the dual space not aligned, thus we can not average the primal states directly. Then even if the adaptive learning rates are fixed locally, the primal space might be nonlinear *w.r.t.* the dual space, *e.g.*, when we solve a constrained optimization problem. In summary, we propose two principles to apply adaptive gradients in FL. First, the local dual spaces should be aligned with each other; Second, we should average dual states.

More specifically, we propose the FL adaptive gradients framework **FedDA**, which is short for **Fed**erated **D**ual-averaging **A**daptive-gradient. In each global round of **FedDA**, the clients aggregate gradients (dual states) locally, and the server averages the dual states of the clients in the synchronization step. Local weights (primal states) are used as gradient query points in local updates and are recovered through the inverse mirror map (defined by the adaptive gradients). The global primal state is updated on the basis of the averaged dual states and the inverse mirror map. In addition, we utilize a restarting technique to make sure that all clients share the same dual space during local updates; more precisely, we refresh the adaptive gradients at every global epoch and use a fixed one in the local update. Our **FedDA** framework is general and can incorporate a large family of adaptive gradient methods to the FL setting. In particular, **FedDA-MVR**, an instantiation of our framework, achieves the best-known gradient complexity and communication complexity in the FL setting. Finally, we highlight our **contribution** as follows:

(i) We propose **FedDA**, a framework for federated adaptive gradient methods. The framework uses a restarted dual averaging technique and adapts a large family of adaptive gradient methods to the FL setting;

(ii) **FedDA-MVR**, an instantiation of our framework, obtains the gradient complexity of $\tilde{O}(\epsilon^{-1.5})$ and communication complexity of $\tilde{O}(\epsilon^{-1})$. This matches the optimal rate of non-adaptive federated algorithms and outperforms existing adaptive federated algorithms. **FedDA-MVR** uses the momentum-based variance-reduction gradient estimation, and exponential moving average of the gradient square as adaptive learning rates;

(iii) We empirically verify the efficacy of the framework **FedDA** by performing a colorrectal cancer prediction task and a classification task over the CIFAR10 and FEMNIST datasets.

**Notations.** $\nabla f(x)$ $(\nabla f^{(k)}(x))$ denotes the first-order derivatives of the function $f(x)$ $(f^{(k)}(x))$ *w.r.t.* variable $x$. $\xi$ denotes a random sample and $\nabla f(x;\xi)(\nabla f^{(k)}(x;\xi))$ is the stochastic estimate $\nabla f(x)$ $(\nabla f^{(k)}(x))$. $O(\cdot)$ is the big O notation, and $\tilde{O}(\cdot)$ hides logarithmic terms. $I_d$ denotes a $d$-dimensional identity matrix. $Diag(x)$ denotes the matrix whose diagonal is the vector $x$. $\|\cdot\|$ denotes the $\ell_2$ norm for vectors and the spectral norm for matrices, respectively. $\langle\cdot,\cdot\rangle$ denotes the Euclidean inner product. [K] denotes the set of $\{1, 2, ..., K\}$. For a random variable $X$, $\mathbb{E}[X]$ denotes its expectation.

## 2 RELATED WORKS

**Optimization Algorithms in Federated Learning.** The term Federated Learning was first coined in McMahan et al. (2017), where the task is learned from a set of distributed located clients under the coordination of a server. In the paper McMahan et al. (2017), the authors proposed the FedAvg algorithm, in which each client performs multiple steps of gradient descent with its local data and then sends the updated model to the server for averaging. The idea of FedAvg algorithm resembles the Local-SGD algorithm, which is studied in a more general distributed setting for a longer time Mangasarian & Solodov (1993). The convergence of the local-SGD method has been heavily analyzed in the literature Stich (2018); Karimireddy et al. (2019b); Dieuleveut & Patel (2019); Khaled et al. (2020); Yu et al. (2019); Woodworth et al. (2020); Woodworth (2021); Glasgow et al. (2022). Recently, Glasgow et al. (2022) proved a convergence rate of the Local-SGD under convex setting that matches the lower bound. On top of the vanilla Local-SGD, various acceleration methods are considered; we list a few representatives here. Karimireddy et al. (2020b) adopted the idea of variance reduction technique for non-distributed finite sum problems: a 'control variate' which contains historical full gradient information is used to correct the bias of local gradients. Then in Karimireddy et al. (2020a), the authors proposed a general framework (MIME) to translate a centralized optimizer into the FL setting, including adaptive gradient methods. In MIME, the states of an optimizer are fixed during local update steps and only updated at the server-average step. In Das et al. (2020); Khanduri et al. (2021b), momentum-based variance reduction is applied to the FL setting to control the noise of the stochastic gradients. In Das et al. (2020), the authors maintained a server momentum state and a client momentum state, while in Khanduri et al. (2021b), the authors maintained a momentum state and the momentum was averaged periodically similar to the primal state.

Adaptive gradient methods are also studied in the FL setting. The 'Adaptive Federated Optimization' Reddi et al. (2020) method proposed to use adaptive gradients on the server side while the local gradients are used to update the states of the adaptive gradient methods. In Chen et al. (2020b), the authors first showed the divergence of a naive local AMSGrad method that directly averages the primal states periodically. The authors then proposed Local-AMSGrad, a method in which clients update adaptive learning rates locally and average at the synchronization step. Finally, another line of research Tang et al. (2020; 2021); Lu et al. (2022); Chen et al. (2020a) considers federated adaptive learning rates through the compression approach, these methods communicate local gradients at every step, but the compression techniques are used to reduce the communication cost.

**Adaptive Gradients in the Non-distributed Learning.** Adaptive gradient methods are widely used in the non-distributed machine learning setting. The first adaptive gradient method *i.e.* Adagrad was proposed in (Duchi et al., 2011), where the method was shown to outperform SGD in the sparse gradient setting. Since Adagrad does not perform well under dense gradient setting and non-convex setting, some of its variants are proposed, such as SC-Adagra Mukkamala & Hein (2017)

and SAdagrad Chen et al. (2018b). Furthermore, Adam Kingma & Ba (2014) and YOGI Zaheer et al. (2018) proposed to use the exponential moving average instead of the arithmetic average used in Adagrad. Adam/YOGI is widely used and very successful in deep learning applications; however, Adam diverges in some settings and the gradient information quickly disappears, so AMSGrad Reddi et al. (2018) is proposed, and it applies an extra 'long term memory' variable to preserve the past gradient information to handle the convergence issue of Adam. The convergence of Adam-type methods is also studied in the literature Chen et al. (2019); Zhou et al. (2018); Liu et al. (2019); Guo et al. (2021); Huang et al. (2021). Adaptive gradient methods with good generalization performance are also proposed, such as AdamW (Loshchilov & Hutter, 2018), Padam (Chen et al., 2018a), Adabound Luo et al. (2019), Adabelief Zhuang et al. (2020) and AaGrad-Norm Ward et al. (2019).

## 3 PRELIMINARIES

In this section, we introduce some preliminaries before introducing our framework. First, we consider the following formulation of Federated Learning:

$$\min_{x \in \mathcal{X} \subset \mathbb{R}^d} \left\{ f(x) := \frac{1}{K} \sum_{k=1}^{K} \left\{ f^{(k)}(x) := \mathbb{E}_{\xi^{(k)} \sim \mathcal{D}^{(k)}}[f^{(k)}(x; \xi^{(k)})] \right\} \right\}. \tag{1}$$

which considers $K$ clients. For the $k_{th}$ client, we optimize the loss objective $f^{(k)}(x) : \mathcal{X} \to \mathbb{R}$ which is smooth and possibly non-convex, and $x$ denotes the variable of interest. $\mathcal{X} \subset \mathbb{R}^d$ is a compact and convex set. $\xi^{(k)} \sim \mathcal{D}^{(k)}$ is a random example that follows an unknown data distribution $\mathcal{D}^{(k)}$. The formulation in equation 1 includes both the homogeneous case *i.e.* $f^{(k)}(x) = f^{(j)}(x)$ for any $k, j \in [K]$, and the heterogeneous case *i.e.* $f^{(k)}(x) \neq f^{(j)}(x)$ for some $k, j \in [K]$.

Next, we introduce some basics of adaptive gradient methods from a mirror-descent perspective. Generally, mirror descent is associated with a mirror map $\Phi(x)$. Given the objective $f(x)$ and the primal state $x_t \in \mathcal{X}$ at $t_{th}$ step, we first map the primal state to the mirror space as $y_t = \nabla \Phi(x_t)$, then we perform the gradient descent step in the mirror space: $y_{t+1} = y_t - \eta \nabla f(x)$, where $\eta$ is the learning rate, finally, we map $y_{t+1}$ back to the primal space as $x_{t+1} = \arg\min_{x \in \mathcal{X}} D_\Phi(x, y_{t+1})$, where $D_\Phi(x, y)$ denotes the Bregman Divergence associated to $\Phi$, *i.e.* $D_\Phi(x, y) = f(x) - f(y) - \langle \nabla f(y), x - y \rangle$, In summary, the mirror descent step can be written as a Bregman proximal gradient step as follows:

$$x_{t+1} = \arg\min_{x \in \mathcal{X}} \eta \langle \nabla f(x_t), x \rangle + D_\Phi(x, x_t)$$

For the adaptive gradient methods, we uses the following mirror map: $\Phi(x) = \frac{1}{2} x^T H x$, where $H$ is the adaptive matrix and is positive definite. Many adaptive gradient methods can be written in the following proximal gradient descent form:

$$x_{t+1} = \arg\min_{x \in \mathcal{X}} \eta \langle \nu_t, x \rangle + \frac{1}{2}(x - x_t)^T H_t(x - x_t), \tag{2}$$

we replace the gradient $\nabla f(x)$ with the generalized gradient estimation $\nu_t$, besides, we replace $H$ with $H_t$ based on the fact that the adaptive matrix is updated at every step. Next, we show some examples of adaptive gradients methods that can be phrased as the above formulation. For the Adagrad Duchi et al. (2011) method, we set

$$\nu_t = \nabla f(x_t, \xi_t), \; H_t = Diag(\sqrt{\mu_t}), \; \mu_t = \frac{1}{t} \sum_{i=1}^{t} \nu_i^2 \tag{3}$$

For Adam Kingma & Ba (2014), we have:

$$\hat{\nu}_t = (1 - \beta_1)\nabla f(x_t, \xi_t) + \beta_1 \hat{\nu}_{t-1}, \; \hat{\mu}_t = (1 - \beta_2)\nabla f(x_t, \xi_t)^2 + \beta_2 \hat{\mu}_{t-1}$$
$$\nu_t = \hat{\nu}_t/(1 - \gamma_1^t), \; \mu_t = \hat{\mu}_t/(1 - \gamma_2^t), \; H_t = Diag(\sqrt{\mu_t} + \epsilon) \tag{4}$$

where $\beta_1, \beta_2, \gamma_1, \gamma_2$ are some constants. For other adaptive gradient methods, please refer to Huang et al. (2021).

---

**Algorithm 1 FedDA**-Server

1: **Input:** Number of global epochs $E$, tuning parameters $\{\beta_\tau\}_{i=1}^{E}$;
2: **Initialize:** Choose $x_0 \in \mathcal{X}$ and compute $\nu_0 = \frac{1}{K} \sum_{j=1}^{K} \nabla f^{(j)}(x_0, \mathcal{B}_0^{(k)})$ where $\{\mathcal{B}_0^{(k)}\}_{k=1}^{K}$ are a mini-batch of random points selected from each of $K$ clients;
3: **for** $\tau = 0$ **to** $E - 1$ **do**
4:     Server selects a set $\mathcal{S}_\tau$ of $r$ clients chosen uniformly at random w/o replacement;
5:     **for** the client $k \in \mathcal{S}_\tau$ in parallel **do**
6:         $(z_{\tau+1,I}^{(k)}, \nu_{\tau+1,I}^{(k)}) = $ **FedDA**-client$(x_\tau, \nu_\tau, H_\tau)$
7:     **end for**
8:     Compute $z_{\tau+1} = \frac{1}{r} \sum_{k \in \mathcal{S}_\tau} z_{\tau+1,I}^{(k)}$;
9:     Compute $x_{\tau+1} = \arg\min_{x \in \mathcal{X}} \{ -\langle x, z_{\tau+1} \rangle + \frac{1}{2\lambda}(x - x_\tau)^T H_\tau (x - x_\tau) \}$;
10:    Compute $\nu_{\tau+1} = \frac{1}{r} \sum_{k \in \mathcal{S}_\tau} \nu_{\tau+1,I}^{(k)}$;
11:    Compute $H_{\tau+1} = \mathcal{V}(H_\tau, z_{\tau+1})$;
12: **end for**

---

**Algorithm 2 FedDA**-Client $(x_\tau, \nu_\tau, H_\tau)$

1: **Input:** Number of local steps $I$, tuning parameters $\{\eta_{\tau+1,i}\}_{i=0}^{I-1}$, $\{\alpha_{\tau+1,i}\}_{i=1}^{I}$;
2: **Initialize:** $x_{\tau+1,0}^{(k)} = x_\tau$; $\nu_{\tau+1,0}^{(k)} = \nu_\tau$; $z_{\tau+1,0}^{(k)} = 0$;
3: **for** $i = 0$ **to** $I - 1$ **do**
4:     Compute $z_{\tau+1,i+1}^{(k)} = z_{\tau+1,i}^{(k)} - \eta_{\tau+1,i}\nu_{\tau+1,i}^{(k)}$;
5:     Compute $x_{\tau+1,i+1}^{(k)} = \arg\min_{x \in \mathcal{X}} \{ -\langle x, z_{\tau+1,i+1}^{(k)} \rangle + \frac{1}{2\lambda}(x - x_{\tau+1,0}^{(k)})^T H_\tau (x - x_{\tau+1,0}^{(k)}) \}$;
6:     Compute $\nu_{\tau+1,i+1}^{(k)} = \mathcal{U}(\nu_{\tau+1,i}^{(k)}, x_{\tau+1,i+1}^{(k)}, x_{\tau+1,i}^{(k)}; \alpha_{\tau+1,i+1}, \mathcal{B}_{\tau+1,i+1}^{(k)})$, where $\mathcal{B}_{\tau+1,i+1}^{(k)}$ is a minibatch of random samples from the client $k$;
7: **end for**
8: **Output:** Send $z_{\tau+1,I}^{(k)}, \nu_{\tau+1,I}^{(k)}$ to the server.

---

## 4 LOCAL ADAPTIVE GRADIENTS VIA DUAL AVERAGING

In this section, we introduce **FedDA**, a framework of federated adaptive gradient methods. The procedure of **FedDA** is summarized in Algorithm 1.

In Algorithm 1, we perform $E$ global steps and at each global step, we select a subset of clients for training. All selected clients at each step will run Algorithm 2. In Algorithm 2, clients receive the current model weight $x_\tau$, gradient estimation $\nu_\tau$ and adaptive gradient matrix $H_\tau$. The clients then perform $I$ local training steps: line 3- line 7 in Algorithm 2. For each step, we first accumulate the dual state in the variable $z_{\tau,i}^{(k)}$ (line 4), then we calculate the local primal state $x_{\tau,i}^{(k)}$ (line 5), which is a proximal gradient step similar to equation 2. The function of this step is to map the aggregated dual state $z_{\tau,i}^{(k)}$ back to the primal space, and we use the primal state to query the gradient to update the estimation of the gradient $\nu_{\tau,i}^{(k)}$ (line 6). Note, we use a fixed adaptive matrix $H_\tau$ during local steps, this makes the clients share the same dual space. In line 6 of Algorithm 2, we update the gradient estimation $\nu_{\tau,i}^{(k)}$. The update rule $\mathcal{U}(\cdot)$ is general, *e.g.*,the momentum-based variance reduction update equation 5 and the momentum update equation 6 as follows ($\alpha_{\tau,i}$ is some constant):

$$\nu_{\tau+1,i+1}^{(k)} = \nabla f^{(k)}(x_{\tau+1,i+1}^{(k)}, \mathcal{B}_{\tau+1,i+1}^{(k)}) + (1 - \alpha_{\tau+1,i+1})(\nu_{\tau+1,i}^{(k)} - \nabla f^{(k)}(x_{\tau+1,i}^{(k)}, \mathcal{B}_{\tau+1,i+1}^{(k)})) \tag{5}$$

and

$$\nu_{\tau+1,i+1}^{(k)} = \alpha_{\tau+1,i+1}\nabla f^{(k)}(x_{\tau+1,i+1}^{(k)}, \mathcal{B}_{\tau+1,i+1}^{(k)}) + (1 - \alpha_{\tau+1,i+1})\nu_{\tau+1,i}^{(k)} \tag{6}$$

After the client runs Algorithm 2, it returns the aggregated local dual states $z_{\tau+1,I}^{(k)}$ and the local gradient estimation $\nu_{\tau+1,I}^{(k)}$ to the server. The server first averages the local dual states (line 8 of

Algorithm 1) to get $z_{\tau+1}$. We can average local dual states as all clients have a common dual space. The server then calculates the new primal states $x_{\tau+1}$ as in line 9 of Algorithm 1. Next, the gradient estimation $\nu_\tau$ is also updated by averaging local states (line 10 of Algorithm 1). Finally, we update the adaptive matrix $H_\tau$ (line 11 of Algorithm 1). The update rule $\mathcal{V}$ is general, *e.g.*,

$$\mu_{\tau+1} = \beta_{\tau+1} z_{\tau+1}^2 / \eta_{\tau+1,I-1}^2 + (1 - \beta_{\tau+1})\mu_{\tau+1}, \ H_{\tau+1} = Diag(\sqrt{\mu_{\tau+1}} + \epsilon) \tag{7}$$

and

$$\mu_{\tau+1} = \beta_{\tau+1} \|z_{\tau+1}\| / \eta_{\tau+1,I-1} + (1 - \beta_{\tau+1})\mu_{\tau+1}, \ H_{\tau+1} = (\mu_{\tau+1} + \epsilon)I_d \tag{8}$$

where we set $\mu_0 = 0$, $\epsilon$ is some constant. In summary, Algorithm 1 aggregates and averages dual states at each global round. The adaptive matrix $H_\tau$ is fixed during local updates and is refreshed on the server side at each global round. Since the algorithm uses a new mirror map (adaptive gradient matrix) at each global round, we call our framework to be restarted dual averaging.

*Remark* 1. In contrast to our dual-averaging strategy, some existing adaptive FL algorithms Praneeth Karimireddy et al. (2020) average the local primal states. In the unconstrained case, the primal and dual spaces are linear with each other, but in the constrained case, the linearity does not exist, and the averaging in the primal space and dual space is not equivalent. As we show in the subsequent theoretical analysis, dual averaging leads to the convergence in the constrained case.

*Remark* 2. Note that we use the averaged dual states $z_{\tau+1}$ when we update the adaptive matrix (line 11 of Algorithm 1). An alternative choice is to use the most recent gradient Praneeth Karimireddy et al. (2020). In comparison, the dual state aggregates information of whole round and offers smoother estimation of the problem's local curvature. Another possible choice, as used in the Local-AMSGrad method Chen et al. (2020b), is to update the state of the adaptive matrix $\mu_\tau$ (see equation 7) locally and then average in the server synchronization step. The limitation of this approach is that $\mu_\tau$ is not linear *w.r.t* gradient, and thus averaging $\mu_\tau$ does not offer a linear speed-up *w.r.t.* the number of clients; in contrast, the dual state satisfies linearity.

*Remark* 3. By choosing different update rules $\mathcal{U}$ and $\mathcal{V}$, we can create many variants of FedDA. An representative is **FedDA-MVR**, in which we update $\nu_{\tau,i}^{(k)}$ with momentum-based variance reduction ( equation 5) and the adaptive matrix $H_\tau$ with an exponential average of the square of the gradient ( equation 7). In the subsequent discussion, we focus on this variant and perform both theoretical and empirical analysis.

## 5 THEORETICAL ANALYSIS

In this section, we provide the theoretical analysis of our **FedDA** framework; more specifically, we focus on the analysis of **FedDA-MVR**. FedDA-MVR uses equation 7 to update the adaptive matrix $H_\tau$ and equation 5 to update the gradient estimation $\nu_{\tau,i}^{(k)}$. We first state the assumptions we need in our analysis:

### 5.1 SOME MILD ASSUMPTIONS

**Assumption 1** (Bounded Client Heterogeneity). The difference of gradients between different workers are bounded:
$$\|\nabla f^{(k)}(x) - \nabla f^{(\ell)}(x)\|^2 \le \zeta^2, \ \forall k, \ell \in [K].$$

We measure the heterogeneity of the clients in terms of gradient dissimilarity. The above assumption or its similar form is also exploited in the analysis of other Federated Learning Algorithms, such as in Khanduri et al. (2021a); Das et al. (2020).

**Assumption 2.** The function $f(x)$ is bounded from below in $\mathcal{X}$, *i.e.,* $f^* = \inf_{x \in \mathcal{X}} f(x)$.

**Assumption 3** (Unbiased and Bounded-variance Stochastic Gradient). The stochastic gradients are unbiased with bounded variance, *i.e.*
$$\mathbb{E}[\nabla f^{(k)}(x; \xi^{(k)})] = \nabla f^{(k)}(x)$$

and there exists a constant $\sigma$ such that

$$\mathbb{E}\|\nabla f^{(k)}(x; \xi^{(k)}) - \nabla f^{(k)}(x)\|^2 \le \sigma^2, \ \forall \xi^{(k)} \sim \mathcal{D}^{(k)}, \ \forall k \in [K].$$

Assumption 2 guarantees the feasibility of the Federated Learning problem equation 1, and Assumption 3 is widely used in stochastic optimization analysis.

**Assumption 4.** The adaptive matrix $H_\tau$ is symmetric positive definite, *i.e.* there exists a constant $\rho > 0$ such that

$$H_\tau \succeq \rho I_d \succ 0, \ \forall t \geq 1,$$

In our analysis, we assume the adaptive matrix is positive definite, and this requirement can be easily satisfied by many adaptive gradient methods. Firstly, most adaptive gradient methods always have non-negative adaptive learning rates, such as equation 3 and equation 4. To make it positive, we can add a bias term $\epsilon$ such as in the Adam update rule equation 4.

**Assumption 5** (Sample Gradient Lipschitz Smoothness). The stochastic functions $f^{(k)}(x, \xi^{(k)})$ with $\xi^{(k)} \sim \mathcal{D}^{(k)}$ for all $k \in [K]$, satisfy the mean squared smoothness property, i.e, we have

$$\mathbb{E}\|\nabla f^{(k)}(x; \xi^{(k)}) - \nabla f^{(k)}(y; \xi^{(k)})\|^2 \leq L^2\|x - y\|^2 \quad \text{for all } x, y \in \mathbb{R}^d,$$

The smoothness assumption above is a slightly stronger requirement than the standard smooth condition, but this assumption is widely used in the analysis of variance reduction methods, such as SPIDER Fang et al. (2018) and STORM Cutkosky & Orabona (2019).

**Assumption 6.** All clients participate in the training at each step, *i.e.* choose $r = K$ in Algorithm 1.

We make the full participation assumption to simplify the exposition of the theoretical results. All the results presented can be easily generalized to the partial participation case.

## 5.2 CONVERGENCE PROPERTY OF **FED-MVR**

In this subsection, we provide the convergence property of our **FedDA-MVR** variant. For convenience of discussion, we redefine the subscript $t = \tau I + i$, *i.e.* we denote the $t$ step as the $i$ local step in the $\tau$ global round. Similarly, we denote the total number of running steps as $T = EI$. We analyze our algorithm through the following measure:

$$\mathcal{G}_t = \frac{\rho^2}{\lambda^2\eta_t^2}\|\tilde{x}_t - \tilde{x}_{t+1}\|^2 + \|\bar{\nu}_t - \nabla f(\tilde{x}_t)\|^2 \tag{9}$$

where $\bar{\nu}_t$ denotes the average gradient estimation at the $t$ step and $\tilde{x}_t$ denotes the virtual global primal state at the $t$ step (see Section 9.2 in the appendix for formal definitions). In Remark 7 of the appendix, we discuss the intuition of the measure $\mathcal{G}_t$. In particular, in the unconstrained case *i.e.* when $\mathcal{X} = R^d$, the measure upper-bounds the square norm of the gradient. Therefore, the convergence of our measure $\mathcal{G}_t$ means the convergence to a first-order stationary point. Now, we are ready to provide the main result of our convergence theorem.

**Theorem 5.1.** *In Algorithm 1, we choose the parameters as* $\kappa = \dfrac{\rho K^{2/3}}{\lambda L}$, $c = \dfrac{96\lambda^2 L^2}{K\rho^2} + \dfrac{\rho}{72\kappa^3\lambda LI^2}$,

$w_t = \max\left\{48^3 I^6 K^2 - t - I, 14^3 K^{0.5}\right\}$, $\lambda > 0$, *and choose* $\eta_t = \dfrac{\kappa}{(\omega_t + t + I)^{1/3}}$, *then we have:*

$$\frac{1}{T}\sum_{t=0}^{T-1}\mathbb{E}[\mathcal{G}_t] \leq \left[\frac{96LI^2}{T} + \frac{2L}{K^{2/3}T^{2/3}}\right](f(x_0) - f^*) + \left[\frac{72I^4}{bT} + \frac{3I^2}{2bK^{2/3}T^{2/3}}\right]\sigma^2$$

$$+ 192^2 \times \left(\frac{48I^2}{T} + \frac{1}{K^{2/3}T^{2/3}}\right) \times \left(\frac{\sigma^2}{4b_1} + \frac{2\zeta^2}{21}\right)\log(T+1).$$

Note, by choosing a proper value of local updates $I$ and using a minibatch of samples for the first iteration to decrease the noise, our result matches the best known convergence rate for stochastic federated gradient methods Khanduri et al. (2021a), *i.e.* our algorithms has gradient complexity of $\tilde{O}(\epsilon^{-1.5})$ and communication complexity of $\tilde{O}(\epsilon^{-1})$, moreover we achieve linear speed up *w.r.t* the number of clients $K$. More formally, we have the following corollary:

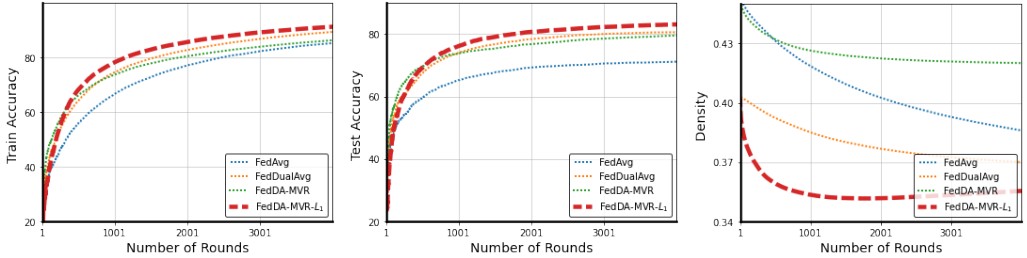

Figure 1: Results for the PATHMNIST Yang et al. (2021) dataset. Plots show the Train Accuracy, Test Accuracy, Density vs Number of Rounds ($E$ in Algorithm 1) respectively. The post-fix of $L_1$ means we consider the $L_1$ constraints. $I$ is chosen as 5.

**Corollary 1.** *Suppose in Algorithm 1, we set $I = O((T/K^2)^{1/6})$, and use sample minibatch of size $O(I^2)$ in the initialization, then we have:*

$$\frac{1}{T}\sum_{t=1}^{T}\left(\mathbb{E}[\mathcal{G}_t]\right) = \tilde{O}(\frac{1}{K^{2/3}T^{2/3}})$$

*and to reach an $\epsilon$-stationary point, we need to make $\tilde{O}(\epsilon^{-1.5}/K)$ number of steps and need $\tilde{O}(\epsilon^{-1})$ number of communication rounds.*

## 6 NUMERICAL EXPERIMENTS

In this section, we perform numerical experiments to verify the efficacy of the proposed adaptive federated learning framework *i.e.* **FedDA**. More specifically, we consider the variant of **FedDA-MVR** here, and defer experiments for other variants to Section 8 of the appendix. We performed two sets of experiments. In the first experiment, we consider a biomedical prediction task: predicting the survival of colorectal cancer. In this task, we impose a $L_1$ sparsity constraint. $L_1$ constraint improves the explainability of the model, which is essential for biomedical applications. Then in the second experiment, we consider a federated multiclass image classification task. More specifically, we consider two datasets: CIFAR10 Krizhevsky et al. (2009) and FEMNIST Caldas et al. (2018). All experiments are run on a machine with an Intel Xeon Gold 6248 CPU and 4 Nvidia Tesla V100 GPUs. The code is written in Pytorch. We simulate the Federated Learning environment through the Pytorch.distributed package.

### 6.1 COLORRECTAL CANCER SURVIVAL PREDICTION WITH SPARSE CONSTRAINTS

In this subsection, we consider a colorrectal cancer prediction task on the PATHMNST dataset Yang et al. (2021); Kather et al. (2019), which contains 9 different classes. It has 89996 training images, and we equally randomly split the training set into 10 clients. We used the original test set for the metric. In this task, we impose the $L_1$ sparsity constraint to improve the explainability of the model.

In this task, we compare with the following baselines: FedAvg McMahan et al. (2017) and FedDualAvg Yuan et al. (2021). FedDualAvg is a recently proposed federated algorithm that deals with composite optimization problems. In FedDualAvg, clients maintain dual states locally, but adaptive gradients are not applied. For our FedDA-MVR, we train with and without the $L_1$ constraint. We tune the hyper-parameters for each method and choose the best setting. The results are summarized in Figure 1, the plots are averaged over 5 independent runs and then smoothed. In Figure 1, FedDualAvg and FedDA-MVR-$L_1$ consider the $L_1$ constraint, while FedAvg and FedDA-MVR do not. We show results of Train/Test Accuracy and also the number of non-zero (below a threshold) elements in the parameter (*i.e.* the rightmost plot in Figure 1). As shown in the plots, FedDA-MVR-$L_1$ outperforms unconstrained FedDA-MVR in all metrics. This shows the importance of considering constrained problems in Federated Learning. Furthermore, FedDA-MVR-$L_1$ also outperforms FedAvg and FedDualAvg in all metrics. This shows that our algorithm can effectively exploit adaptive gradient information in the constrained case. For more details of this experiment, such as the hyper-parameter choices, please refer to Section 8 of the appendix.

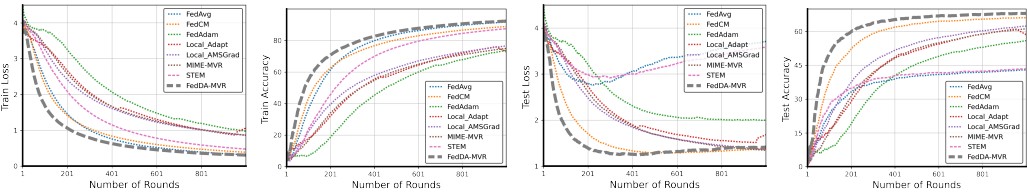

Figure 2: Results for CIFAR10 dataset. From left to right, we show Train Loss, Train Accuracy, Test Loss, Test Accuracy *w.r.t* the number of rounds (E in Algorithm 1), respectively. $I$ is chosen as 5.

Figure 3: Results for FEMNIST dataset. From left to right, we show Train Loss, Train Accuracy, Test Loss, Test Accuracy *w.r.t* the number of rounds (E in Algorithm 1), respectively. $I$ is chosen as 5.

## 6.2 IMAGE CLASSIFICATION TASK WITH CIFAR10 AND FEMNIST

In this subsection, we consider an unconstrained image classification task for both homogeneous and heterogeneous cases. More specifically, we consider two datasets: CIFAR10 Krizhevsky et al. (2009) and FEMNIST Caldas et al. (2018). CIFAR10 is a widely used image classification benchmark dataset which contains 50000 training images, and we construct both homogeneous and heterogeneous cases based on it. For the homogeneous case, we uniformly randomly distribute them into 10 clients. The Heterogeneous case is deferred to Appendix 8.3. FEMNIST is a Federated dataset of handwritten digits; it contains hand-written digits of 3550 users (we randomly sample 500 users in our experiments). Data distribution of FEMNIST is heterogeneous for different writing styles of people.

In this task, we compare our method with the following baselines: the non-adaptive methods: FedAvg McMahan et al. (2017), FedCM Xu et al. (2021), STEM Khanduri et al. (2021a) and adaptive methods: FedAdam (Reddi et al., 2020), Local-Adapt Wang et al. (2021), Local-AMSGrad Chen et al. (2020b), MIME-MVR Praneeth Karimireddy et al. (2020). For all methods, we tune their hyper-parameters to find the best setting. The results are summarized in Figure 2 (CIFAR10) and Figure 3 (FEMNIST), the plots are averaged over 5 runs and then smoothed. As shown in the figures, our FedDA-MVR outperforms all baselines. In addition, the FedAvg algorithm has competitive training performance; however, it tends to overfit the training data severely. Then we observe that adaptive methods in general get better train and test performance. Finally, the superior performance of our method compared with the three adaptive baselines shows that our method exploits adaptive information better; for example, MIME-MVR also exploits the momentum-based variance reduction technique, but it fixes all optimizer states during local updates, in contrast, we only fix the adaptive matrix but update the momentum $\nu_t^{(k)}, k \in [K]$ at every step. For more details, including the hyper-parameter selection, please refer to Section 8 of the appendix.

## 7 CONCLUSION

In this paper, we proposed the FedDA framework to incorporate adaptive gradients into the Federated Learning environment. More specifically, we adopted the Mirror Descent view of adaptive gradients, furthermore, we proposed to maintain and average the dual states in the training, meanwhile we fixed the adaptive matrix during local training such that the dual space is shared by all clients. We also analyze the convergence property of our Framework: for the variant FedDA-MVR, we proved that it reaches an $\epsilon$-optimal stationary point with $\tilde{O}(\epsilon^{-1.5})$ gradient queries and $\tilde{O}(\epsilon^{-1})$ communication rounds, these results match the best known gradient complexity and communication complexity of stochastic federated algorithms under the non-convex case. Finally, we validate our algorithm for both constrained and unconstrained tasks. The numerical results show the superior performance of our algorithm compared to various baseline methods.

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

# 8 MORE EXPERIMENTAL DETAILS AND RESULTS

In this section, we add additional experiments. In Section 8.1, we consider more variants of **FedDA** besides **FedDA-MVR**. More specifically, we consider four variants of **FedDA**. We introduce two cases for the update of the adaptive matrix $H_\tau$ in equation 7 and equation 8 and we denote them as case 1 and case 2, similarly, we denote equation 5 and equation 6 as case 1 and case 2 of gradient estimation respectively. So we have four different variants, we denote them as **FedDA-$i$-$j$**, for $i, j \in \{1, 2\}$, where $i$ shows the choice of gradient estimation and $j$ shows the choice of adaptive matrix update rule. Note **FedDA-MVR** corresponds to **FedDA-1-1** as we choose Case 1 of gradient estimation and Case 1 of adaptive matrix update in Algorithm 1. We also introduce more details such as the hyper-parameter choices. Then in Section 8.2, we perform some ablation studies and compare our FedDA with other baselines in more detail; In Section 8.3, we include experiments when we construct heterogeneous dataset from CIFAR10; Finally in Section 8.4, we show the form of our FedDA when $I = 1$, i.e. no local steps.

## 8.1 OTHER VARIANTS OF **FEDDA** FOR TASKS IN SECTION 6

### 8.1.1 COLORRECTAL CANCER SURVIVAL PREDICTION WITH SPARSE CONSTRAINTS

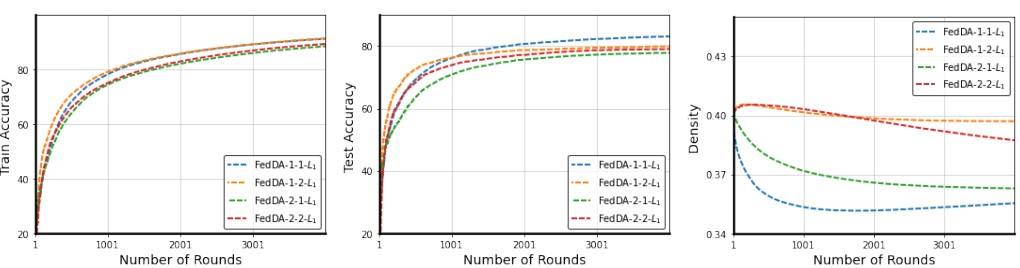

Figure 4: Results for the PATHMNIST dataset. Plots show the Train Accuracy, Test Accuracy, Density vs Number of Rounds ($E$ in Algorithm 1) respectively. The post-fix of $L_1$ means we consider the $L_1$ constraints.

In this task, we use a 4-layer convolutional neural network with 32 filters at each layer. We have 10 clients and run 20000 steps ($T$), average states with interval 5 ($I$) and use mini-batch size of 16. Besides, we calculate density with threshold 0.01. For other hyper-parameters, we perform grid search and choose the best setting for each method. More specifically, for the SGD method, we use learning rate 0.01; for the FedDualAvg algorithm, we use local learning rate 0.1, global learning rate 0.1, $L_1$ constraint 0.01; for our FedDA-MVR, we use learning rate 0.01, $w$ as 100000, $c$ as 5000000, $\beta$ as 0.999 and $\tau$ as 0.01, for the $L_1$ regularized version FedDA-MVR-$L_1$, we also add $L_1$ constraint 0.01. For other variants of FedDA: for FedDA-2-1, we use learning rate 0.001, $\alpha$ as 0.9, $\beta$ as 0.999, $\tau$ as 0.01; for FedDA-1-2, we use learning rate 1, $w$ as 10000, $c$ as 200, $\beta$ as 0.999, $\tau$ as 0.001, $L_1$ constraint 0.01; for FedDA-2-2, we use learning rate 0.01, $\alpha$ 0.9, $\beta$ as 0.999, $\tau$ as 0.01, $L_1$ constraint 0.01. The experimental results for different variants of FedDA is summarized in Figure 4. As shown by the plots, all variants of FedDA get good performance, but we find FedDA-MVR (FedDA-1-1) gets most sparse model as measured by the density metric.

### 8.1.2 IMAGE CLASSIFICATION TASK WITH CIFAR10 AND FEMNIST

In this unconstrained federated image classification task, we use a 4-layer convolutional neural network with 64 filters at each layer. For the FEMNIST dataset, we randomly sample 50 users at each global round. We run 20000 steps ($T$), average states with interval 5 ($I$) and use mini-batch size of 16. For other hyper-parameters, we perform grid search and choose the best setting for each method. In the CIFAR10 related experiments, for the SGD method, we use learning rate 0.005; for the FedCM algorithm, we use learning rate 0.01, momentum coefficient $\alpha$ as 0.9; for the FedAdam algorithm, we use local learning rate 0.001, global learning rate 0.002, momentum coefficient 0.9, coefficient for adaptive matrix $\beta$ as 0.999; for the Local-Adapt algorithm, we use local learning rate 0.001, global learning rate 0.002, momentum coefficient 0.9, coefficient for adaptive matrix $\beta$ as

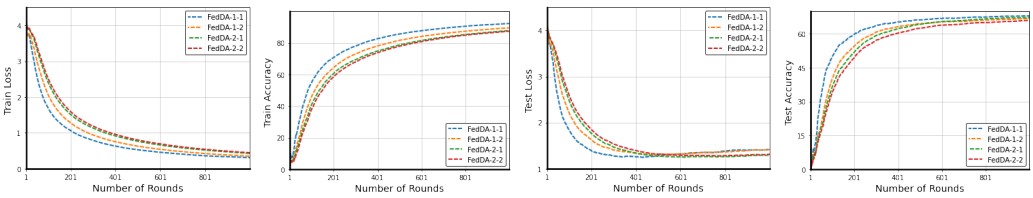

Figure 5: Results for CIFAR10 dataset. From left to right, we show Train Loss, Train Accuracy, Test Loss, Test Accuracy *w.r.t* the number of global rounds (E in Algorithm 1), respectively.

0.999; for the Local-AMSGrad algorithm, we use learning rate 0.001, momentum coefficient 0.9, adaptive matrix coefficient 0.999; for the MIME-MVR algorithm, we use learning rate 0.1, $w$ 100, $c$ as 2000; for the STEM algorithm, we use learning rate 0.1, $w$ 100 and $c$ 2000; for our FedDA-MVR, we use learning rate 0.02, $w$ as 10000, $c$ as 1000000, $\beta$ as 0.999 and $\tau$ as 0.01. For other variants of FedDA: for FedDA-2-1, we use learning rate 0.001, $\alpha$ as 0.9, $\beta$ as 0.999, $\tau$ as 0.01; for FedDA-1-2, we use learning rate 1, $w$ as 5000, $c$ as 100, $\beta$ as 0.999, $\tau$ as 0.01; for FedDA-2-2, we use learning rate 0.01, $\alpha$ 0.9, $\beta$ as 0.999, $\tau$ as 0.01.

Figure 6: Results for FEMNIST dataset. From left to right, we show Train Loss, Train Accuracy, Test Loss, Test Accuracy *w.r.t* the number of global rounds (E in Algorithm 1), respectively.

Then in the FEMNIST experiments, for the SGD method, we use learning rate 0.1; for the FedCM algorithm, we use learning rate 0.1, momentum coefficient $\alpha$ as 0.9; for the FedAdam algorithm, we use local learning rate 0.02, global learning rate 0.04, momentum coefficient 0.9, coefficient for adaptive matrix $\beta$ as 0.999; for the Local-Adapt algorithm, we use local learning rate 0.02, global learning rate 0.02, momentum coefficient 0.9, coefficient for adaptive matrix $\beta$ as 0.999; for the Local-AMSGrad algorithm, we use learning rate 0.0005, momentum coefficient 0.9, adaptive matrix coefficient 0.999; for the MIME-MVR algorithm, we use learning rate 1, $w$ 10000, $c$ as 400; for the STEM algorithm, we use learning rate 1, $w$ 10000 and $c$ 400; for our FedDA-MVR, we use learning rate 0.02, $w$ as 10000, $c$ as 1000000, $\beta$ as 0.999 and $\tau$ as 0.01. For other variants of FedDA: for FedDA-2-1, we use learning rate 0.001, $\alpha$ as 0.9, $\beta$ as 0.999, $\tau$ as 0.01; for FedDA-1-2, we use learning rate 1, $w$ as 5000, $c$ as 100, $\beta$ as 0.999, $\tau$ as 0.01; for FedDA-2-2, we use the learning rate 0.01, $\alpha$ 0.9, $\beta$ as 0.999, $\tau$ as 0.01.

The experimental results for different variants of FedDA is summarized in Figure 5 and 6. As shown by plots, all variants of FedDA get good performance. FedDA-MVR (FedDA-1-1) gets the best performance in most metrics, we observe that its test loss show some extent of overfitting in the late training stage.

## 8.2 MORE DISCUSSION OF EXPERIMENTAL RESULTS

In this subsection, we make more detailed comparison between our FedDA and other baselines (The experiments are over homogeneous CIFAR10 dataset). In Figure 7, we compare FedCM with FedDA-2-1 and FedDA-2-2 for different values of local steps $I$. Since FedDA-2-1 and FedDA-2-2 do not use variance reduction acceleration, the superior performance shows the effectiveness of using adaptive gradients in our framework. Next, In Figure 8, we compare Local-AMSGrad vs FedDA-2-1 for different values of $I$, FedDA-2-1 outperforms Local-AMSGrad for all $I$ and with a greater margin for larger $I$. Note both Local-AMSGrad and FedDA-2-1 use Adam-style adaptive gradients (equation 6 and equation 7) and have same communication cost per epoch. In Figure 9,

we compare FedAdam and Local-Adapt with FedDA-2-1. All methods use Adam-style adaptive gradients. FedAdam only performs adaptive gradients over the server, Local-Adapt performs both local and global adaptive gradients, but the state of the local adaptive gradient is refreshed per epoch. We have two observations: First, the Local-Adapt method has very marginal improvement over FedAdam, which shows the restarted strategy used by Local-Adapt is less effective than our method; Second, both FedAdam and Local-Adapt benefit little from increasing the $I$ value (compared to our FedDA-2-1). For FedAdam, this shows the limitation of only applying adaptive gradients at the server level. Finally, in Figure 10, we change $I$ for all four variants of our FedDA. As shown by the figure, our framework can benefit from more local steps.

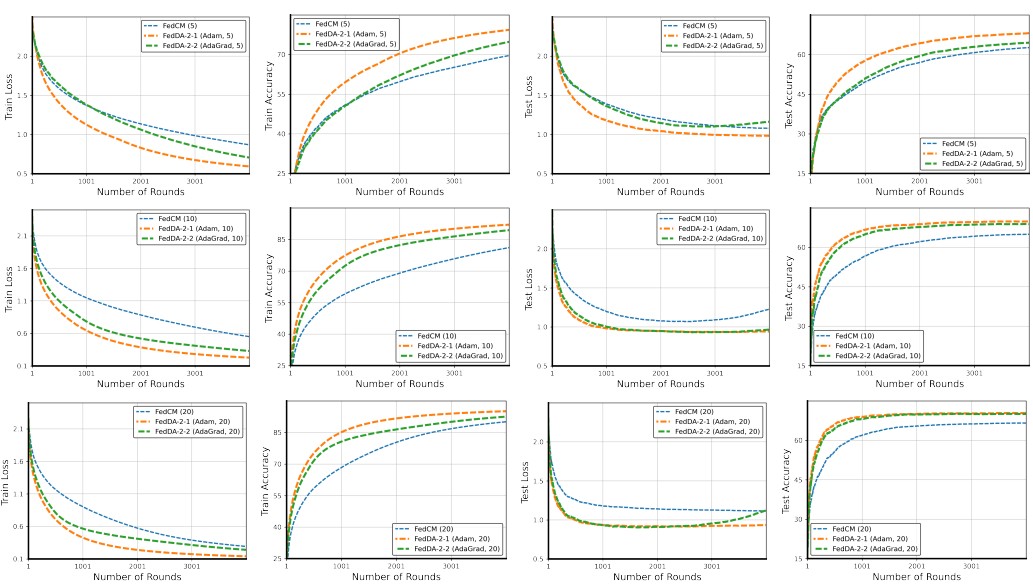

Figure 7: Comparison between FedCM vs FedDA-2-1 and FedDA-2-2. From top to bottom, we show $I = 5, 10, 20$ respectively. The number inside the parentheses is the value of $I$.

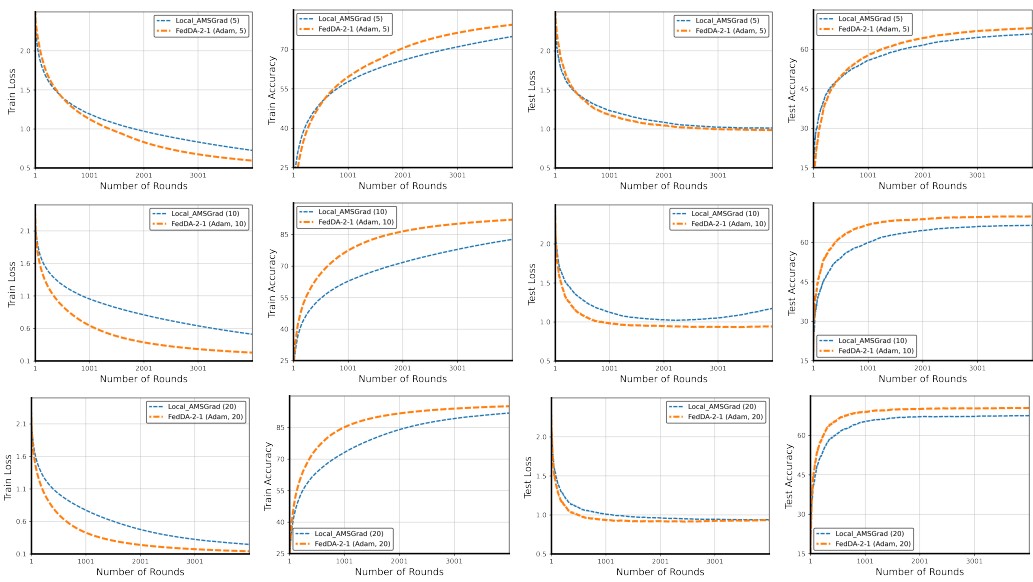

Figure 8: Comparison between Local-AMSGrad vs FedDA-2-1. From top to bottom, we show $I = 5, 10, 20$ respectively. The number inside the parentheses is the value of $I$.

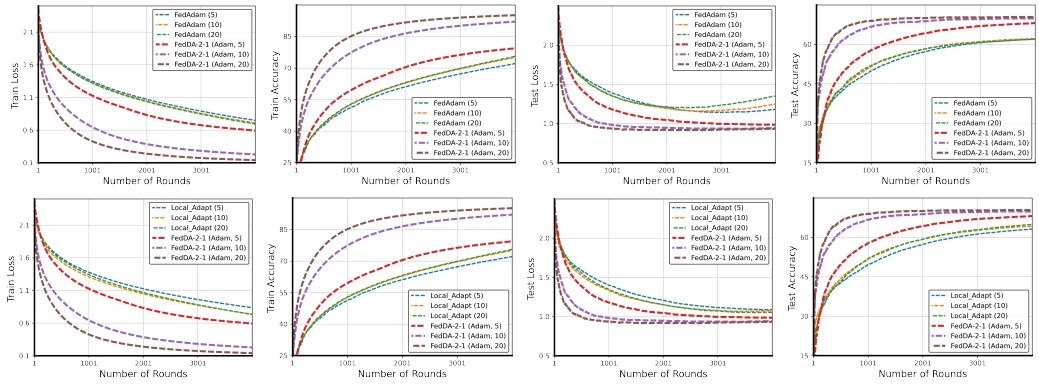

Figure 9: Comparison between FedAdam and Local-Adapt vs FedDA-2-1. The number inside the parentheses is the value of $I$.

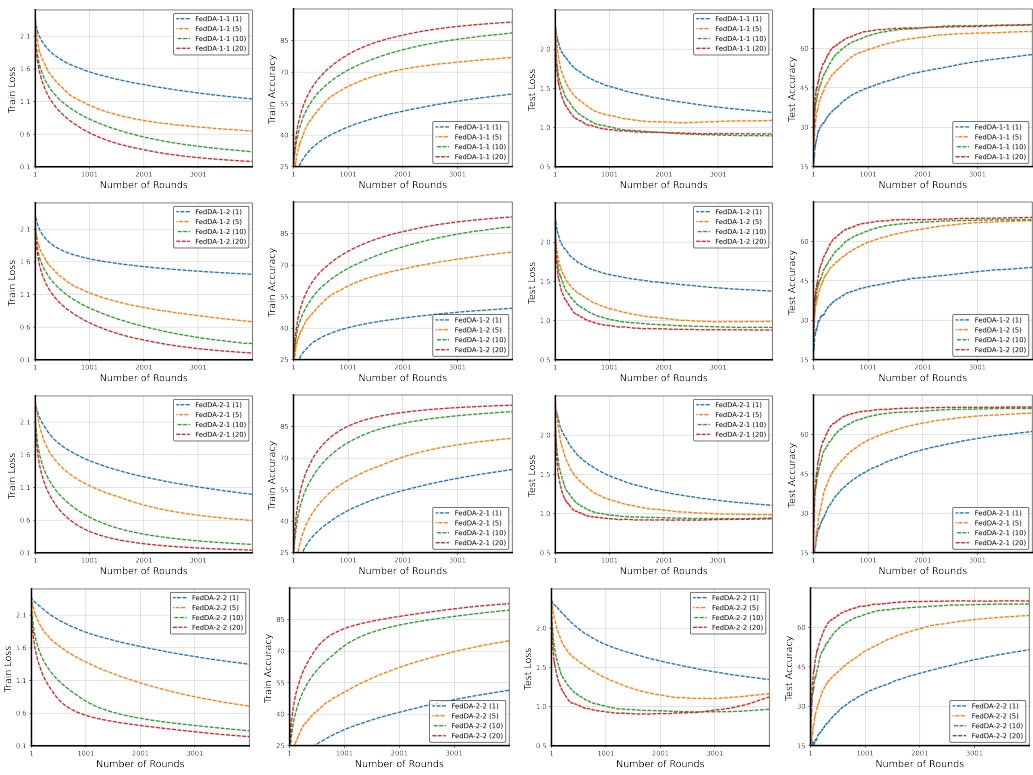

Figure 10: Ablation study of local steps $I$. From top row to the bottom row, we show results for FedDA-1-1, FedDA-1-2, FedDA-2-1 and FedDA-2-2. The number inside the parentheses is the value of $I$.

## 8.3 IMAGE CLASSIFICATION TASK WITH HETEROGENEOUS CIFAR10

Besides the FEMNIST dataset considered in the main text, we validate our **FedDA** over a manually crafted heterogeneous dataset based on CIFAR10 (Similar approaches are seen in literature such as Das et al. (2020); Yang et al. (2022)). Note For CIFAR10, we have 50000 training images and 5000 images per class. We create heterogeneity in the training set as follows: Suppose we have 10 clients, for $i_{th}$ client, we distribute $\rho$-percent samples of $i_{th}$ class, and $(1-\rho)/9$-percent samples of other classes, where $0 < \rho \leq 1$. Note for $\rho$ close to 1, the $i_{th}$ client will be dominated by images of $i_{th}$ class, thus the data distribution among clients will be very different. In our experiments, we choose

$\rho = 0.8$. This means the $i_{th}$ client has 4000 images of $i_{th}$ class and 111 images of other classes. This creates a high level of heterogeneity. Note that we use the original test set of CIFAR10. The results are summarized in Figure 11. Note we compare with the same set of baseline methods as in the homogeneous case. **FedDA-$i$-$j$** represents different variants of our framework, in particular, **FedDA-1-1** represents **FedDA-MVR**. As shown by the figure, all of our variants outperform the baselines. FedAvg suffers most due to heterogeneity and is much worse than other methods. Compared to the homogeneous case (Figure 2 and Figure 5), the methods overfit to the training data slightly more.

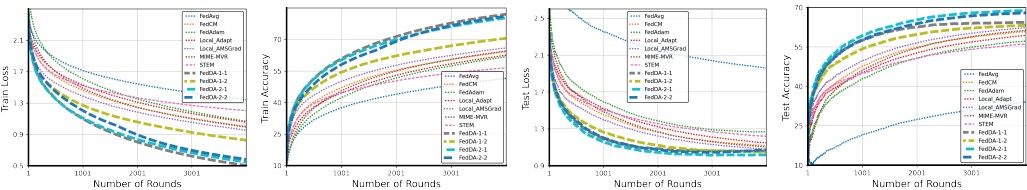

Figure 11: Results for heterogeneous CIFAR10 dataset. From left to right, we show Train Loss, Train Accuracy, Test Loss, Test Accuracy *w.r.t* the number of rounds (E in Algorithm 1), respectively. $I$ is chosen as 5.

For hyper-parameters, we perform grid search and choose the best setting for each method. For the SGD method, we use learning rate 0.01; for the FedCM algorithm, we use learning rate 0.01, momentum coefficient $\alpha$ as 0.9; for the FedAdam algorithm, we use local learning rate 0.001, global learning rate 0.002, momentum coefficient 0.9, coefficient for adaptive matrix $\beta$ as 0.999; for the Local-Adapt algorithm, we use local learning rate 0.001, global learning rate 0.002, momentum coefficient 0.9, coefficient for adaptive matrix $\beta$ as 0.999; for the Local-AMSGrad algorithm, we use learning rate 0.001, momentum coefficient 0.9, adaptive matrix coefficient 0.999; for the MIME-MVR algorithm, we use learning rate 0.1, $w$ 100, $c$ as 2000; for the STEM algorithm, we use learning rate 0.1, $w$ 100 and $c$ 2000; for our FedDA-MVR/FedDA-1-1, we use learning rate 0.02, $w$ as 10000, $c$ as 1000000, $\beta$ as 0.999 and $\tau$ as 0.01. For other variants of FedDA: for FedDA-2-1, we use learning rate 0.001, $\alpha$ as 0.9, $\beta$ as 0.999, $\tau$ as 0.01; for FedDA-1-2, we use learning rate 1, $w$ as 5000, $c$ as 100, $\beta$ as 0.999, $\tau$ as 0.01; for FedDA-2-2, we use learning rate 0.01, $\alpha$ 0.9, $\beta$ as 0.999, $\tau$ as 0.01.

### 8.4 A SPECIAL CASE OF FEDDA: $I = 1$

To better illustrate the structure of our FedDA, we give the form of a special case in this subsection, *i.e.* $I = 1$. The pseudo code is summarized in Algorithm 3: at each epoch, the server first gets new primal state through equation 2 (line 4); then each client (we assume full participation for simplicity) updates gradient estimate $\nu_\tau$ locally (line 6), and the server average these states (line 8), the adaptive matrix is also updated by the server (line 8).

---

**Algorithm 3 FedDA-Distributed**

1: **Input:** Number of global epochs $E$, tuning parameters $\{\alpha_\tau, \beta_\tau, \eta_\tau\}_{i=1}^E$;
2: **Initialize:** Choose $x_0 \in \mathcal{X}$ and compute $\nu_0 = \frac{1}{K} \sum_{j=1}^K \nabla f^{(j)}(x_0, \mathcal{B}_0^{(k)})$ where $\{\mathcal{B}_0^{(k)}\}_{k=1}^K$ are a mini-batch of random points selected from each of $K$ clients;
3: **for** $\tau = 0$ **to** $E - 1$ **do**
4:     Compute $x_{\tau+1} = \arg\min_{x \in \mathcal{X}} \{\eta_{\tau+1}\langle x, \nu_\tau\rangle + \frac{1}{2\lambda}(x - x_\tau)^T H_\tau(x - x_\tau)\}$;
5:     **for** the client $k \in [K]$ in parallel **do**
6:         Compute $\nu_{\tau+1}^{(k)} = \mathcal{U}(\nu_\tau, x_{\tau+1}, x_\tau; \alpha_{\tau+1}, \mathcal{B}_{\tau+1}^{(k)})$, where $\mathcal{B}_{\tau+1}^{(k)}$ is a minibatch of random samples from the client $k$;
7:     **end for**
8:     Compute $\nu_{\tau+1} = \frac{1}{K} \sum_{k \in [K]} \nu_{\tau+1}^{(k)}$ and $H_{\tau+1} = \mathcal{V}(H_\tau, \nu_\tau)$;
9: **end for**

---

## 9 PROOF OF THEOREMS

In this section, we provide the convergence analysis of our algorithm.

### 9.1 PRELIMINARY PROPOSITIONS

**Proposition 1.** *Let* $\{\theta_k\}, k \in K$ *be* $K$ *vectors. Then the following are true:* $||\theta_i + \theta_j||^2 \leq (1 + \lambda)||\theta_i||^2 + (1 + \frac{1}{\lambda})||\theta_j||^2$ *for any* $a > 0$ *and* $||\sum_{k=1}^{K} \theta_k||^2 \leq K \sum_{k=1}^{K} ||\theta_k||^2$

**Proposition 2.** *For a finite sequence* $z^{(k)} \in \mathbb{R}^d$ *for* $k \in [K]$ *define* $\bar{z} := \frac{1}{K} \sum_{k=1}^{K} z^{(k)}$, *we then have* $\sum_{k=1}^{K} ||z^{(k)} - \bar{z}||^2 \leq \sum_{k=1}^{K} ||z^{(k)}||^2$.

**Proposition 3.** *Let* $z_0 > 0$ *and* $z_1, z_2, \ldots, z_T \geq 0$. *We have* $\sum_{t=1}^{T} \frac{z_t}{z_0 + \sum_{i=t}^{t} z_i} \leq \log(1 + \frac{\sum_{i=1}^{t} z_i}{z_0})$.

These propositions are standard results. For proofs, the reader can refer to Lemma 3 of Karimireddy et al. (2019a) for Proposition 1 and Lemma C.1 and Lemma C.2 in Khanduri et al. (2021a) for Propositions 2 and 3.

### 9.2 PRELIMINARY LEMMAS IN LOCAL UPDATES

We first introduce some notation. For $0 \leq i \leq I$, we denote:

$$\psi_{\tau,i}^{(k)}(x) = -\langle x, z_{\tau,i}^{(k)} \rangle + \frac{1}{2\lambda}(x - x_{\tau,0}^{(k)})^T H_{\tau-1}(x - x_{\tau,0}^{(k)}), \tag{10}$$

then, by definition (Line 4 of Algorithm 2), we have:

$$x_{\tau,i}^{(k)} = \arg\min_{x \in \mathcal{X}} \psi_{\tau,i}^{(k)}(x), \tag{11}$$

we also define

$$\tilde{\psi}_{\tau,i}(x) = -\langle x, \bar{z}_{\tau,i} \rangle + \frac{1}{2\lambda}(x - x_{\tau,0})^T H_{\tau-1}(x - x_{\tau,0}), \tag{12}$$

where $\bar{z}_{\tau,i} = \frac{1}{K} \sum_{k=1}^{K} z_{\tau,i}^{(k)}$ is the virtual average of $z_{\tau,i}^{(k)}$ and $x_{\tau,0} = x_\tau$. Then we define

$$\tilde{x}_{\tau,i} = \arg\min_{x \in \mathcal{X}} \tilde{\psi}_{\tau,i}(x), \tag{13}$$

*Remark* 4. In Algorithm 1, at each epoch $\tau$, we only sample $r$ clients from the $K$ clients to perform an update. For $k \notin \mathcal{S}_\tau$, we define the relevant variables for convenience of analysis and they are not really calculated.

*Remark* 5. Note that the global primal state $\tilde{x}_i$ is not the arithmetic mean of the local states $x_i^{(k)}$ in general.

Finally, we also define

$$\tilde{d}_{\tau,i} = \frac{1}{\eta_{\tau,i}}(\tilde{x}_{\tau,i} - \tilde{x}_{\tau,i+1}), \ d_{\tau,i}^{(k)} = \frac{1}{\eta_{\tau,i}}(x_{\tau,i}^{(k)} - x_{\tau,i+1}^{(k)}), k \in [K], i \in [I], \tag{14}$$

Furthermore, recall that by the procedure of Algorithm 2 (line 6), we have

$$\bar{\nu}_{\tau,i} = \frac{1}{\eta_{\tau,i}}(\bar{z}_{\tau,i} - \bar{z}_{\tau,i+1}), \ \nu_{\tau,i}^{(k)} = \frac{1}{\eta_{\tau,i}}(z_{\tau,i}^{(k)} - z_{\tau,i+1}^{(k)}), \ k \in [K], \ i \in [I], \tag{15}$$

*Remark* 6. When it is clear from the context, we omit the global epoch $\tau$ in the subscript of the definitions, *i.e.* we use $\psi_i^{(k)}(x)$, $\tilde{\psi}_i(x)$, $x_i^{(k)}$, $\tilde{x}_i$, $\tilde{d}_i$, $d_i^{(k)}$, $\bar{\nu}_i$, $\nu_i^{(k)}$ and $H$.

Next, we introduce the following lemma related to local updates. We omit the global epoch number $\tau$ in the subscript.

**Lemma 1.** *For any* $i \in [I]$ *and* $k \in [K]$, *we have the following inequalities be satisfied:*

   *1.* $\lambda\langle \nu_i^{(k)}, d_i^{(k)} \rangle \geq \rho||d_i^{(k)}||^2, \lambda||\nu_i^{(k)}|| \geq \rho||d_i^{(k)}||$

2. $\lambda \langle \bar{\nu}_i, \tilde{d}_i \rangle \geq \rho ||\tilde{d}_i||^2, \lambda ||\bar{\nu}_i|| \geq \rho ||\tilde{d}_i||$;

3. $\lambda ||z_i^{(k)} - \bar{z}_i|| \geq \rho ||x_i^{(k)} - \tilde{x}_i||$;

*Proof.* The first and second claims follow similar derivations, and we provide only the derivations for the first claim. First, if $i = 1$, we have

$$x_1^{(k)} = \underset{x \in \mathcal{X}}{\arg \min} \; - \langle x, z_1^{(k)} \rangle + \frac{1}{2\lambda}(x - x_0^{(k)})^T H (x - x_0^{(k)}),$$

by the first-order optimality condition, we have:

$$\langle -z_1^{(k)} + \frac{1}{\lambda} H(x_1^{(k)} - x_0^{(k)}), u - x_1^{(k)} \rangle \geq 0, \; \forall \, u \in \mathcal{X},$$

choose $u = x_0^{(k)}$ and use the fact that $z_1^{(k)} = -\eta_0 \nu_0$, we have:

$$\eta_0 ||\nu_0^{(k)}|| \times ||x_0^{(k)} - x_1^{(k)}|| \geq \eta_0 \langle \nu_0^{(k)}, x_0^{(k)} - x_1^{(k)} \rangle \geq \frac{1}{\lambda}(x_1^{(k)} - x_0^{(k)})^T H(x_1^{(k)} - x_0^{(k)}) \geq \frac{\rho}{\lambda}||x_0^{(k)} - x_1^{(k)}||^2$$

we use the Cauchy-Schwartz inequality in the leftmost inequality and use the strong convexity assumption of the adaptive matrix in the rightmost inequality, we get the result in the lemma.

Next if $i > 0$, by the definition of $\psi_i^{(k)}(x)$, we have:

$$\psi_i^{(k)}(x_{i+1}^{(k)}) - \psi_i^{(k)}(x_i^{(k)}) = -\langle z_i^{(k)}, x_{i+1}^{(k)} - x_i^{(k)} \rangle + \frac{1}{2\lambda}(x_{i+1}^{(k)} - x_i^{(k)})^T H(x_{i+1}^{(k)} + x_i^{(k)} - 2x_0^{(k)})$$

$$(16)$$

Then by the definition of $x_i^{(k)}$, and the first order optimality condition, we have

$$\langle -z_i^{(k)} + \frac{1}{\lambda} H(x_i^{(k)} - x_0^{(k)}), u - x_i^{(k)} \rangle \geq 0, \; \forall \, u \in \mathcal{X},$$

if we pick $u = x_{i+1}^{(k)}$, we have $-\langle z_i^{(k)}, x_{i+1}^{(k)} - x_i^{(k)} \rangle \geq -\frac{1}{\lambda}(x_{i+1}^{(k)} - x_i^{(k)})^T H(x_i^{(k)} - x_0^{(k)})$, plug this inequality to equation 16, we have:

$$\psi_i^{(k)}(x_{i+1}^{(k)}) - \psi_i^{(k)}(x_i^{(k)})$$
$$\geq -\frac{1}{\lambda}(x_{i+1}^{(k)} - x_i^{(k)})^T H(x_i^{(k)} - x_0^{(k)}) + \frac{1}{2\lambda}(x_{i+1}^{(k)} - x_i^{(k)})^T H(x_{i+1}^{(k)} + x_i^{(k)} - 2x_0^{(k)})$$
$$\geq \frac{1}{2\lambda}(x_{i+1}^{(k)} - x_i^{(k)})^T H(x_{i+1}^{(k)} - x_i^{(k)})$$

Similarly for $\psi_{i+1}^{(k)}$, we have:

$$\psi_{i+1}^{(k)}(x_{i+1}^{(k)}) - \psi_{i+1}^{(k)}(x_i^{(k)}) = -\langle z_{i+1}^{(k)}, x_{i+1}^{(k)} - x_i^{(k)} \rangle + \frac{1}{2\lambda}(x_{i+1}^{(k)} - x_i^{(k)})^T H(x_{i+1}^{(k)} + x_i^{(k)} - 2x_0^{(k)})$$

and by the definition of $x_{i+1}^{(k)}$ and the first order optimality condition, we can get

$$\langle -z_{i+1}^{(k)} + \frac{1}{\lambda} H(x_{i+1}^{(k)} - x_0^{(k)}), u - x_{i+1}^{(k)} \rangle \geq 0, \; \forall \, u \in \mathcal{X},$$

pick $u = x_i^{(k)}$, we have $-\langle z_{i+1}^{(k)}, x_{i+1}^{(k)} - x_i^{(k)} \rangle \leq -\frac{1}{\lambda}(x_{i+1}^{(k)} - x_i^{(k)})^T H(x_{i+1}^{(k)} - x_0^{(k)})$, plug this inequality to the above equality, we have:

$$\psi_{i+1}^{(k)}(x_{i+1}^{(k)}) - \psi_{i+1}^{(k)}(x_i^{(k)})$$
$$\leq -\frac{1}{\lambda}(x_{i+1}^{(k)} - x_i^{(k)})^T H(x_{i+1}^{(k)} - x_0^{(k)}) + \frac{1}{2\lambda}(x_{i+1}^{(k)} - x_i^{(k)})^T H(x_{i+1}^{(k)} + x_i^{(k)} - 2x_0^{(k)})$$
$$\leq -\frac{1}{2\lambda}(x_{i+1}^{(k)} - x_i^{(k)})^T H(x_{i+1}^{(k)} - x_i^{(k)})$$

Next, by definition of $\psi_i^{(k)}(x)$ and $\psi_{i+1}^{(k)}(x)$, we have:

$$\psi_{i+1}^{(k)}(x_{i+1}^{(k)}) - \psi_{i+1}^{(k)}(x_i^{(k)}) = \psi_i^{(k)}(x_{i+1}^{(k)}) - \psi_i^{(k)}(x_i^{(k)}) + \eta_i \langle \nu_i^{(k)}, x_{i+1}^{(k)} - x_i^{(k)} \rangle$$

Finally, we combine the above relations and have:

$$\eta_i ||\nu_i^{(k)}|| \times ||x_i^{(k)} - x_{i+1}^{(k)}|| \geq \eta_i \langle \nu_i^{(k)}, x_i^{(k)} - x_{i+1}^{(k)} \rangle \geq \frac{1}{\lambda}(x_{i+1}^{(k)} - x_i^{(k)})^T H(x_{i+1}^{(k)} - x_i^{(k)}) \geq \rho ||x_i^{(k)} - x_{i+1}^{(k)}||^2$$

we use the Cauchy-Schwartz inequality in the leftmost inequality and use the strong convexity assumption of the adaptive matrix in the rightmost inequality, we get the result in the claim of the lemma.

Next, we prove the third claim, by the definition of $\psi_{i+1}^{(k)}$, we have:

$$\psi_{i+1}^{(k)}(x_{i+1}^{(k)}) - \psi_{i+1}^{(k)}(\tilde{x}_{i+1}) = -\langle z_{i+1}^{(k)}, x_{i+1}^{(k)} - \tilde{x}_{i+1} \rangle + \frac{1}{2\lambda}(x_{i+1}^{(k)} - \tilde{x}_{i+1})^T H(x_{i+1}^{(k)} + \tilde{x}_{i+1} - 2x_0^{(k)})$$

By the definition of $x_{i+1}^{(k)}$ and first order optimality condition, we have

$$\langle -z_{i+1}^{(k)} + \frac{1}{\lambda}H(x_{i+1}^{(k)} - x_0^{(k)}), u - x_{i+1}^{(k)} \rangle \geq 0, \ \forall \, u \in \mathcal{X},$$

pick $u = \tilde{x}_{i+1}$, we have $-\langle z_{i+1}^{(k)}, x_{i+1}^{(k)} - \tilde{x}_{i+1} \rangle \leq -\frac{1}{\lambda}(x_{i+1}^{(k)} - \tilde{x}_{i+1})^T H(x_{i+1}^{(k)} - x_0^{(k)})$. Plug this inequality back to the above inequality, we have:

$$\psi_{i+1}^{(k)}(x_{i+1}^{(k)}) - \psi_{i+1}^{(k)}(\tilde{x}_{i+1})$$
$$\leq -\frac{1}{\lambda}(x_{i+1}^{(k)} - \tilde{x}_{i+1})^T H(x_{i+1}^{(k)} - x_0^{(k)}) + \frac{1}{2\lambda}(x_{i+1}^{(k)} - \tilde{x}_{i+1})^T H(x_{i+1}^{(k)} + \tilde{x}_{i+1} - 2x_0^{(k)})$$
$$\leq -\frac{1}{2\lambda}(x_{i+1}^{(k)} - \tilde{x}_{i+1})^T H(x_{i+1}^{(k)} - \tilde{x}_{i+1})$$

Then for $\tilde{\psi}_{i+1}(x)$, we have:

$$\tilde{\psi}_{i+1}^{(k)}(x_{i+1}^{(k)}) - \tilde{\psi}_{i+1}(\tilde{x}_{i+1}) = -\langle \bar{z}_{i+1}, x_{i+1}^{(k)} - \tilde{x}_{i+1} \rangle + \frac{1}{2\lambda}(x_{i+1}^{(k)} - \tilde{x}_{i+1})^T H(x_{i+1}^{(k)} + \tilde{x}_{i+1} - 2\tilde{x}_0)$$

By the definition of $\tilde{x}_{i+1}$ and first order optimality condition, we have:

$$\langle -\bar{z}_{i+1} + \frac{1}{\lambda}H(\tilde{x}_{i+1} - \tilde{x}_0), u - \tilde{x}_{i+1} \rangle \geq 0, \ \forall \, u \in \mathcal{X},$$

pick $u = x_{i+1}^{(k)}$, we have $-\langle \bar{z}_{i+1}, x_{i+1}^{(k)} - \tilde{x}_{i+1} \rangle \geq -\frac{1}{\lambda}(x_{i+1}^{(k)} - \tilde{x}_{i+1})^T H(\tilde{x}_{i+1} - \tilde{x}_0)$. Plug this inequality back to the above inequality, we have:

$$\tilde{\psi}_{i+1}(x_{i+1}^{(k)}) - \tilde{\psi}_{i+1}(\tilde{x}_{i+1})$$
$$\geq -\frac{1}{\lambda}(x_{i+1}^{(k)} - \tilde{x}_{i+1})^T H(\tilde{x}_{i+1} - \tilde{x}_0) + \frac{1}{2\lambda}(x_{i+1}^{(k)} - \tilde{x}_{i+1})^T H(x_{i+1}^{(k)} + \tilde{x}_{i+1} - 2\tilde{x}_0)$$
$$\geq \frac{1}{2\lambda}(x_{i+1}^{(k)} - \tilde{x}_{i+1})^T H(x_{i+1}^{(k)} - \tilde{x}_{i+1})$$

Next, since we have $x_0^{(k)} = \tilde{x}_0$, then by the definition of $\psi_{i+1}^{(k)}(x)$ and $\tilde{\psi}_{i+1}(x)$ we have:

$$\psi_{i+1}^{(k)}(x_{i+1}^{(k)}) - \psi_{i+1}^{(k)}(\tilde{x}_{i+1}) = \tilde{\psi}_{i+1}(x_{i+1}^{(k)}) - \tilde{\psi}_{i+1}(\tilde{x}_{i+1}) - \langle z_{i+1}^{(k)} - \bar{z}_{i+1}, x_{i+1}^{(k)} - \tilde{x}_{i+1} \rangle$$

Next, we combine the above relations and have:

$$||z_{i+1}^{(k)} - \bar{z}_{i+1}|| \times ||x_{i+1}^{(k)} - \tilde{x}_{i+1}|| \geq \langle z_{i+!}^{(k)} - \bar{z}_{i+1}, x_{i+1}^{(k)} - \tilde{x}_{i+1} \rangle$$
$$\geq \frac{1}{\lambda}(x_{i+1}^{(k)} - \tilde{x}_{i+1})^T H(x_{i+1}^{(k)} - \tilde{x}_{i+1}) \geq \rho ||x_{i+1}^{(k)} - \tilde{x}_{i+1}||^2$$

where the first inequality is by the Cauchy-Schwartz inequality and the last inequality is by the positive definiteness of $H$. This concludes the proof of the first inequality in the lemma. □

### 9.3 STATE CONSENSUS ERROR

As each client performs local update, the states *i.e.* $z_{\tau,i}^{(k)}$ and $\nu_{\tau,i}^{(k)}$ drift away, the following lemmas bound this difference. We omit the global epoch number $\tau$ in the subscript.

**Lemma 2.** *For each $0 \leq i \leq I$, and suppose iterates $z_i^{(k)}$, $k \in [K]$ are generated from Algorithm 2, we have:*

$$\sum_{k=1}^{K} \mathbb{E}\|z_i^{(k)} - \bar{z}_i\|^2 \leq (I-1) \sum_{\ell=1}^{i-1} \eta_\ell^2 \sum_{k=1}^{K} \mathbb{E}\|\nu_\ell^{(k)} - \bar{\nu}_\ell\|^2,$$

*where the expectation is w.r.t the stochasticity of the algorithm.*

*Proof.* Based on Algorithm 2, we have $z_0^{(k)} = \bar{z}_0 = 0$, the inequality in the lemma holds trivially. Otherwise, we have

$$z_i^{(k)} = -\sum_{\ell=0}^{i-1} \eta_\ell \nu_\ell^{(k)} \quad \text{and} \quad \bar{z}_i = -\sum_{\ell=0}^{i-1} \eta_\ell \bar{\nu}_\ell.$$

So we have:

$$\sum_{k=1}^{K} \|z_i^{(k)} - \bar{z}_i\|^2 = \sum_{k=1}^{K} \Big\| \sum_{\ell=1}^{i-1} \big( \eta_\ell \nu_\ell^{(k)} - \eta_\ell \bar{\nu}_\ell \big) \Big\|^2 \leq (I-1) \sum_{\ell=1}^{i-1} \eta_\ell^2 \sum_{k=1}^{K} \|\nu_\ell^{(k)} - \bar{\nu}_\ell\|^2$$

where the equality uses the fact $\nu_0^{(k)} = \nu_0$ for $k \in [K]$, the inequality uses the Proposition 1 and the fact that we have $i \leq I$. We get the claim in the lemma by taking expectation on both sides of the above inequality. This completes the proof. $\qquad\square$

**Lemma 3.** *For $i \in [I]$, we have:*

$$\sum_{k=1}^{K} \|d_i^{(k)} - \tilde{d}_i\|^2 \leq \frac{4\lambda^2(I-1)}{\rho^2 \eta_i^2} \sum_{\ell=1}^{i} \eta_\ell^2 \sum_{k=1}^{K} \mathbb{E}\|\nu_\ell^{(k)} - \bar{\nu}_\ell\|^2$$

*where the expectation is w.r.t the stochasticity of the algorithm.*

*Proof.* Firstly, when $i = 0$, $x_0^{(k)} = \tilde{x}_0$, $z_1^{(k)} = \bar{z}_1$, so we have $x_1^{(k)} = \tilde{x}_1$ by Line 5 of Algorithm 2, and then we have $\eta_0 d_0^{(k)} = x_0^{(k)} - x_1^{(k)} = \tilde{x}_0 - \tilde{x}_1 = \eta_t \tilde{d}_0$, the inequality in the lemma holds trivially.

Next when $i > 0$, we have:

$$\eta_i^2 \|d_i^{(k)} - \tilde{d}_i\|^2 = \|x_i^{(k)} - x_{i+1}^{(k)} - (\tilde{x}_i - \tilde{x}_{i+1})\|^2 \leq 2\|x_i^{(k)} - \tilde{x}_i\|^2 + 2\|x_{i+1}^{(k)} - \tilde{x}_{i+1}\|^2$$

$$\leq \frac{2\lambda^2}{\rho^2} \big( \|z_i^{(k)} - \bar{z}_i\|^2 + \|z_{i+1}^{(k)} - \bar{z}_{i+1}\|^2 \big)$$

The last inequality uses claim 3 of Lemma 1. Sum over $k \in [K]$ and use Lemma 2, we have:

$$\rho^2 \eta_i^2 \sum_{k=1}^{K} \|d_i^{(k)} - \tilde{d}_i\|^2 \leq 2\lambda^2(I-1) \sum_{\ell=1}^{i-1} \eta_\ell^2 \sum_{k=1}^{K} \mathbb{E}\|\nu_\ell^{(k)} - \bar{\nu}_\ell\|^2 + 2\lambda^2(I-1) \sum_{\ell=1}^{i} \eta_\ell^2 \sum_{k=1}^{K} \mathbb{E}\|\nu_\ell^{(k)} - \bar{\nu}_\ell\|^2$$

$$\leq 4\lambda^2(I-1) \sum_{\ell=1}^{i} \eta_\ell^2 \sum_{k=1}^{K} \mathbb{E}\|\nu_\ell^{(k)} - \bar{\nu}_\ell\|^2$$

This completes the proof. $\qquad\square$

## 9.4 DESCENT LEMMA

In this subsection, we bound the descent of function value $f(\tilde{x}_{\tau,i})$ over the virtual sequence $\tilde{x}_{\tau,i}$.

**Lemma 4.** *Suppose that the sequence $\{x_{\tau,i}^{(k)}\}_{i=0}^{I-1}$ be generated from Algorithm 2, then we have*

$$f(x_{\tau+1}) \leq f(x_\tau) - \sum_{i=0}^{I-1} \left( \frac{3\rho\eta_{\tau+1,i}}{4\lambda} - \frac{\eta_{\tau+1,i}^2 L}{2} \right) \|\tilde{d}_{\tau+1,i}\|^2 + \sum_{i=0}^{I-1} \frac{\lambda\eta_{\tau+1,i}}{\rho} \left\| \bar{e}_{\tau+1,i} \right\|^2,$$

*where $\bar{e}_{\tau,i} = \bar{\nu}_{\tau,i} - \frac{1}{K}\sum_{k=1}^{K} \nabla f^{(k)}(\tilde{x}_{\tau,i})$.*

*Proof.* Since the function $f(x)$ is $L$-smooth, we have (we omit the global epoch number $\tau$ for ease of notation):

$$f(\tilde{x}_{i+1}) \leq f(\tilde{x}_i) + \langle \nabla f(\tilde{x}_i), \tilde{x}_{i+1} - \tilde{x}_i \rangle + \frac{L}{2}\|\tilde{x}_{i+1} - \tilde{x}_i\|^2 = f(\tilde{x}_i) - \eta_i\langle \nabla f(\tilde{x}_i), \tilde{d}_i \rangle + \frac{L\eta_i^2}{2}\|\tilde{d}_i\|^2$$

$$= f(\tilde{x}_i) - \eta_i\langle \bar{\nu}_i, \tilde{d}_i \rangle - \eta_i\langle \nabla f(\tilde{x}_i) - \bar{\nu}_i, \tilde{d}_i \rangle + \frac{L\eta_i^2}{2}\|\tilde{d}_i\|^2$$

$$\stackrel{(a)}{\leq} f(\tilde{x}_i) - (\frac{\rho\eta_i}{\lambda} - \frac{L\eta_i^2}{2})\|\tilde{d}_i\|^2 - \eta_i\langle \nabla f(\tilde{x}_i) - \bar{\nu}_i, \tilde{d}_i \rangle$$

$$\stackrel{(b)}{\leq} f(\tilde{x}_i) - \left( \frac{\rho\eta_i}{\lambda} - \frac{\eta_i^2 L}{2} \right)\|\tilde{d}_i\|^2 + \frac{\rho\eta_i}{4\lambda}\|\tilde{d}_i\|^2 + \frac{\lambda\eta_i}{\rho}\|\bar{\nu}_i - \nabla f(\tilde{x}_i)\|^2$$

$$\stackrel{(c)}{\leq} f(\tilde{x}_i) - \left( \frac{3\rho\eta_i}{4\lambda} - \frac{\eta_i^2 L}{2} \right)\|\tilde{d}_i\|^2 + \frac{\lambda\eta_i}{\rho}\|\bar{e}_i\|^2$$

In inequality (a), we use claim 1 of Lemma 1; inequality (b) uses Young's inequality; inequality (c) denotes $\bar{e}_i = \bar{\nu}_i - \frac{1}{K}\sum_{k=1}^{K} \nabla f^{(k)}(\tilde{x}_i)$.

For the $\tau$ global epoch, we sum over $i = 0$ to $I - 1$, we have:

$$f(\tilde{x}_{\tau+1,I}) \leq f(\tilde{x}_{\tau+1,0}) - \sum_{i=0}^{I-1} \left( \frac{3\rho\eta_{\tau+1,i}}{4\lambda} - \frac{\eta_{\tau+1,i}^2 L}{2} \right) \|\tilde{d}_{\tau+1,i}\|^2 + \sum_{i=0}^{I-1} \frac{\lambda\eta_{\tau+1,i}}{\rho} \left\| \bar{e}_{\tau+1,i} \right\|^2,$$

Follow the update rules in Algorithm 1 and Algorithm 2, we have $\tilde{x}_{\tau+1,0} = x_\tau$ and $\tilde{x}_{\tau+1,I} = x_{\tau+1}$. This completes the proof. $\square$

## 9.5 GRADIENT ERROR CONTRACTION

In this subsection, we bound the gradient estimation error $\bar{e}_{\tau,i}$, where we have $\bar{e}_{\tau,i} = \bar{\nu}_{\tau,i} - \frac{1}{K}\sum_{k=1}^{K} \nabla f^{(k)}(\tilde{x}_{\tau,i})$ as defined in Lemma 4, additionally, we also define the global gradient estimation error $e_\tau$ as $e_\tau = \nu_\tau - \frac{1}{K}\sum_{k=1}^{K} \nabla f^{(k)}(x_\tau) = \nu_\tau - \nabla f(x_\tau)$. Note we have $e_\tau = \bar{e}_{\tau,I} = \bar{e}_{\tau+1,0}$. We first show a fact about $\bar{e}_0$, the initial gradient estimation error.

**Lemma 5.** *For $e_0 := \nu_0 - \frac{1}{K}\sum_{k=1}^{K} \nabla f^{(k)}(x_0)$, suppose we choose mini-batch size of $|\mathcal{B}_0^{(k)}| = b, k \in [K]$, we have: $\mathbb{E}\|e_0\|^2 \leq \frac{\sigma^2}{bK}$.*

*Proof.* By line 1 of Algorithm 1, we have:

$$\mathbb{E}\|e_0\|^2 = \mathbb{E}\left\| \nu_0 - \frac{1}{K}\sum_{k=1}^{K} \nabla f^{(k)}(x_0) \right\|^2$$

$$= \mathbb{E}\left\| \frac{1}{K}\sum_{k=1}^{K} \nabla f^{(k)}(x_0; \mathcal{B}_0^{(k)}) - \frac{1}{K}\sum_{k=1}^{K} \nabla f^{(k)}(x_0) \right\|^2$$

$$\stackrel{(a)}{\leq} \frac{1}{K^2}\sum_{k=1}^{K} \mathbb{E}\left\| \nabla f^{(k)}(x_0; \mathcal{B}_0^{(k)}) - \nabla f^{(k)}(x_0) \right\|^2 \stackrel{(b)}{\leq} \frac{\sigma^2}{bK}.$$

where $(a)$ follows from the following: From the unbiased gradient assumption, we have: $\mathbb{E}\big[\nabla f^{(k)}(x_0^{(k)}; \mathcal{B}_0^{(k)})\big] = \nabla f^{(k)}(x_0^{(k)})$, for all $k \in [K]$. Moreover, the samples $\mathcal{B}_0^{(k)}$ and $\mathcal{B}_0^{(\ell)}$ at the $k^{\text{th}}$ and the $\ell^{\text{th}}$ clients are chosen uniformly randomly, and independent of each other for all $k, \ell \in [K]$ and $k \neq \ell$.

$$\mathbb{E}\left[\left\langle (x_0^{(k)}; \mathcal{B}_0^{(k)}) - \nabla f^{(k)}(x_0)), \left(\nabla f^{(\ell)}(x_0^{(\ell)}; \mathcal{B}_0^{(\ell)}) - \nabla f^{(\ell)}(\bar{x}_0))\right\rangle\right]$$

$$= \mathbb{E}\left[\left\langle \underbrace{\mathbb{E}\big[\nabla f^{(k)}(x_0^{(k)}; \mathcal{B}_0^{(k)})\big] - \nabla f^{(k)}(x_0^{(k)})\big]}_{=0}, \underbrace{\mathbb{E}\big[\nabla f^{(\ell)}(x_0^{(\ell)}; \mathcal{B}_0^{(\ell)}) - \nabla f^{(\ell)}(x_0^{(\ell)})\big]}_{=0} \right\rangle\right] = 0.$$

Inequality $(c)$ results from the bounded variance assumption. This completes the proof. $\qquad\square$

**Lemma 6.** *Define $\bar{e}_{\tau,i} := \bar{\nu}_{\tau,i} - \frac{1}{K}\sum_{k=1}^{K}\nabla f^{(k)}(\tilde{x}_{\tau,i})$, then for every $\tau \geq 1$ and $i \geq 0$, suppose $\alpha_i < 1$ and clients use batchsize $b_1$ in the training, then we have:*

$$\mathbb{E}\|\bar{e}_{\tau,i}\|^2 \leq (1-\alpha_{\tau,i})^2 \mathbb{E}\|\bar{e}_{\tau,i-1}\|^2 + \frac{40\lambda^2(I-1)L^2}{\rho^2 K^2}\sum_{\ell=1}^{i-1}\eta_{\tau,\ell}^2\sum_{k=1}^{K}\mathbb{E}\|\nu_{\tau,\ell}^{(k)} - \bar{\nu}_{\tau,\ell}\|^2$$

$$+ \frac{8\eta_{\tau,i-1}^2 L^2}{K}\mathbb{E}\|\tilde{d}_{\tau,i-1}\|^2 + \frac{4\alpha_{\tau,i}^2\sigma^2}{b_1 K}$$

*where the expectation is w.r.t the stochasticity of the algorithm.*

*Proof.* Consider the error term $\|\bar{e}_i\|^2$, $i \geq 1$ (we omit the global epoch number $\tau$ for ease of notation), we have:

$$\mathbb{E}\|\bar{e}_i\|^2 = \mathbb{E}\left\|\bar{\nu}_i - \frac{1}{K}\sum_{k=1}^{K}\nabla f^{(k)}(\tilde{x}_i)\right\|^2$$

$$= \mathbb{E}\left\|\frac{1}{K}\sum_{k=1}^{K}\nabla f^{(k)}(x_i^{(k)}; \mathcal{B}_i^{(k)}) + (1-\alpha_i)\left(\bar{\nu}_{i-1} - \frac{1}{K}\sum_{k=1}^{K}\nabla f^{(k)}(x_{i-1}^{(k)}; \mathcal{B}_i^{(k)})\right) - \frac{1}{K}\sum_{k=1}^{K}\nabla f^{(k)}(\tilde{x}_i)\right\|^2$$

$$= \mathbb{E}\left\|\frac{1}{K}\sum_{k=1}^{K}\left(\left(\nabla f^{(k)}(x_i^{(k)}; \mathcal{B}_i^{(k)}) - \nabla f^{(k)}(\tilde{x}_i)\right)\right.\right.$$

$$\left.\left. - (1-\alpha_i)\left(\nabla f^{(k)}(x_{i-1}^{(k)}; \mathcal{B}_i^{(k)}) - \nabla f^{(k)}(\tilde{x}_{i-1})\right)\right) + (1-\alpha_i)\bar{e}_{i-1}\right\|^2$$

$$= (1-\alpha_i)^2\mathbb{E}\|\bar{e}_{i-1}\|^2 + \frac{1}{K^2}\mathbb{E}\left\|\sum_{k=1}^{K}\left[\left(\nabla f^{(k)}(x_i^{(k)}; \mathcal{B}_i^{(k)}) - \nabla f^{(k)}(\tilde{x}_i)\right)\right.\right.$$

$$\left.\left. - (1-\alpha_i)\left(\nabla f^{(k)}(x_{i-1}^{(k)}; \mathcal{B}_i^{(k)}) - \nabla f^{(k)}(\tilde{x}_{i-1})\right)\right]\right\|^2$$

$$= (1-\alpha_i)^2\mathbb{E}\|\bar{e}_{i-1}\|^2 + \frac{1}{K^2}\sum_{k=1}^{K}\mathbb{E}\left\|\left(\nabla f^{(k)}(x_i^{(k)}; \mathcal{B}_i^{(k)}) - \nabla f^{(k)}(\tilde{x}_i)\right)\right.$$

$$\left. - (1-\alpha_i)\left(\nabla f^{(k)}(x_{i-1}^{(k)}; \mathcal{B}_i^{(k)}) - \nabla f^{(k)}(\tilde{x}_{i-1})\right)\right\|^2,$$

where the first equality uses the definition of $\bar{\nu}_i$; last equality follows from expanding the norm using the inner products across $k \in [K]$ and noting that the cross term is zero in expectation because of the

samples are sampled independently at different workers. Now we consider the 2nd term above:

$$
\mathbb{E}\big\|\big(\nabla f^{(k)}(x_i^{(k)};\mathcal{B}_i^{(k)}) - \nabla f^{(k)}(\tilde{x}_i)\big) - (1-\alpha_i)\big(\nabla f^{(k)}(x_{i-1}^{(k)};\mathcal{B}_i^{(k)}) - \nabla f^{(k)}(\tilde{x}_{i-1})\big)\big\|^2
$$
$$
= \mathbb{E}\big\|\big(\nabla f^{(k)}(x_i^{(k)};\mathcal{B}_i^{(k)}) - \nabla f^{(k)}(x_i^{(k)})\big) - (1-\alpha_i)\big(\nabla f^{(k)}(x_{i-1}^{(k)};\mathcal{B}_i^{(k)}) - \nabla f^{(k)}(x_{i-1}^{(k)})\big)
$$
$$
\qquad + \nabla f^{(k)}(x_i^{(k)}) - \nabla f^{(k)}(\tilde{x}_i) - (1-\alpha_i)\big(\nabla f^{(k)}(x_{i-1}^{(k)}) - \nabla f^{(k)}(\tilde{x}_{i-1})\big)\big\|^2
$$
$$
\leq 2\mathbb{E}\big\|\big(\nabla f^{(k)}(x_i^{(k)};\mathcal{B}_i^{(k)}) - \nabla f^{(k)}(x_i^{(k)})\big) - (1-\alpha_i)\big(\nabla f^{(k)}(x_{i-1}^{(k)};\mathcal{B}_i^{(k)}) - \nabla f^{(k)}(x_{i-1}^{(k)})\big)\big\|^2
$$
$$
\qquad + 2\mathbb{E}\big\|\nabla f^{(k)}(x_i^{(k)}) - \nabla f^{(k)}(\tilde{x}_i) - (1-\alpha_i)\big(\nabla f^{(k)}(x_{i-1}^{(k)}) - \nabla f^{(k)}(\tilde{x}_{i-1})\big)\big\|^2
$$

For the first term of the above inequality, we have:

$$
\mathbb{E}\big\|\big(\nabla f^{(k)}(x_i^{(k)};\mathcal{B}_i^{(k)}) - \nabla f^{(k)}(x_i^{(k)})\big) - (1-\alpha_i)\big(\nabla f^{(k)}(x_{i-1}^{(k)};\mathcal{B}_i^{(k)}) - \nabla f^{(k)}(x_{i-1}^{(k)})\big)\big\|^2
$$
$$
= \mathbb{E}\big\|(1-a_i)\big[\big(\nabla f^{(k)}(x_i^{(k)};\mathcal{B}_i^{(k)}) - \nabla f^{(k)}(x_i^{(k)})\big) - \big(\nabla f^{(k)}(x_{i-1}^{(k)};\mathcal{B}_i^{(k)}) - \nabla f^{(k)}(x_{i-1}^{(k)})\big)\big]
$$
$$
\qquad + \alpha_i\big(\nabla f^{(k)}(x_i^{(k)};\mathcal{B}_i^{(k)}) - \nabla f^{(k)}(x_i^{(k)})\big)\big\|^2
$$
$$
\leq 2(1-\alpha_i)^2\mathbb{E}\big\|\big(\nabla f^{(k)}(x_i^{(k)};\mathcal{B}_i^{(k)}) - \nabla f^{(k)}(x_{i-1}^{(k)};\mathcal{B}_i^{(k)})\big) - \big(\nabla f^{(k)}(x_i^{(k)}) - \nabla f^{(k)}(x_{i-1}^{(k)})\big)\big\|^2
$$
$$
\qquad + 2\alpha_i^2\mathbb{E}\big\|\nabla f^{(k)}(x_i^{(k)};\mathcal{B}_i^{(k)}) - \nabla f^{(k)}(x_i^{(k)})\big\|^2
$$
$$
\leq 2(1-\alpha_i)^2\mathbb{E}\big\|\nabla f^{(k)}(x_i^{(k)};\mathcal{B}_i^{(k)}) - \nabla f^{(k)}(x_{i-1}^{(k)};\mathcal{B}_i^{(k)})\big\|^2 + 2\alpha_i^2\sigma^2/b_1
$$
$$
\leq 2(1-\alpha_i)^2 L^2\mathbb{E}\|x_i^{(k)} - x_{i-1}^{(k)}\|^2 + 2a_i^2\sigma^2/b_1 \leq 2(1-\alpha_i)^2 L^2\eta_{i-1}^2\mathbb{E}\|d_{i-1}^{(k)}\|^2 + 2\alpha_i^2\sigma^2/b_1
$$
$$
\leq 4(1-\alpha_i)^2 L^2\eta_{i-1}^2\mathbb{E}\|d_{i-1}^{(k)} - \tilde{d}_{i-1}\|^2 + 4(1-\alpha_i)^2 L^2\eta_{i-1}^2\mathbb{E}\|\tilde{d}_{i-1}\|^2 + 2\alpha_i^2\sigma^2/b_1
$$

where uses Proposition 1 in the first inequality and the bounded variance assumption in the second inequality. For the second inequality, we have:

$$
\mathbb{E}\big\|\nabla f^{(k)}(x_i^{(k)}) - \nabla f^{(k)}(\tilde{x}_i) - (1-\alpha_i)\big(\nabla f^{(k)}(x_{i-1}^{(k)}) - \nabla f^{(k)}(\tilde{x}_{i-1})\big)\big\|^2
$$
$$
\stackrel{(a)}{\leq} 2\mathbb{E}\big\|\nabla f^{(k)}(x_i^{(k)}) - \nabla f^{(k)}(\tilde{x}_i)\big\|^2 + 2\mathbb{E}\big\|(1-\alpha_i)\big(\nabla f^{(k)}(x_{i-1}^{(k)}) - \nabla f^{(k)}(\tilde{x}_{i-1})\big)\big\|^2
$$
$$
\leq 2L^2\mathbb{E}\big\|x_i^{(k)} - \tilde{x}_i\big\|^2 + 2L^2(1-\alpha_i)^2\mathbb{E}\big\|x_{i-1}^{(k)} - \tilde{x}_{i-1}\big\|^2
$$
$$
\stackrel{(b)}{\leq} \frac{2\lambda^2 L^2}{\rho^2}\mathbb{E}\big\|z_i^{(k)} - \bar{z}_i\big\|^2 + \frac{2\lambda^2 L^2(1-\alpha_i)^2}{\rho^2}\mathbb{E}\big\|z_{i-1}^{(k)} - \bar{z}_{i-1}\big\|^2
$$

where (a) uses Proposition 1; (b) uses claim 3 of Lemma 1; Next, we combine the above inequalities together to get:

$$\mathbb{E}\|\bar{e}_i\|^2 \leq (1-\alpha_i)^2 \mathbb{E}\|\bar{e}_{i-1}\|^2 + \frac{4\alpha_i^2\sigma^2}{b_1 K} + \frac{8(1-\alpha_i)^2\eta_{i-1}^2 L^2}{K^2}\sum_{k=1}^{K}\mathbb{E}\|d_{i-1}^{(k)} - \tilde{d}_{i-1}\|^2$$

$$+ \frac{8(1-\alpha_i)^2\eta_{i-1}^2 L^2}{K}\mathbb{E}\|\tilde{d}_{i-1}\|^2 + \frac{4\lambda^2 L^2}{K^2\rho^2}\sum_{k=1}^{K}\mathbb{E}\|z_i^{(k)} - \bar{z}_i\|^2$$

$$+ \frac{4\lambda^2 L^2(1-\alpha_i)^2}{K^2\rho^2}\sum_{k=1}^{K}\mathbb{E}\|z_{i-1}^{(k)} - \bar{z}_{i-1}\|^2$$

$$\leq (1-\alpha_i)^2\mathbb{E}\|\bar{e}_{i-1}\|^2 + \frac{4\alpha_i^2\sigma^2}{b_1 K} + \frac{32\lambda^2(I-1)(1-\alpha_i)^2 L^2}{K^2\rho^2}\sum_{\ell=1}^{i-1}\eta_\ell^2\sum_{k=1}^{K}\mathbb{E}\|\nu_\ell^{(k)} - \bar{\nu}_\ell\|^2$$

$$+ \frac{8(1-\alpha_i)^2\eta_{i-1}^2 L^2}{K}\mathbb{E}\|\tilde{d}_{i-1}\|^2 + \frac{4\lambda^2(I-1)L^2}{K^2\rho^2}\sum_{\ell=1}^{i-1}\eta_\ell^2\sum_{k=1}^{K}\mathbb{E}\|\nu_\ell^{(k)} - \bar{\nu}_\ell\|^2$$

$$+ \frac{4\lambda^2(I-1)L^2(1-\alpha_{i-1})^2}{K^2\rho^2}\sum_{\ell=1}^{i-2}\eta_\ell^2\sum_{k=1}^{K}\mathbb{E}\|\nu_\ell^{(k)} - \bar{\nu}_\ell\|^2$$

$$\leq (1-\alpha_i)^2\mathbb{E}\|\bar{e}_{i-1}\|^2 + \frac{4\alpha_i^2\sigma^2}{b_1 K} + \frac{40\lambda^2(I-1)L^2}{K^2\rho^2}\sum_{\ell=1}^{i-1}\eta_\ell^2\sum_{k=1}^{K}\mathbb{E}\|\nu_\ell^{(k)} - \bar{\nu}_\ell\|^2 + \frac{8\eta_{i-1}^2 L^2}{K}\mathbb{E}\|\tilde{d}_{i-1}\|^2,$$

The second inequality uses Lemma 2 and Lemma 3 and the last inequality uses the assumption that $\alpha_i < 1$. This completes the proof. $\qquad\square$

**Lemma 7.** *For $\tau \geq 0$. Suppose we choose $\eta_{\tau,i} = \kappa/(\omega_i + i + \tau I)^{1/3}$, additionally, suppose $\alpha_i < 1$, $w_i \leq w_{i-1}$, $w_i \geq 2$, $\eta_{\tau,i} \leq \frac{\rho}{48\lambda L I^2}$ be satisfied, we have:*

$$\frac{\rho K}{64 L^2}\left(\frac{\mathbb{E}\|\bar{e}_{\tau+1}\|^2}{\eta_{\tau+1,I-1}} - \frac{\mathbb{E}\|\bar{e}_\tau\|^2}{\eta_{\tau,I-1}}\right) \leq -\sum_{i=0}^{I-1}\frac{3\eta_{\tau+1,i}}{2\rho}\mathbb{E}\|\bar{e}_{\tau+1,i}\|^2 + \sum_{i=0}^{I-1}\frac{\eta_{\tau+1,i}\rho}{8}\mathbb{E}\|\tilde{d}_{\tau+1,i}\|^2 + \sum_{i=0}^{I-1}\frac{\sigma^2 c^2\eta_{\tau+1,i}^3\rho}{16 L^2}$$

$$+ \frac{5I(I-1)}{4K\rho}\sum_{\ell=1}^{I}\eta_{\tau+1,\ell}\sum_{k=1}^{K}\mathbb{E}\|\nu_{\tau+1,\ell}^{(k)} - \bar{\nu}_{\tau+1,\ell}\|^2$$

*Proof.* Using Lemma 6 at the global epoch $\tau - 1$, then for $i \geq 0$ (we denote $\eta_{\tau,-1} = \eta_{\tau-1,I-1}$ for all $\tau \geq 1$), we have:

$$\frac{\mathbb{E}\|\bar{e}_{\tau,i+1}\|^2}{\eta_{\tau,i}} - \frac{\mathbb{E}\|\bar{e}_{\tau,i}\|^2}{\eta_{\tau,i-1}}$$

$$\leq \left[\frac{(1-a_{\tau,i+1})^2}{\eta_{\tau,i}} - \frac{1}{\eta_{\tau,i-1}}\right]\mathbb{E}\|\bar{e}_{\tau,i}\|^2 + \frac{40\lambda^2(I-1)L^2}{\rho^2 K^2\eta_{\tau,i}}\sum_{\ell=1}^{i}\eta_{\tau,\ell}^2\sum_{k=1}^{K}\mathbb{E}\|\nu_{\tau,\ell}^{(k)} - \bar{\nu}_{\tau,\ell}\|^2$$

$$+ \frac{8L^2\eta_{\tau,i}}{K}\mathbb{E}\|\tilde{d}_{\tau,i}\|^2 + \frac{4a_{\tau,i+1}^2\sigma^2}{\eta_{\tau,i}b_1 K}$$

$$\overset{(a)}{\leq} \left(\eta_{\tau,i}^{-1} - \eta_{\tau,i-1}^{-1} - c\eta_{\tau,i}\right)\mathbb{E}\|\bar{e}_{\tau,i}\|^2 + \frac{80\lambda^2(I-1)L^2}{\rho^2 K^2}\sum_{\ell=1}^{i}\eta_{\tau,\ell}\sum_{k=1}^{K}\mathbb{E}\|\nu_{\tau,\ell}^{(k)} - \bar{\nu}_{\tau,\ell}\|^2$$

$$+ \frac{8L^2\eta_{\tau,i}}{K}\mathbb{E}\|\tilde{d}_{\tau,i}\|^2 + \frac{4\sigma^2 c^2\eta_{\tau,i}^3}{b_1 K},$$

where inequality $(a)$ utilizes the fact that $(1-\alpha_{\tau,i})^2 \leq 1 - \alpha_{\tau,i} \leq 1$ and $a_{\tau,i+1} = c\eta_{\tau,i}^2$ for all $i \in [I]$, and the following fact: suppose we choose $\eta_{\tau,i} = \kappa/(\omega_i + i + \tau I)^{1/3}$, then for $0 \leq l \leq i < I$,

we have:

$$\frac{\eta_{\tau,l}}{\eta_{\tau,i}} = \frac{(w_i + i + \tau I)^{1/3}}{(w_l + l + \tau I)^{1/3}} = \left(1 + \frac{w_i + i - w_l - l}{w_l + l + \tau I}\right)^{1/3}$$

$$\leq \left(1 + \frac{(I-1)}{w_l + l + \tau I}\right)^{1/3} \leq 1 + \frac{(I-1)}{3(w_l + l + \tau I)} \leq 2 \tag{17}$$

The first inequality is by the fact that $0 < i - l < I - 1$, the second last inequality uses the concavity of $x^{1/3}$ as: $(x + y)^{1/3} - x^{1/3} \leq y/3x^{2/3}$, while the last inequality uses the fact that $w_l \geq 0$, $I \geq 1$, $l \geq 0$, $\tau \geq 1$.

For the difference $\eta_i^{-1} - \eta_{i-1}^{-1}$, we have:

$$\frac{1}{\eta_{\tau,i}} - \frac{1}{\eta_{\tau,i-1}} = \frac{(w_i + i + \tau I)^{1/3}}{\kappa} - \frac{(w_{i-1} + i - 1 + \tau I)^{1/3}}{\kappa}$$

$$\overset{(a)}{\leq} \frac{(w_i + i + \tau I)^{1/3}}{\kappa} - \frac{(w_i + i - 1 + \tau I)^{1/3}}{\kappa}$$

$$\overset{(b)}{\leq} \frac{1}{3\kappa(w_i + i - 1 + \tau I)^{2/3}} \overset{(c)}{\leq} \frac{2^{2/3}\kappa^2}{3\kappa^3(w_i + i + \tau I)^{2/3}} \overset{(d)}{=} \frac{2^{2/3}}{3\kappa^3}\eta_i^2 \overset{(e)}{\leq} \frac{\rho}{72\kappa^3\lambda L I^2}\eta_i, \tag{18}$$

where inequality $(a)$ is because that we choose $w_i \leq w_{i-1}$, $(b)$ results from the concavity of $x^{1/3}$ as: $(x+y)^{1/3} - x^{1/3} \leq y/(3x^{2/3})$, $(c)$ used the fact that $w_i \geq 2$, finally, $(d)$ and $(e)$ utilize the definition of $\eta_{\tau,i}$ and the condition that $\eta_{\tau,i} \leq \frac{\rho}{48\lambda L I^2}$, respectively. So if we choose $c = \frac{96\lambda^2 L^2}{K\rho^2} + \frac{\rho}{72\kappa^3\lambda L I^2}$ we have: $\eta_{\tau,i}^{-1} - \eta_{\tau,i-1}^{-1} - c\eta_{\tau,i} \leq -\frac{96\lambda^2 L^2}{K\rho^2}\eta_{\tau,i}$,

Therefore, we have:

$$\frac{\mathbb{E}\|\bar{e}_{\tau,i+1}\|^2}{\eta_{\tau,i}} - \frac{\mathbb{E}\|\bar{e}_{\tau,i}\|^2}{\eta_{\tau,i-1}} \leq -\frac{96\lambda^2 L^2 \eta_{\tau,i}}{K\rho^2}\mathbb{E}\|\bar{e}_{\tau,i}\|^2 + \frac{80\lambda^2(I-1)L^2}{\rho^2 K^2}\sum_{\ell=1}^{i}\eta_{\tau,\ell}\sum_{k=1}^{K}\mathbb{E}\|\nu_{\tau,\ell}^{(k)} - \bar{\nu}_{\tau,\ell}\|^2$$

$$+ \frac{8L^2\eta_{\tau,i}}{K}\mathbb{E}\|\tilde{d}_{\tau,i}\|^2 + \frac{4\sigma^2 c^2 \eta_{\tau,i}^3}{b_1 K},$$

Multiplying $\rho K/64\lambda L^2$ on both sides, we have:

$$\frac{\rho K}{64\lambda L^2}\left(\frac{\mathbb{E}\|\bar{e}_{\tau,i+1}\|^2}{\eta_{\tau,i}} - \frac{\mathbb{E}\|\bar{e}_{\tau,i}\|^2}{\eta_{\tau,i-1}}\right) \leq -\frac{3\lambda\eta_{\tau,i}}{2\rho}\mathbb{E}\|\bar{e}_{\tau,i}\|^2 + \frac{5\lambda(I-1)}{4K\rho}\sum_{\ell=1}^{i}\eta_{\tau,\ell}\sum_{k=1}^{K}\mathbb{E}\|\nu_{\tau,\ell}^{(k)} - \bar{\nu}_{\tau,\ell}\|^2$$

$$+ \frac{\eta_{\tau,i}\rho}{8\lambda}\mathbb{E}\|\tilde{d}_{\tau,i}\|^2 + \frac{\sigma^2 c^2 \eta_{\tau,i}^3 \rho}{16\lambda L^2 b_1}.$$

Then we sum the above inequality from 0 to $I - 1$ and get:

$$\frac{\rho K}{64\lambda L^2}\left(\frac{\mathbb{E}\|\bar{e}_{\tau,I}\|^2}{\eta_{\tau,I-1}} - \frac{\mathbb{E}\|\bar{e}_{\tau,0}\|^2}{\eta_{\tau-1,I-1}}\right) \leq -\sum_{i=0}^{I-1}\frac{3\lambda\eta_i}{2\rho}\mathbb{E}\|\bar{e}_{\tau,i}\|^2 + \sum_{i=0}^{I-1}\frac{5\lambda(I-1)}{4K\rho}\sum_{\ell=1}^{i}\eta_\ell\sum_{k=1}^{K}\mathbb{E}\|\nu_{\tau,\ell}^{(k)} - \bar{\nu}_{\tau,\ell}\|^2$$

$$+ \sum_{i=0}^{I-1}\frac{\eta_{\tau,i}\rho}{8\lambda}\mathbb{E}\|\tilde{d}_{\tau,i}\|^2 + \sum_{i=0}^{I-1}\frac{\sigma^2 c^2 \eta_{\tau,i}^3 \rho}{16\lambda L^2 b_1}$$

$$\leq -\sum_{i=0}^{I-1}\frac{3\lambda\eta_i}{2\rho}\mathbb{E}\|\bar{e}_{\tau,i}\|^2 + \frac{5\lambda I(I-1)}{4K\rho}\sum_{\ell=1}^{I}\eta_\ell\sum_{k=1}^{K}\mathbb{E}\|\nu_{\tau,\ell}^{(k)} - \bar{\nu}_{\tau,\ell}\|^2$$

$$+ \sum_{i=0}^{I-1}\frac{\eta_{\tau,i}\rho}{8\lambda}\mathbb{E}\|\tilde{d}_{\tau,i}\|^2 + \sum_{i=0}^{I-1}\frac{\sigma^2 c^2 \eta_{\tau,i}^3 \rho}{16\lambda L^2 b_1}$$

By definition, we have $\bar{e}_{\tau,0} = e_{\tau-1}$ and $\bar{e}_{\tau,I} = e_\tau$, then we get the results in the lemma by replacing $\tau$ by $\tau + 1$.

$\square$

### 9.6 DESCENT IN POTENTIAL FUNCTION

We define the potential function as follows:

$$\Phi_\tau := f(\tilde{x}_\tau) + \frac{\rho K}{64\lambda L^2} \frac{\|e_\tau\|^2}{\eta_{\tau-1,I-1}}. \tag{19}$$

Next, we characterize the descent in the potential function.

**Lemma 8.** *For any $\tau \geq 0$, we have:*

$$\mathbb{E}[\Phi_{\tau+1} - \Phi_\tau] \leq -\sum_{i=0}^{I-1} \left( \frac{5\rho\eta_{\tau+1,i}}{8\lambda} - \frac{\eta_{\tau+1,i}^2 L}{2} \right) \mathbb{E}\|\tilde{d}_i\|^2 - \frac{\lambda}{2\rho} \sum_{i=0}^{I-1} \eta_{\tau+1,i} \mathbb{E}\|\bar{e}_{\tau+1,i}\|^2$$

$$+ \frac{\sigma^2 c^2 \rho}{16\lambda L^2 b_1} \sum_{i=0}^{I-1} \eta_{\tau+1,i}^3 + \frac{5\lambda I(I-1)}{4K\rho} \sum_{i=1}^{I} \eta_{\tau+1,i} \sum_{k=1}^{K} \mathbb{E}\|\nu_{\tau+1,i}^{(k)} - \bar{\nu}_{\tau+1,i}\|^2,$$

*where the expectation is w.r.t the stochasticity of the algorithm.*

*Proof.* We can the inequality in the lemma by combining Lemma 4 and Lemma 7

$\square$

### 9.7 ACCUMULATED GRADIENT ERROR

In this subsection, we bound the gradient consensus error given by term $\sum_{k=1}^{K} \mathbb{E}\|\nu_{\tau,i}^{(k)} - \bar{\nu}_{\tau,i}\|^2$.

**Lemma 9.** *For $i \geq 1$ and $\alpha_i < 1$, we have:*

$$\sum_{k=1}^{K} \mathbb{E}\|\nu_{\tau,i}^{(k)} - \bar{\nu}_{\tau,i}\|^2 \leq (1 + \frac{1}{I}) \sum_{k=1}^{K} \mathbb{E}\|\nu_{\tau,i-1}^{(k)} - \bar{\nu}_{\tau,i-1}\|^2 + 8KIL^2\eta_{\tau,i-1}^2 \mathbb{E}\|\tilde{d}_{\tau,i-1}\|^2 + \frac{8KI\sigma^2 c^2 \eta_{\tau,i-1}^4}{b_1}$$

$$+ 16KI\zeta^2 c^2 \eta_{\tau,i-1}^4 + \frac{96\lambda^2 I^2 L^2}{\rho^2} \sum_{\ell=1}^{i-1} \eta_{\tau,\ell}^2 \sum_{k=1}^{K} \mathbb{E}\|\nu_{\tau,\ell}^{(k)} - \bar{\nu}_{\tau,\ell}\|^2$$

*where the expectation is w.r.t. the stochasticity of the algorithm.*

*Proof.* By the update rule of $\nu_i^{(k)}$ (we omit the global epoch step for convenience), we have:

$$\mathbb{E}\|\nu_i^{(k)} - \bar{\nu}_i\|^2$$

$$= \mathbb{E}\Big\| \nabla f^{(k)}(x_i^{(k)}; \mathcal{B}_i^{(k)}) + (1-\alpha_i)(\nu_{i-1}^{(k)} - \nabla f^{(k)}(x_{i-1}^{(k)}; \mathcal{B}_i^{(k)}))$$

$$- \Big( \frac{1}{K} \sum_{j=1}^{K} \nabla f^{(j)}(x_i^{(j)}; \mathcal{B}_i^{(j)}) + (1-\alpha_i)(\bar{\nu}_{i-1} - \frac{1}{K} \sum_{j=1}^{K} \nabla f^{(j)}(x_{i-1}^{(j)}; \mathcal{B}_i^{(j)})) \Big) \Big\|^2$$

$$= \mathbb{E}\Big\| (1-\alpha_i)(\nu_{i-1}^{(k)} - \bar{\nu}_{i-1}) + \nabla f^{(k)}(x_i^{(k)}; \mathcal{B}_i^{(k)}) - \frac{1}{K} \sum_{j=1}^{K} \nabla f^{(j)}(x_i^{(j)}; \mathcal{B}_i^{(j)})$$

$$- (1-\alpha_i)\Big( \nabla f^{(k)}(x_{i-1}^{(k)}; \mathcal{B}_i^{(k)}) - \frac{1}{K} \sum_{j=1}^{K} \nabla f^{(j)}(x_{i-1}^{(j)}; \mathcal{B}_i^{(j)}) \Big) \Big\|^2$$

$$\leq (1+\beta)(1-\alpha_i)^2 \mathbb{E}\Big\| \nu_{i-1}^{(k)} - \bar{\nu}_{i-1} \Big\|^2 + \Big( 1 + \frac{1}{\beta} \Big) \mathbb{E}\Big\| \nabla f^{(k)}(x_i^{(k)}; \mathcal{B}_i^{(k)}) - \frac{1}{K} \sum_{j=1}^{K} \nabla f^{(j)}(x_i^{(j)}; \mathcal{B}_i^{(j)})$$

$$- (1-\alpha_i)\Big( \nabla f^{(k)}(x_{i-1}^{(k)}; \mathcal{B}_i^{(k)}) - \frac{1}{K} \sum_{j=1}^{K} \nabla f^{(j)}(x_{i-1}^{(j)}; \mathcal{B}_i^{(j)}) \Big) \Big\|^2, \tag{20}$$

where the last inequality uses Proposition 1.

Next, we consider the second term:

$$
\mathbb{E}\left\| \nabla f^{(k)}(x_i^{(k)}; \mathcal{B}_i^{(k)}) - \frac{1}{K}\sum_{j=1}^{K} \nabla f^{(j)}(x_i^{(j)}; \mathcal{B}_i^{(j)}) \right.
$$

$$
\left. - (1-\alpha_i)\left( \nabla f^{(k)}(x_{i-1}^{(k)}; \mathcal{B}_i^{(k)}) - \frac{1}{K}\sum_{j=1}^{K} \nabla f^{(j)}(x_{i-1}^{(j)}; \mathcal{B}_i^{(j)}) \right) \right\|^2
$$

$$
\overset{(a)}{\le} 2\mathbb{E}\left\| \nabla f^{(k)}(x_i^{(k)}; \mathcal{B}_i^{(k)}) - \frac{1}{K}\sum_{j=1}^{K} \nabla f^{(j)}(x_i^{(j)}; \mathcal{B}_i^{(j)}) \right.
$$

$$
\left. - \left( \nabla f^{(k)}(x_{i-1}^{(k)}; \mathcal{B}_i^{(k)}) - \frac{1}{K}\sum_{j=1}^{K} \nabla f^{(j)}(x_{i-1}^{(j)}; \mathcal{B}_i^{(j)}) \right) \right\|^2
$$

$$
+ 2\alpha_i^2 \mathbb{E}\left\| \nabla f^{(k)}(x_{i-1}^{(k)}; \mathcal{B}_i^{(k)}) - \frac{1}{K}\sum_{j=1}^{K} \nabla f^{(j)}(x_{i-1}^{(j)}; \mathcal{B}_i^{(j)}) \right\|^2
$$

$$
\overset{(b)}{\le} 2\mathbb{E}\left\| \left( \nabla f^{(k)}(x_i^{(k)}; \mathcal{B}_i^{(k)}) - \nabla f^{(k)}(x_{i-1}^{(k)}; \mathcal{B}_i^{(k)}) \right) \right\|^2
$$

$$
+ 2\alpha_i^2 \mathbb{E}\left\| \nabla f^{(k)}(x_{i-1}^{(k)}; \mathcal{B}_i^{(k)}) - \frac{1}{K}\sum_{j=1}^{K} \nabla f^{(j)}(x_{i-1}^{(j)}; \mathcal{B}_i^{(j)}) \right\|^2
$$

$$
\overset{(c)}{\le} 2L^2 \mathbb{E}\left\| x_i^{(k)} - x_{i-1}^{(k)} \right\|^2 + 2\alpha_i^2 \mathbb{E}\left\| \nabla f^{(k)}(x_{i-1}^{(k)}; \mathcal{B}_i^{(k)}) - \frac{1}{K}\sum_{j=1}^{K} \nabla f^{(j)}(x_{i-1}^{(j)}; \mathcal{B}_i^{(j)}) \right\|^2, \quad (21)
$$

where inequality (a) uses Proposition 1; inequality (b) uses Proposition 2; inequality (c) uses the smoothness assumption.

Next, we consider the second term in equation 21 above, we have

$$\mathbb{E}\left\|\nabla f^{(k)}(x_{i-1}^{(k)};\mathcal{B}_i^{(k)}) - \frac{1}{K}\sum_{j=1}^K \nabla f^{(j)}(x_{i-1}^{(j)};\mathcal{B}_i^{(j)})\right\|^2$$

$$= \mathbb{E}\left\|\left(\nabla f^{(k)}(x_{i-1}^{(k)};\mathcal{B}_i^{(k)}) - \nabla f^{(k)}(x_{i-1}^{(k)})\right)\right.$$

$$\left. - \frac{1}{K}\sum_{j=1}^K \left(\nabla f^{(j)}(x_{i-1}^{(j)};\mathcal{B}_i^{(j)}) - \nabla f^{(j)}(x_{i-1}^{(j)})\right) + \nabla f^{(k)}(x_{i-1}^{(k)}) - \frac{1}{K}\sum_{j=1}^K \nabla f^{(j)}(x_{i-1}^{(j)})\right\|^2$$

$$\leq 2\mathbb{E}\left\|\left(\nabla f^{(k)}(x_{i-1}^{(k)};\mathcal{B}_i^{(k)}) - \nabla f^{(k)}(x_{i-1}^{(k)})\right)\right.$$

$$\left. - \frac{1}{K}\sum_{j=1}^K \left(\nabla f^{(j)}(x_{i-1}^{(j)};\mathcal{B}_i^{(j)}) - \nabla f^{(j)}(x_{i-1}^{(j)})\right)\right\|^2$$

$$+ 2\mathbb{E}\left\|\nabla f^{(k)}(x_{i-1}^{(k)}) - \frac{1}{K}\sum_{j=1}^K \nabla f^{(j)}(x_{i-1}^{(j)})\right\|^2$$

$$\overset{(a)}{\leq} 2\mathbb{E}\left\|\left(\nabla f^{(k)}(x_{i-1}^{(k)};\mathcal{B}_i^{(k)}) - \nabla f^{(k)}(x_{i-1}^{(k)})\right)\right\|^2 + 2\mathbb{E}\left\|\nabla f^{(k)}(x_{i-1}^{(k)}) - \frac{1}{K}\sum_{j=1}^K \nabla f^{(j)}(x_{i-1}^{(j)})\right\|^2$$

$$\leq 2\mathbb{E}\left\|\left(\nabla f^{(k)}(x_{i-1}^{(k)};\mathcal{B}_i^{(k)}) - \nabla f^{(k)}(x_{i-1}^{(k)})\right)\right\|^2 + 4\mathbb{E}\left\|\nabla f^{(k)}(\tilde{x}_{i-1}) - \nabla f(\tilde{x}_{i-1})\right\|^2$$

$$+ 8\mathbb{E}\left\|\nabla f^{(k)}(x_{i-1}^{(k)}) - \nabla f^{(k)}(\tilde{x}_{i-1})\right\|^2 + 8\mathbb{E}\left\|\nabla f(\tilde{x}_{i-1}) - \frac{1}{K}\sum_{j=1}^K \nabla f^{(j)}(x_{i-1}^{(j)})\right\|^2$$

$$\overset{(b)}{\leq} \frac{2\sigma^2}{b_1} + \frac{4}{K}\sum_{j=1}^K \mathbb{E}\|\nabla f^{(k)}(\tilde{x}_{i-1}) - \nabla f^{(j)}(\bar{x}_{i-1})\|^2$$

$$+ 8L^2\mathbb{E}\|x_{i-1}^{(k)} - \tilde{x}_{i-1}\|^2 + \frac{8L^2}{K}\sum_{j=1}^K \mathbb{E}\|x_{i-1}^{(j)} - \tilde{x}_{i-1}\|^2$$

$$\overset{(c)}{\leq} \frac{2\sigma^2}{b_1} + 4\zeta^2 + 8L^2\mathbb{E}\|x_{i-1}^{(k)} - \tilde{x}_{i-1}\|^2 + \frac{8L^2}{K}\sum_{j=1}^K \mathbb{E}\|x_{i-1}^{(j)} - \tilde{x}_{i-1}\|^2, \tag{22}$$

where inequality $(a)$ uses Proposition 2; inequality $(b)$ utilizes bounded variance assumption; $(c)$ uses the bounded heterogeneity assumption. Finally, substituting equation 22 and equation 21 into equation 20 and sum over all K workers, we get

$$\sum_{k=1}^K \mathbb{E}\|\nu_i^{(k)} - \bar{\nu}_i\|^2$$

$$\leq (1-\alpha_i)^2(1+\beta)\sum_{k=1}^K \mathbb{E}\|\nu_{i-1}^{(k)} - \bar{\nu}_{i-1}\|^2 + 2L^2\left(1+\frac{1}{\beta}\right)\sum_{k=1}^K \mathbb{E}\|x_i^{(k)} - x_{i-1}^{(k)}\|^2$$

$$+ \frac{4K\sigma^2}{b_1}\left(1+\frac{1}{\beta}\right)\alpha_i^2 + 8K\zeta^2\left(1+\frac{1}{\beta}\right)\alpha_i^2 + 32L^2\left(1+\frac{1}{\beta}\right)\alpha_i^2\sum_{k=1}^K \mathbb{E}\|x_{i-1}^{(k)} - \tilde{x}_{i-1}\|^2$$

$$\leq (1-\alpha_i)^2(1+\beta)\sum_{k=1}^K \mathbb{E}\|\nu_{i-1}^{(k)} - \bar{\nu}_{i-1}\|^2 + 2L^2\eta_{i-1}^2\left(1+\frac{1}{\beta}\right)\sum_{k=1}^K \mathbb{E}\|d_{i-1}^{(k)}\|^2$$

$$+ \frac{4K\sigma^2}{b_1}\left(1+\frac{1}{\beta}\right)\alpha_i^2 + 8K\zeta^2\left(1+\frac{1}{\beta}\right)\alpha_i^2 + \frac{32\lambda^2 L^2 a_i^2}{\rho^2}\left(1+\frac{1}{\beta}\right)\sum_{k=1}^K \mathbb{E}\|z_{i-1}^{(k)} - \bar{z}_{i-1}\|^2$$

where the second inequality uses claim 3 of the Lemma 1.

Next using Lemma 2, we have:

$$\sum_{k=1}^{K} \mathbb{E}\|\nu_i^{(k)} - \bar{\nu}_i\|^2 \leq (1-\alpha_i)^2(1+\beta)\sum_{k=1}^{K}\mathbb{E}\|\nu_{i-1}^{(k)} - \bar{\nu}_{i-1}\|^2 + 2L^2\eta_{i-1}^2\left(1+\frac{1}{\beta}\right)\sum_{k=1}^{K}\mathbb{E}\|d_{i-1}^{(k)}\|^2$$
$$+ \frac{4K\sigma^2}{b_1}\left(1+\frac{1}{\beta}\right)\alpha_i^2 + 8K\zeta^2\left(1+\frac{1}{\beta}\right)\alpha_i^2$$
$$+ \frac{32\lambda^2L^2a_i^2}{\rho^2}\left(1+\frac{1}{\beta}\right)(I-1)\sum_{\ell=1}^{i-1}\eta_\ell^2\sum_{k=1}^{K}\mathbb{E}\|\nu_\ell^{(k)} - \bar{\nu}_\ell\|^2 \quad (23)$$

For the second term of the above inequality, we have:

$$2L^2\eta_{i-1}^2\left(1+\frac{1}{\beta}\right)\sum_{k=1}^{K}\mathbb{E}\|d_{i-1}^{(k)}\|^2$$
$$\leq 4L^2\eta_{i-1}^2\left(1+\frac{1}{\beta}\right)\sum_{k=1}^{K}\mathbb{E}\|d_{i-1}^{(k)} - \tilde{d}_{i-1}\|^2 + 4KL^2\eta_{i-1}^2\left(1+\frac{1}{\beta}\right)\mathbb{E}\|\tilde{d}_{i-1}\|^2$$
$$\leq \frac{16\lambda^2L^2(I-1)}{\rho^2}\left(1+\frac{1}{\beta}\right)\sum_{\ell=1}^{i-1}\eta_\ell^2\sum_{k=1}^{K}\mathbb{E}\|\nu_\ell^{(k)} - \bar{\nu}_\ell\|^2 + 4KL^2\eta_{i-1}^2\left(1+\frac{1}{\beta}\right)\mathbb{E}\|\tilde{d}_{i-1}\|^2$$

where the first inequality uses Proposition 1 and the second inequality uses Lemma 3. Next plug the above inequality back to Eq. equation 23, we have:

$$\sum_{k=1}^{K}\mathbb{E}\|\nu_i^{(k)} - \bar{\nu}_i\|^2 \leq (1-\alpha_i)^2(1+\beta)\sum_{k=1}^{K}\mathbb{E}\|\nu_{i-1}^{(k)} - \bar{\nu}_{i-1}\|^2 + 4KL^2\eta_{i-1}^2\left(1+\frac{1}{\beta}\right)\mathbb{E}\|\tilde{d}_{i-1}\|^2$$
$$+ \frac{4K\sigma^2}{b_1}\left(1+\frac{1}{\beta}\right)\alpha_i^2 + 8K\zeta^2\left(1+\frac{1}{\beta}\right)\alpha_i^2$$
$$+ \frac{16\lambda^2L^2(1+2a_i^2)(I-1)}{\rho^2}\left(1+\frac{1}{\beta}\right)\sum_{\ell=1}^{i-1}\eta_\ell^2\sum_{k=1}^{K}\mathbb{E}\|\nu_\ell^{(k)} - \bar{\nu}_\ell\|^2$$
$$\leq (1+\frac{1}{I})\sum_{k=1}^{K}\mathbb{E}\|\nu_{i-1}^{(k)} - \bar{\nu}_{i-1}\|^2 + 8KIL^2\eta_{i-1}^2\mathbb{E}\|\tilde{d}_{i-1}\|^2 + \frac{8KI\sigma^2c^2\eta_{i-1}^4}{b_1}$$
$$+ 16KI\zeta^2c^2\eta_{i-1}^4 + \frac{96\lambda^2I^2L^2}{\rho^2}\sum_{\ell=1}^{i-1}\eta_\ell^2\sum_{k=1}^{K}\mathbb{E}\|\nu_\ell^{(k)} - \bar{\nu}_\ell\|^2,$$

In the last inequality, we choose $\beta = 1/I$, then we have $(1+1/\beta) \leq (1+I) \leq 2I$, we also use the fact that $(1-\alpha_i)^2 < 1$ and $a_i = c\eta_{i-1}^2 < 1$. This completes the proof. $\qquad\square$

**Lemma 10.** *For $\eta_i \leq \frac{\rho}{48LI^2}$, then we have*

$$\frac{I^2}{\rho K} \sum_{i=1}^{I} \eta_i \sum_{k=1}^{K} \mathbb{E}\|\nu_i^{(k)} - \bar{\nu}_i\|^2 \leq \frac{\rho}{84} \sum_{i=0}^{I-1} \eta_i \mathbb{E}\|\tilde{d}_i\|^2 + \left(\frac{\rho\sigma^2 c^2}{84b_1 L^2} + \frac{\rho\zeta^2 c^2}{42L^2}\right) \sum_{i=0}^{I-1} \eta_i^3$$

*Proof.* By Lemma 9 (we omit the global epoch number for convenience) we have:

$$\sum_{k=1}^{K} \mathbb{E}\|\nu_i^{(k)} - \bar{\nu}_i\|^2 \leq (1 + \frac{1}{I}) \sum_{k=1}^{K} \mathbb{E}\|\nu_{i-1}^{(k)} - \bar{\nu}_{i-1}\|^2 + 8KIL^2\eta_{i-1}^2 \mathbb{E}\|\tilde{d}_{i-1}\|^2 + \frac{8KI\sigma^2 c^2 \eta_{i-1}^4}{b_1}$$

$$+ 16KI\zeta^2 c^2 \eta_{i-1}^4 + \frac{96\lambda^2 I^2 L^2}{\rho^2} \sum_{\ell=1}^{i-1} \eta_\ell^2 \sum_{k=1}^{K} \mathbb{E}\|\nu_\ell^{(k)} - \bar{\nu}_\ell\|^2$$

$$\leq (1 + \frac{1}{I}) \sum_{k=1}^{K} \mathbb{E}\|\nu_{i-1}^{(k)} - \bar{\nu}_{i-1}\|^2 + \frac{KL\rho\eta_{i-1}}{6\lambda I} \mathbb{E}\|\tilde{d}_{i-1}\|^2 + \frac{K\rho\sigma^2 c^2 \eta_{i-1}^3}{6\lambda ILb_1}$$

$$+ \frac{K\rho\zeta^2 c^2 \eta_{i-1}^3}{3\lambda IL} + \frac{96\lambda^2 I^2 L^2}{\rho^2} \sum_{\ell=1}^{i-1} \eta_\ell^2 \sum_{k=1}^{K} \mathbb{E}\|\nu_\ell^{(k)} - \bar{\nu}_\ell\|^2, \tag{24}$$

where in the second inequality, we use the condition that $\eta_i \leq \frac{\rho}{48\lambda LI^2}$. Applying equation 24 recursively from 1 to $i$. We have:

$$\sum_{k=1}^{K} \mathbb{E}\|\nu_i^{(k)} - \bar{\nu}_i\|^2 \leq \frac{KL\rho}{6\lambda I} \sum_{\ell=0}^{i-1} \left(1 + \frac{1}{I}\right)^{i-1-\ell} \eta_\ell \mathbb{E}\|\tilde{d}_\ell\|^2 + \frac{K\rho\sigma^2 c^2}{6\lambda ILb_1} \sum_{\ell=0}^{i-1} \left(1 + \frac{1}{I}\right)^{i-1-\ell} \eta_\ell^3$$

$$+ \frac{K\rho\zeta^2 c^2}{3\lambda IL} \sum_{\ell=0}^{i-1} \left(1 + \frac{1}{I}\right)^{i-1-\ell} \eta_\ell^3$$

$$+ \frac{96\lambda^2 L^2 I^2}{\rho^2} \sum_{\ell=0}^{i-1} \left(1 + \frac{1}{I}\right)^{i-1-\ell} \sum_{\bar{\ell}=0}^{\ell} \eta_{\bar{\ell}}^2 \sum_{k=1}^{K} \mathbb{E}\|\nu_{\bar{\ell}}^{(k)} - \bar{\nu}_{\bar{\ell}}\|^2$$

$$\overset{(a)}{\leq} \frac{KL\rho}{6\lambda I} \left(1 + \frac{1}{I}\right)^{I} \sum_{\ell=0}^{i-1} \eta_\ell \mathbb{E}\|\tilde{d}_\ell\|^2 + \frac{K\rho\sigma^2 c^2}{6\lambda ILb_1} \left(1 + \frac{1}{I}\right)^{I} \sum_{\ell=0}^{i-1} \eta_\ell^3$$

$$+ \frac{K\rho\zeta^2 c^2}{3\lambda IL} \left(1 + \frac{1}{I}\right)^{I} \sum_{\ell=0}^{i-1} \eta_\ell^3 + \frac{96\lambda^2 L^2 I^3}{\rho^2} \left(1 + \frac{1}{I}\right)^{I} \sum_{\bar{\ell}=0}^{i-1} \eta_{\bar{\ell}}^2 \sum_{k=1}^{K} \mathbb{E}\|\nu_{\bar{\ell}}^{(k)} - \bar{\nu}_{\bar{\ell}}\|^2$$

$$\overset{(b)}{\leq} \frac{KL\rho}{2\lambda I} \sum_{\ell=0}^{i-1} \eta_\ell \mathbb{E}\|\tilde{d}_\ell\|^2 + \frac{K\rho\sigma^2 c^2}{2\lambda ILb_1} \sum_{\ell=0}^{i-1} \eta_\ell^3 + \frac{K\rho\zeta^2 c^2}{\lambda IL} \sum_{\ell=0}^{i-1} \eta_\ell^3$$

$$+ \frac{288\lambda^2 L^2 I^3}{\rho^2} \sum_{\ell=0}^{i-1} \eta_\ell^2 \sum_{k=1}^{K} \mathbb{E}\|\nu_\ell^{(k)} - \bar{\nu}_\ell\|^2, \tag{25}$$

where inequality $(a)$ is by the fact that $1 + 1/I > 1$ and $i - 1 - \ell \leq I$ for $i \in [I]$ and $\ell \in [i]$ and inequality $(b)$ is because that $(1 + 1/I)^I \leq \mathrm{e} < 3$.

Next, multiplying both sides of equation 25 by $\eta_i$ and summing over $i = 1$ to $I$:

$$\sum_{i=1}^{I} \eta_i \sum_{k=1}^{K} \mathbb{E}\|\nu_i^{(k)} - \bar{\nu}_i\|^2 \leq \frac{KL\rho}{2\lambda I} \sum_{i=1}^{I} \eta_i \sum_{\ell=0}^{i-1} \eta_\ell \mathbb{E}\|\tilde{d}_\ell\|^2 + \frac{K\rho\sigma^2 c^2}{2\lambda I L b_1} \sum_{i=1}^{I} \eta_i \sum_{\ell=0}^{i-1} \eta_\ell^3$$

$$+ \frac{K\rho\zeta^2 c^2}{\lambda IL} \sum_{i=1}^{I} \eta_i \sum_{\ell=0}^{i-1} \eta_\ell^3 + \frac{288\lambda^2 L^2 I^3}{\rho^2} \sum_{i=1}^{I} \eta_i \sum_{\ell=0}^{i-1} \eta_\ell^2 \sum_{k=1}^{K} \mathbb{E}\|\nu_\ell^{(k)} - \bar{\nu}_\ell\|^2$$

$$\overset{(a)}{\leq} \frac{KL\rho}{2\lambda I} \left( \sum_{i=1}^{I} \eta_i \right) \sum_{\ell=0}^{I-1} \eta_\ell \mathbb{E}\|\tilde{d}_\ell\|^2 + \left( \frac{K\rho\sigma^2 c^2}{2\lambda I L b_1} + \frac{K\rho\zeta^2 c^2}{\lambda IL} \right) \left( \sum_{i=1}^{I} \eta_i \right) \sum_{\ell=0}^{I-1} \eta_\ell^3$$

$$+ \frac{288\lambda^2 L^2 I^3}{\rho^2} \left( \sum_{i=1}^{I} \eta_i \right) \sum_{\ell=0}^{I-1} \eta_\ell^2 \sum_{k=1}^{K} \mathbb{E}\|\nu_\ell^{(k)} - \bar{\nu}_\ell\|^2$$

$$\overset{(b)}{\leq} \frac{K\rho^2}{96\lambda^2 I^2} \sum_{i=0}^{I-1} \eta_i \mathbb{E}\|\tilde{d}_i\|^2 + \left( \frac{K\rho^2\sigma^2 c^2}{96\lambda^2 I^2 L^2 b_1} + \frac{K\rho^2\zeta^2 c^2}{48\lambda^2 I^2 L^2} \right) \sum_{i=0}^{I-1} \eta_i^3 + \frac{1}{8} \sum_{\ell=1}^{I-1} \eta_\ell \sum_{k=1}^{K} \mathbb{E}\|\nu_\ell^{(k)} - \bar{\nu}_\ell\|^2$$

where inequality $(a)$ uses the fact that $i \leq I$ and $(b)$ uses that we choose $\eta_i \leq \rho/(48\lambda LI^2)$. Rearranging the terms we have:

$$\frac{7}{8} \sum_{i=1}^{I} \eta_i \sum_{k=1}^{K} \mathbb{E}\|\nu_i^{(k)} - \bar{\nu}_i\|^2 \leq \frac{K\rho^2}{96\lambda^2 I^2} \sum_{i=0}^{I-1} \eta_i \mathbb{E}\|\tilde{d}_i\|^2 + \left( \frac{K\rho^2\sigma^2 c^2}{96\lambda^2 I^2 L^2 b_1} + \frac{K\rho^2\zeta^2 c^2}{48\lambda^2 I^2 L^2} \right) \sum_{i=0}^{I-1} \eta_i^3$$

Multiplying $8\lambda I^2/(7K\rho)$ on both sides, we have:

$$\frac{\lambda I^2}{K\rho} \sum_{i=1}^{I} \eta_i \sum_{k=1}^{K} \mathbb{E}\|\nu_i^{(k)} - \bar{\nu}_i\|^2 \leq \frac{\rho}{84\lambda} \sum_{i=0}^{I-1} \eta_i \mathbb{E}\|\tilde{d}_i\|^2 + \left( \frac{\rho\sigma^2 c^2}{84\lambda L^2 b_1} + \frac{\rho\zeta^2 c^2}{42\lambda L^2} \right) \sum_{i=0}^{I-1} \eta_i^3$$

This completes the proof. $\qquad\square$

### 9.8 PROOF OF THE MAIN CONVERGENCE THEOREM

In this subsection, we prove Theorem 5.1 and Corollary 5.7. To prove Theorem 5.1, we firstly show the following theorem hold:

**Theorem 9.1.** *Choosing the parameters as* $\kappa = \frac{\rho K^{2/3}}{\lambda L}$, $c = \frac{96\lambda^2 L^2}{K\rho^2} + \frac{\rho}{72\kappa^3 \lambda LI^2}$, $w_t = \max\left\{ 48^3 I^6 K^2 - t - I, 14^3 K^{0.5} \right\}$, $\lambda > 0$, *and choose* $\eta_t = \frac{\kappa}{(\omega_t + t + I)^{1/3}}$, *then we have:*

$$\frac{1}{T} \sum_{t=0}^{T-1} \left( \mathbb{E}\|\tilde{d}_t\|^2 + \frac{\lambda^2}{\rho^2} \mathbb{E}\|\bar{e}_t\|^2 \right)$$

$$\leq \left[ \frac{96\lambda^2 LI^2}{\rho^2 T} + \frac{2\lambda^2 L}{\rho^2 K^{2/3} T^{2/3}} \right] (f(x_0) - f^*) + \left[ \frac{72\lambda^2 I^4}{b\rho^2 T} + \frac{3\lambda^2 I^2}{2b\rho^2 K^{2/3} T^{2/3}} \right] \sigma^2$$

$$+ \frac{192^2 \lambda^2}{\rho^2} \times \left( \frac{48I^2}{T} + \frac{1}{K^{2/3} T^{2/3}} \right) \times \left( \frac{\sigma^2}{4b_1} + \frac{2\zeta^2}{21} \right) \log(T+1).$$

*Proof.* By definition, we have $\eta_t \leq \eta_0 < \kappa/w_0^{1/3} = \rho K^{2/3}/48\lambda LI^2 K^{2/3} = \rho/48\lambda LI^2$, then $c = \lambda^2 L^2 \left( \frac{96}{K\rho^2} + \frac{1}{72K^2\rho^2 I^2} \right) \leq \frac{192\lambda^2 L^2}{K\rho^2}$ and:

$$c\eta_t^2 \leq c\eta_0^2 < \frac{192\lambda^2 L^2}{K\rho^2} * \frac{\kappa^2}{w_0^{2/3}} = \frac{192L^2}{K\rho^2} * \frac{\rho^2 K^{4/3}}{L^2 w_0^{2/3}} = \frac{192K^{1/3}}{w_0^{2/3}} \leq \frac{192K^{1/3}}{196K^{1/3}} < 1,$$

So we have $\alpha_t < 1$, then the conditions of Lemma 8-Lemma 10 are satisfied.

Firstly, substitute the gradient consensus error in Lemma 10 to Lemma 8, we can write the descent of potential function as:

$$
\begin{aligned}
\mathbb{E}[\Phi_{\tau+1} - \Phi_\tau] \leq & -\sum_{i=0}^{I-1}\left(\frac{5\rho\eta_{\tau+1,i}}{8\lambda} - \frac{\eta_{\tau+1,i}^2 L}{2}\right)\mathbb{E}\|\tilde{d}_{\tau+1,i}\|^2 - \frac{\lambda}{2\rho}\sum_{i=0}^{I-1}\eta_{\tau+1,i}\mathbb{E}\|\bar{e}_{\tau+1,i}\|^2 \\
& + \frac{\sigma^2 c^2\rho}{16\lambda L^2 b_1}\sum_{i=0}^{I-1}\eta_{\tau+1,i}^3 + \frac{\rho}{42\lambda}\sum_{i=0}^{I-1}\eta_{\tau+1,i}\mathbb{E}\|\tilde{d}_{\tau+1,i}\|^2 + \left(\frac{\rho\sigma^2 c^2}{42\lambda L^2 b_1} + \frac{\rho\zeta^2 c^2}{21\lambda L^2}\right)\sum_{i=0}^{I-1}\eta_{\tau+1,i}^3 \\
\leq & -\sum_{i=0}^{I-1}\left(\frac{3\rho\eta_{\tau+1,i}}{5\lambda} - \frac{\eta_{\tau+1,i}^2 L}{2}\right)\mathbb{E}\|\tilde{d}_{\tau+1,i}\|^2 - \frac{\lambda}{2\rho}\sum_{i=0}^{I-1}\eta_{\tau+1,i}\mathbb{E}\|\bar{e}_{\tau+1,i}\|^2 \\
& + \left(\frac{\rho\sigma^2 c^2}{8\lambda L^2 b_1} + \frac{\rho\zeta^2 c^2}{21\lambda L^2}\right)\sum_{i=0}^{I-1}\eta_{\tau+1,i}^3 \\
\overset{(a)}{\leq} & -\sum_{i=0}^{I-1}\frac{\rho\eta_i}{2\lambda}\mathbb{E}\|\tilde{d}_{\tau+1,i}\|^2 - \frac{\lambda}{2\rho}\sum_{i=0}^{I-1}\eta_{\tau+1,i}\mathbb{E}\|\bar{e}_{\tau+1,i}\|^2 + \left(\frac{\rho\sigma^2 c^2}{8\lambda L^2 b_1} + \frac{\rho\zeta^2 c^2}{21\lambda L^2}\right)\sum_{i=0}^{I-1}\eta_{\tau+1,i}^3,
\end{aligned}
$$

where $(a)$ follows from the fact that $\eta_i \leq \frac{\rho}{48\lambda L I^2} \leq \frac{\rho}{48\lambda L}$.

Suppose we denote $T = EI$, and $t = \tau I + i$ for $t \geq 0$ and $\tau \geq 0$. Then we have $\eta_t = \eta_{\tau+1,i}$, $\tilde{d}_t = \tilde{d}_{\tau+1,i}$, $\bar{e}_t = \bar{e}_{\tau+1,i}$. In particular, we denote $\eta_{-1} = \eta_0$ for convenience.

Then we sum the above inequality for $\tau$ from 0 to $E - 1$, and get:

$$
\mathbb{E}[\Phi_E - \Phi_0] \leq -\sum_{t=0}^{T-1}\left(\frac{\rho\eta_t}{2\lambda}\right)\mathbb{E}\|\tilde{d}_t\|^2 - \sum_{t=0}^{T-1}\frac{\lambda\eta_t}{2\rho}\mathbb{E}\|\bar{e}_t\|^2 + \left(\frac{\rho\sigma^2 c^2}{8\lambda L^2 b_1} + \frac{\rho\zeta^2 c^2}{21\lambda L^2}\right)\sum_{t=0}^{T}\eta_t^3,
$$

Rearranging terms, we get:

$$
\begin{aligned}
\sum_{t=1}^{T}\left(\frac{\rho\eta_t}{2\lambda}\mathbb{E}\|\tilde{d}_t\|^2 + \frac{\lambda\eta_t}{2\rho}\mathbb{E}\|\bar{e}_t\|^2\right) \leq & \ \mathbb{E}[\Phi_0 - \Phi_E] + \left(\frac{\rho\sigma^2 c^2}{8\lambda L^2 b_1} + \frac{\rho\zeta^2 c^2}{21\lambda L^2}\right)\sum_{t=0}^{T-1}\eta_t^3 \\
\overset{(a)}{\leq} & \ f(x_0) - f^* + \frac{\rho K}{64\lambda L^2}\frac{\mathbb{E}\|e_0\|^2}{\eta_0} + \left(\frac{\rho\sigma^2 c^2}{8\lambda L^2 b_1} + \frac{\rho\zeta^2 c^2}{21\lambda L^2}\right)\sum_{t=0}^{T-1}\eta_t^3 \\
\overset{(b)}{\leq} & \ f(x_0) - f^* + \frac{\sigma^2\rho}{64\lambda bL^2\eta_0} + \left(\frac{\rho\sigma^2 c^2}{8\lambda L^2 b_1} + \frac{\rho\zeta^2 c^2}{21\lambda L^2}\right)\sum_{t=0}^{T-1}\eta_t^3,
\end{aligned}
$$
(26)

where $(a)$ follows from the fact that $f^* \leq \Phi_E$ and $(b)$ results from application of Lemma 5 and $b$ is the minibatch size at the first iteration.

Next for the last term of the equation 26 above, we have:

$$
\sum_{t=0}^{T-1}\eta_t^3 = \sum_{t=0}^{T-1}\frac{\kappa^3}{w_t + t} \overset{(a)}{\leq} \sum_{t=0}^{T-1}\frac{\kappa^3}{1 + t} = \kappa^3\sum_{t=0}^{T-1}\frac{1}{1+t} \overset{(b)}{\leq} \kappa^3\ln(T+1).
\tag{27}
$$

where inequality $(a)$ above follows from the fact that we have $w_t > 1$ and inequality $(b)$ follows from the application of Proposition 3.

Substituting equation 27 in equation 26, multiplying both sides by $2\lambda/(\rho\eta_T T)$ and using the fact that $\eta_t$ is non-increasing in $t$ we have

$$
\frac{1}{T}\sum_{t=0}^{T-1}\left(\mathbb{E}\|\tilde{d}_t\|^2 + \frac{\lambda^2}{\rho^2}\mathbb{E}\|\bar{e}_t\|^2\right) \leq \frac{2\lambda(f(x_0) - f^*)}{\rho\eta_T T} + \frac{1}{\eta_T T}\frac{\sigma^2}{32bL^2\eta_0} + \frac{\kappa^3}{\eta_T T}\left(\frac{\sigma^2 c^2}{4b_1 L^2} + \frac{2\zeta^2 c^2}{21L^2}\right)\ln(T+1).
\tag{28}
$$

Now considering each term of equation 28 above separately. For the first term:

$$
\frac{1}{\eta_T T} = \frac{(w_T + T)^{1/3}}{\kappa T} \overset{(a)}{\leq} \frac{w_T^{1/3}}{\kappa T} + \frac{1}{\kappa T^{2/3}} = \frac{48\lambda L I^2}{\rho T} + \frac{\lambda L}{\rho K^{2/3}T^{2/3}}.
\tag{29}
$$

where inequality $(a)$ follows from identity $(x+y)^{1/3} \leq x^{1/3} + y^{1/3}$ and inequality $(b)$ follows from the definition of $\kappa$ and $w_T$

$$w_T = \max \left\{ (I+1), 48^3 I^6 K^2 - T, 2*320^{1.5} K^{0.5} \right\} \leq 48^3 I^6 K^2,$$

Similarly, for the second term of equation 28, we have from the definition of $\eta_0$ and $\eta_T$

$$
\begin{aligned}
\frac{1}{\eta_T T} \frac{\sigma^2}{32bL^2 \eta_0} &\leq \left( \frac{48\lambda L I^2}{\rho T} + \frac{\lambda L}{\rho K^{2/3} T^{2/3}} \right) \times \frac{\sigma^2}{32bL^2} \times \frac{w_0^{1/3}}{\kappa} \\
&\leq \left( \frac{48\lambda L I^2}{\rho T} + \frac{\lambda L}{\rho K^{2/3} T^{2/3}} \right) \times \frac{\sigma^2}{32bL^2} \times \frac{48\lambda L I^2}{\rho} \\
&\leq \frac{72\lambda^2 I^4}{b\rho^2 T} \sigma^2 + \frac{3\lambda^2 I^2}{2b\rho^2 K^{2/3} T^{2/3}} \sigma^2.
\end{aligned}
\tag{30}
$$

Finally, for the last term in equation 28 above, we have from the definition of the stepsize, $\eta_t$,

$$
\begin{aligned}
&\frac{\kappa^3 c^2}{\eta_T T L^2} \left( \frac{\sigma^2}{4b_1} + \frac{2\zeta^2}{21} \right) \ln(T+1) \\
&\leq \left( \frac{48\lambda L I^2}{\rho T} + \frac{\lambda L}{\rho K^{2/3} T^{2/3}} \right) \times \frac{192^2 \lambda}{L\rho} \times \left( \frac{\sigma^2}{4b_1} + \frac{2\zeta^2}{21} \right) \log(T+1) \\
&\leq \frac{192^2 \lambda^2}{\rho^2} \times \left( \frac{48 I^2}{T} + \frac{1}{K^{2/3} T^{2/3}} \right) \times \left( \frac{\sigma^2}{4b_1} + \frac{2\zeta^2}{21} \right) \log(T+1).
\end{aligned}
\tag{31}
$$

Finally, substituting the bounds obtained in equation 29, equation 30 and equation 31 into equation 28, we get

$$
\begin{aligned}
&\frac{1}{T} \sum_{t=0}^{T-1} \left( \mathbb{E}\|\tilde{d}_t\|^2 + \frac{\lambda^2}{\rho^2} \mathbb{E}\|\bar{e}_t\|^2 \right) \\
&\leq \left[ \frac{96\lambda^2 L I^2}{\rho^2 T} + \frac{2\lambda^2 L}{\rho^2 K^{2/3} T^{2/3}} \right] (f(x_0) - f^*) + \left[ \frac{72\lambda^2 I^4}{b\rho^2 T} + \frac{3\lambda^2 I^2}{2b\rho^2 K^{2/3} T^{2/3}} \right] \sigma^2 \\
&\quad + \frac{192^2 \lambda^2}{\rho^2} \times \left( \frac{48 I^2}{T} + \frac{1}{K^{2/3} T^{2/3}} \right) \times \left( \frac{\sigma^2}{4b_1} + \frac{2\zeta^2}{21} \right) \log(T+1).
\end{aligned}
$$

This completes the proof of the theorem. $\qquad\square$

Now we are ready to show Theorem 5.1. Firstly notice that:

$$\frac{\lambda^2 \mathcal{G}_t}{\rho^2} = \frac{1}{\eta_t^2} \|\tilde{x}_t - \tilde{x}_{t+1}\|^2 + \frac{\lambda^2}{\rho^2} \|\bar{\nu}_t - \nabla f(\tilde{x}_t)\|^2 = \|\tilde{d}_t\|^2 + \frac{\lambda^2}{\rho^2} \|\bar{e}_t\|^2$$

Combine with Theorem 9.1, we have:

$$
\begin{aligned}
\frac{1}{T} \sum_{t=0}^{T-1} \mathbb{E}[\mathcal{G}_t] &\leq \left[ \frac{96 L I^2}{T} + \frac{2L}{K^{2/3} T^{2/3}} \right] (f(x_0) - f^*) + \left[ \frac{72 I^4}{bT} + \frac{3 I^2}{2b K^{2/3} T^{2/3}} \right] \sigma^2 \\
&\quad + 192^2 \times \left( \frac{48 I^2}{T} + \frac{1}{K^{2/3} T^{2/3}} \right) \times \left( \frac{\sigma^2}{4b_1} + \frac{2\zeta^2}{21} \right) \log(T+1).
\end{aligned}
$$

*Remark* 7. For the measure $\mathcal{G}_t$, we discuss its intuition under both the unconstrained and constrained case. First, for unconstrained case, *i.e.* when $\mathcal{X} = R^d$, we have:

$$
\begin{aligned}
\|\nabla f(\tilde{x}_{\tau,i})\| / \|H_\tau\| &= \|H_\tau \times H_\tau^{-1} \nabla f(\tilde{x}_{\tau,i})\| / \|H_\tau\| \leq \|H_\tau^{-1} \nabla f(\tilde{x}_{\tau,i})\| \\
&= \|H_\tau^{-1} \nabla f(\tilde{x}_{\tau,i}) - H_\tau^{-1} \bar{\nu}_{\tau,i} + H_\tau^{-1} \bar{\nu}_{\tau,i}\| \leq \|H_\tau^{-1} \nabla f(\tilde{x}_{\tau,i}) - H_\tau^{-1} \bar{\nu}_{\tau,i}\| + \|H_\tau^{-1} \bar{\nu}_{\tau,i}\| \\
&\leq \frac{1}{\rho} \|\bar{\nu}_{\tau,i} - \nabla f(\tilde{x}_{\tau,i})\| + \frac{1}{\lambda \eta_{\tau,i}} \|\tilde{x}_{\tau,i} - \tilde{x}_{\tau,i+1}\| \leq \sqrt{2} \sqrt{\mathcal{G}_{\tau,i}} / \rho
\end{aligned}
$$

In the last inequality, we use Jensen inequality, and in the second last inequality, we use Assumption 4 and the fact that $\tilde{x}_{\tau,i+1} = x_{\tau,0} + \lambda H_\tau^{-1} \bar{z}_{\tau,i+1}$ and $\tilde{x}_{\tau,i} = x_{\tau,0} + \lambda H_\tau^{-1} \bar{z}_{\tau,i}$ and $\eta_{\tau,i} \bar{\nu}_{\tau,i} = \bar{z}_{\tau,i+1} - \bar{z}_{\tau,i}$ in the unconstrained case. In other words, we have $||\nabla f(\tilde{x}_t)||^2 \leq \frac{2||H_\tau||^2}{\rho^2} \mathcal{G}_\tau$. Note the coefficient of the right-side is an upper bound of the square condition number of $H_\tau$. It is common assumption in the analysis of adaptive gradient methods that $H_t$ has a finite condition number Huang et al. (2021). In sum, the convergence of our measure $\mathcal{G}_t$ means the convergence to a first order stationary point in the unconstrained case.

Next, for the constrained case, our measure upper bounds the gradient mapping $\frac{1}{\eta_{\tau+1,i}} ||x_\tau - x_{\tau+1,i}^*||$, $x_t^*$ is defined as follows:

$$x_{\tau+1,i}^* = \underset{x \in \mathcal{X}}{\arg\min}\{-\langle x, z_{\tau+1,i}^*\rangle + \frac{1}{2\lambda}(x - x_\tau)^T H_\tau (x - x_\tau)\}$$

where $z_{\tau+1,i}^* = \sum_{\ell=0}^i -\eta_\ell \nabla f(\tilde{x}_{\tau+1,i})$ is the accumulation of true gradient. Next follow Lemma 1, we have:

$$\|x_{\tau+1,i}^* - \tilde{x}_{\tau+1,i}\| \leq \frac{\lambda}{\rho}\|z_{\tau+1,i}^* - \bar{z}_{\tau+1,i}\|$$

$$= \frac{\lambda}{\rho}\|\sum_{l=0}^{i-1} -\eta_{\tau+1,\ell}(\nabla f(\tilde{x}_{\tau+1,\ell}) - \bar{\nu}_{\tau+1,\ell}))\| \overset{(a)}{\leq} \sum_{l=0}^{i-1} \frac{\lambda\eta_{\tau+1,\ell}}{\rho}\|\nabla f(\tilde{x}_{\tau+1,\ell}) - \bar{\nu}_{\tau+1,\ell}\|$$

where inequality $(a)$ is due to the triangle inequality. Next we have:

$$\|x_\tau - x_{\tau+1,i}^*\| = \|x_\tau - \tilde{x}_{\tau+1,i} + \tilde{x}_{\tau+1,i} - x_{\tau+1,i}^*\| \leq \|x_\tau - \tilde{x}_{\tau+1,i}\| + \|\tilde{x}_{\tau+1,i} - x_{\tau+1,i}^*\|$$

$$\leq \|\sum_{l=0}^{i-1} \tilde{d}_{\tau+1,i}\| + \|\tilde{x}_{\tau+1,i} - x_{\tau+1,i}^*\| \leq \sum_{l=0}^{i-1}\left(\|\tilde{d}_{\tau+1,\ell}\| + \frac{\lambda\eta_{\tau+1,\ell}}{\rho}\|\nabla f(\tilde{x}_{\tau+1,\ell}) - \bar{\nu}_{\tau+1,\ell}\|\right)$$

By Jensen inequality and the definition of the measure equation 9, we have

$$\|\tilde{d}_t\| + \frac{\lambda\eta_t}{\rho}\|\nabla f(\tilde{x}_t) - \bar{\nu}_t\| \leq \frac{\sqrt{2}\lambda\eta_t}{\rho}\sqrt{\mathcal{G}_t},$$

So we have

$$\frac{1}{\eta_{\tau+1,i}}\|x_\tau - x_{\tau+1,i}^*\| \leq \frac{\sqrt{2}\lambda}{\rho}\sum_{l=0}^{i-1}\frac{\eta_{\tau+1,l}}{\eta_{\tau+1,i}}\sqrt{\mathcal{G}_{\tau+1,l}} \leq \frac{2\sqrt{2}\lambda}{\rho}\sum_{l=0}^{i-1}\sqrt{\mathcal{G}_{\tau+1,\ell}},$$

the last inequality is because of Eq. equation 17. In all, when the measure $\mathcal{G}_{\tau+1,\ell} \to 0$, the gradient mapping $\frac{1}{\eta_{\tau+1,i}}\|x_\tau - x_{\tau+1,i}^*\|$ converges to 0.

**Corollary 2.** *With the hyper-parameters chosen as in Theorem 9.1. Suppose we set $I = O((T/K^2)^{1/6})$ and use sample minibatch of size $O(I^2)$ in the first step, Then we have:*

$$\mathbb{E}[\mathcal{G}_t] = O\left(\frac{f(x_0) - f^*}{K^{2/3}T^{2/3}}\right) + \tilde{O}\left(\frac{\sigma^2}{K^{2/3}T^{2/3}}\right) + \tilde{O}\left(\frac{\zeta^2}{K^{2/3}T^{2/3}}\right).$$

*and to reach an $\epsilon$-stationary point, we need to make $\tilde{O}(\epsilon^{-1.5}/K)$ number of steps and need $\tilde{O}(\epsilon^{-1})$ number of communication rounds.*

*Proof.* It is straightforward to verify the expression for $\mathbb{E}[\mathcal{G}_t]$ in the corollary by applying Theorem 9.1 and choosing $I$ and $b$ as corresponding values. As for the gradient and communication complexity of the algorithm. We have the following results: The number of total steps $T$ needed to achieve an $\epsilon$-stationary point, *i.e.* $\tilde{O}(\frac{1}{K^{2/3}T^{2/3}}) = \epsilon$ are $\mathcal{O}(\frac{1}{K\epsilon^{3/2}})$, *i.e.* the gradient complexity. Total rounds of communication steps to achieve an $\epsilon$-stationary point is $E = T/I$, as we have $I = O((T/K^2)^{1/6})$, then $T/I = \tilde{O}(K^{1/3}T^{5/6})$. Assume we have large number of clients compared, more specifically, assume $K \geq \sqrt{T}$. Then we have $T/I = \tilde{O}(K^{1/3}T^{5/6}) = \tilde{O}(K^{2/3}T^{2/3})$, in other words, we have $E = \tilde{O}(\epsilon^{-1})$. This completes the proof of the corollary. $\qquad\square$

