# OpenReview forum: "FedDA: Faster Framework of Local Adaptive Gradient Methods via Restarted Dual Averaging"
_ICLR.cc/2023/Conference — Submitted to ICLR 2023_

### Official Review · Reviewer_M1w3 · 2022-10-24

**Confidence:** 3
**Correctness:** 4
**Technical Novelty And Significance:** 3
**Empirical Novelty And Significance:** 3
**Recommendation:** 6

**Clarity, Quality, Novelty And Reproducibility:**

Clarity:

From my perspective, it is a little bit hard to understand the details of FedDA, especially because there is a difference between the global update (Line 9) and the equation (2): the $\eta$ and $v$ disappear and are included in the dual difference $z$. I suggest the author show an example that sets the number of local steps to $I=1$ and reduces FedDA to an adaptive algorithm w/o local step.

All other parts are clear and well-written.

Quality: Good

Novelty: Neutral to Good

Reproducibility: Good

**Strength And Weaknesses:**

Strength:

1. Introduce a new federated learning algorithm that combines adaptive gradient and local steps
2. The proposed algorithm supports the clients to use adaptive gradient because it averages the gradient information in the dual space
3. The theory looks good, since it has the $1/T$ communication round.
4. Numerical experiments corroborate the theoretical findings.

Weakness:

1. For theory, it needs the bounded dissimilarity assumption, which is a huge drawback for federated learning scenarios but it is acceptable for the analysis of adaptive gradient methods.
2. In the numerical experiment, current discussion cannot conclude that FEMNIST experiment is heterogeneous. It is not clear how you split the data to client: if you have 500 clients and each client has the data of a single person, then maybe you can say it is heterogeneous. If you only has 10 clients and each client random get 50 people's data, then maybe it is not heterogeneous enough.
3. To me, more ablation studies/numerical experiments on the effectiveness of the local steps are needed, e.g., compare the difference between $I=1$ and $I=5$ and $I=50$ etc.

**Summary Of The Paper:**

This paper proposes FedDA, which is a federated learning algorithm that uses adaptive gradient and local steps. Different from the previous work, this paper supports the clients to run the adaptive gradient locally and update the global model using dual averaging - average the gradient information in the dual space. Theoretical analysis shows the effectiveness of the method, and numerical experiments are presented to corroborate the theoretical findings.

**Summary Of The Review:**

Although there are some weaknesses, I still think that the contribution is enough and I would like to weakly accept this paper.

After rebuttal:
I still vote for weak accept of this paper.

---

> ### Author Response · Authors · 2022-11-15
> **Response to Reviewer M1w3**
>
> Firstly, we want to thank the reviewer for identifying our contribution and providing insightful comments. Our responses to your questions are as follows:
>
> **Q1**: the usage of bounded dissimilarity assumption?
>
> **A1**: the bounded dissimilarity assumption is commonly used in the analysis of (non-adaptive) heterogeneous FL algorithms, e.g. [1,2]. In this assumption, we assume the gradient difference among clients at any given point is bounded, we believe it is a reasonable measure of the heterogeneity among clients. However, we also agree with the reviewer that better heterogeneity measure is possible (even without heterogeneity assumption), but it is beyond the scope of this manuscript, and we leave it as a future work.
>
> **Q2**: how do you split FEMNIST data?
>
> **A2**: As we mentioned in the beginning of section 8.1.2 of the Appendix, we randomly select 50 users from 500 users at each epoch. Each client only has the data of one user when performing local steps. To further validate the performance of our framework over heterogeneous data, we manually create a heterogeneous dataset based on the CIFAR10. This set of experiments are described in Section 8.3 of the Appendix (revised version). Simply speaking, we create heterogeneity by making each client have a different dominant class. As shown by the experimental results, our FedDA still outperforms other baseline methods.
>
> **Q3**: more ablation studies on the effectiveness of local steps are needed?
>
> **A3**: In Figure 10 of Appendix (revised version), we show results for different values of I for four variants of FedDA. As shown by the experimental results, our FedDA framework can benefit from local steps, but we also see that the benefit of local steps decreases when $I$ gets larger.
>
> **Q4**: show special case of $I=1$ (no local steps)?
>
> **A4**: The form of FedDA when $I=1$ is shown in Algorithm 3 (revised version). When $I=1$, FedDA simply distributedly evaluates eq.2: The clients only need to update the gradient estimate $\nu$ locally.
>
> **[1]** Das, Rudrajit, et al. "Faster non-convex federated learning via global and local momentum." Uncertainty in Artificial Intelligence. PMLR, 2022.
>
> **[2]** Khanduri, Prashant, et al. "Stem: A stochastic two-sided momentum algorithm achieving near-optimal sample and communication complexities for federated learning." Advances in Neural Information Processing Systems 34 (2021): 6050-6061.

---

### Official Review · Reviewer_pitV · 2022-10-25

**Confidence:** 3
**Correctness:** 4
**Technical Novelty And Significance:** 3
**Empirical Novelty And Significance:** 3
**Recommendation:** 8

**Clarity, Quality, Novelty And Reproducibility:**

Clarity and quality:

The paper is well written. There are (too) detailed explanations of the meta algorithm and explanations of the FedDA-MVR. It was a bit difficult to understand the relationship between mirror descent and adaptive method in the context of FL, but that minor. Maybe the authors could give the algo first and then explain that taking a particular matrix H recovers adaptive gradient methods.

Q: What do you mean by constraint problem? The fact that the optimization problem is over a compact set?


Novelty:

The paper provides SOTA complexity result for adaptive gradient methods in the context of FL. I think that this significant and somewhat new (some adaptive methods for FL were proposed see Table 1).

Reproducibility: No issue

**Strength And Weaknesses:**

Strength

- While this is a theory paper, the authors provide numerical evidence for the effectiveness of FedDA-MVR in a ML context

-The paper is clearly written and fits with ICLR in terms of topic (adaptive gradient methods for FL) and format (one algo, one clear theorem with a clear contribution and some numerical exp)

-I didn't check the proofs, but the main theorem and its Lyapunov approach seem reasonable.

Weaknesses

-Why do we need to assume the variance of the gradient bounded if we use variance reduction later on?

-It seems that the theory only cover the case of minimization over a compact set.





**Summary Of The Paper:**

This papers proposes a meta algorithm for the federated learning optimization problem with heterogeneous data. This algorithm uses adaptive learning rates. The authors view the adaptive methods from the point of view of mirror descent: the proposed algorithm ressembles a Federated mirror descent algorithm where the clients perform several steps and the central server aggregates the dual vectors. The proposed algorithm has some flexibility in the way the gradients are computed. When the gradient are variance reduced and some momentum is introduced (algo FedDA-MVR), the authors show a gradient complexity 1/eps^1.5 and communication complexity 1/eps for the resulting adaptive method to find an eps stationary point.

**Summary Of The Review:**

Good theory contribution which fits well with ICLR and the practice of Federated Learning

---

> ### Author Response · Authors · 2022-11-15
> **Response to Reviewer pitV**
>
> We first want to thank the reviewer for identifying our contribution and providing the positive comments. The following is some responses to your questions.
>
> **Q1**: Why do we need to assume the variance of the gradient bounded?
>
> **A1**: Yes, even if we use the variance reduction technique, the stochastic gradient still needs to have finite/bounded variance. The effect of variance reduction is reducing a fraction of the noise proportional to $\eta^2$ (suppose $\eta$ is the learning rate) per step, thus, leading to a better overall convergence rate. The bounded variance assumption is standard in stochastic optimization literature including the momentum-based variance reduction work. [1]
>
> **Q2**: It seems that the theory only cover the case of minimization over a compact set?
>
> **A2**: Our theory covers the case when $\mathcal{X}\subset \mathbb{R}^{d}$ is a compact and convex set or the whole space $\mathcal{X} = \mathbb{R}^{d}$. In fact, when $\mathcal{X} = \mathbb{R}^{d}$, many steps in the theoretical analysis can be simplified.
>
> **Q3**: about clarity and quality: the relationship between mirror descent and adaptive gradients?
>
> **A3**: Your understanding about the relationship between mirror descent and adaptive gradients is correct: a specific matrix H corresponds to a particular adaptive gradient method.
>
> **Q4**: What do you mean by constraint problem? The fact that the optimization problem is over a compact set?
>
> **A4**: The constraint problem in our manuscript means the optimization problem over a convex and compact set.
>
> If The above response addresses your questions, please kindly raise the confidence score. If you have any further questions, we are also happy to discuss.
>
> **[1]** Cutkosky, Ashok, and Francesco Orabona. "Momentum-based variance reduction in non-convex sgd." Advances in neural information processing systems 32 (2019).

---

> > ### Comment · Reviewer_pitV · 2022-12-05
> > **Thanks for the clarification**
> >
> > Thanks for the answers, but I will keep my confidence score.

---

### Official Review · Reviewer_u8LZ · 2022-10-26

**Confidence:** 4
**Correctness:** 3
**Technical Novelty And Significance:** 4
**Empirical Novelty And Significance:** 2
**Recommendation:** 5

**Clarity, Quality, Novelty And Reproducibility:**

The literature are throughly covered, the writing is technical which makes reader harder to digest the result.

**Strength And Weaknesses:**

Strength:
- Theoretical strong

Weakness
- Lack of explanation on theoretical analysis
- The experimental section can be improved, for example, include non iid experiments

**Summary Of The Paper:**

The paper proposes a novel framework of adaptive gradient methods for Federated Learning. By incorporating momentum-based variance reduction techniques, the proposed method is able to achieve acceleration in computation complexity $O(\epsilon^{-1.5})$ and communication complexity $O(\epsilon^{-1})$. Experiments are provided to demonstrate the acceleration in practice.

**Summary Of The Review:**

The paper proposes a novel framework of adaptive gradient methods for Federated Learning. The framework is base on a dual averaging step along with Bregman proximal gradient step. By setting the Bregman distance with a quadratic mirror map, the proposed framework enables adaptive stepsizes where AdaGrad and Adam can be retrieved. The main contribution of the paper is to combine a momentum based variance reduction step in the local update to achieve faster computational and communication complexity.

**My main concern is how exactly is the theoretical acceleration achieved? As far as I understand, the adaptiveness does not contribute to the theoretical acceleration. Moreover, I don't think the algorithm has the "momentum-based variance reduction" component, the equation 5 and 6 are simple average, which is neither extrapolation not variance reduction. Hence I am confused how the acceleration is achieved.**

First, from Theorem 5.1, the analysis only requires the adaptive matrix is larger than $\rho Id$. In other words, the acceleration does not come from the adaptive step. This is also commonly known in standard stochastic optimization literature where adaptive method has better practical performance but no theoretical improvement. Hence the theoretical acceleration is not coming from the adaptiveness.

Second, usually acceleration is achieved by extrapolation. In equation 5 and 6 where the paper claims as momentum-based variance
reduction update, it is a simple weighted average, it is neither variance reduction nor momentum-based, see for example [1] for an example algorithm with variance reduction and dual averaging. Hence I don't fully understand how the acceleration is achieved. Please clarify this point, if possible provide some details within the simplest unconstraint case with adaptive matrix being constant i.e. $\rho Id$.

[1] Song et al, Variance Reduction via Accelerated Dual Averaging for Finite-Sum Optimization

---

> ### Author Response · Authors · 2022-11-15
> **Response to Reviewer u8LZ**
>
> Thanks for spending time reviewing our manuscript. The following is responses to your questions:
>
> **Q1**: The experimental section can be improved to include non iid experiments
>
> **A1**: In our original experiments, the set of experiments for FEMNIST is non-iid, as each person has different writing styles. Furthermore, we add a new set of non-iid experiments, where we manually construct heterogeneous dataset based on CIFAR10. Simply speaking, we create heterogeneity by making each client have a different dominant class. Please see section 8.3 for more details. Our FedDA still outperforms other baselines.
>
>
> **Q2**: How is the theoretical acceleration achieved?
>
> **A2**: First, we agree with the reviewer that the adaptiveness does not contribute to the convergence rate improvement. Our theoretical improvement comes from the 'momentum-based variance reduction', in other words, the update rule in eq.5. This acceleration technique was firstly presented in [1]. In fact, to see why eq.5 can reduce variance, we rewrite (omit $\tau$ and $k$ for ease of notation) eq.5 as: $\nu_{i+1} = (1 - \alpha_{i+1})\nu_{i} + \alpha_{i+1}\nabla f (x_{i+1}, B_{i+1}) + (1 - \alpha_{i+1})(\nabla f (x_{i+1}, B_{i+1}) - \nabla f(x_{i}, B_{i+1})).$
> Note the first two terms correspond to the exact update rule for momentum SGD, the third term evaluates the gradient difference of two consecutive states. The third term is the key component to reduce variance, in fact, it reduces the variance by incorporating the fact that $v$ is the average of historical gradients. We recommend the reviewer to read Section 4 and Lemma 2 of [1] for more detailed explanation.
>
> Finally, compared to the line of work which accelerates SGD by exploiting the finite sum structure, e.g. SAG, SVRG and the reference cited by the reviewer. The momentum-based variance reduction does not assume the finite sum structure or calculate the full gradient at checkpoints.
>
> **[1]** Cutkosky, Ashok, and Francesco Orabona. "Momentum-based variance reduction in non-convex sgd." Advances in neural information processing systems 32 (2019).

---

> > ### Author Response · Authors · 2022-11-17
> > **Connection between the momentum-based variance reduction and variance reduced technique of SPIDER**
> >
> > Besides viewing momentum-based variance reduction as the sum of momentum and a correction item as above, we can also view the momentum-based variance reduction as the **weighted sum** between the **the basic stochastic gradient** and **the variance reduced technique of SPIDER** [1]. More precisely, we can rewrite  eq. 5 as
> >
> > $\nu_{i+1} = \alpha_{i+1}\nabla f (x_{i+1}, B_{i+1}) + (1 - \alpha_{i+1})(\nu_{i} + \nabla f (x_{i+1}, B_{i+1}) - \nabla f(x_{i}, B_{i+1}))$,
> >
> > where the **first term**
> >
> >  $\nabla f (x_{i+1}, B_{i+1})$
> >
> > is the **basic stochastic gradient estimator**, and the **second term**
> >
> > $\nu_{i} + \nabla f (x_{i+1}, B_{i+1}) - \nabla f(x_{i}, B_{i+1})$
> >
> > is the **SPIDER gradient estimator**. Furthermore, we have $\alpha_{i+1} = c\eta_i^2$, where $c$ is some constant and $\eta_i$ is the learning rate. Since we choose decreasing learning rate $\eta_i$,  we have $\alpha_{i+1}$ converges to $0$ as the iteration steps increase. **As a result, the update rule of eq.5 will converge to the **variance reduced SPIDER gradient estimator** as the iteration steps increase.**
> >
> > **[1]** Fang, Cong, et al. "Spider: Near-optimal non-convex optimization via stochastic path-integrated differential estimator." Advances in Neural Information Processing Systems 31 (2018).

---

> ### Author Response · Authors · 2022-12-04
> **Is your concern addressed?**
>
> Dear Reviewer,
>
> Thanks for your time in reviewing our manuscript again. We want to send this message to check if our response has addressed your main concern of how does the variance reduction be achieved in FedDA. If you still have concerns about this point or you have other concerns, please feel free to post a comment and we are very happy to discuss.
>
> Best regards,
>
> Authors

---

### Official Review · Reviewer_7JLR · 2022-11-03

**Confidence:** 3
**Correctness:** 3
**Technical Novelty And Significance:** 3
**Empirical Novelty And Significance:** 3
**Recommendation:** 6

**Clarity, Quality, Novelty And Reproducibility:**

The paper is well-written, especially the introduction. The theoretical results are solid.

**Strength And Weaknesses:**

Strength
1. I really enjoy reading the introduction. I think this paper brings a very novel and important view to adaptive federated learning optimization algorithms. This provide meaningful guidance on how to combine different local optimizer states.
2. The theoretical results are solid. It is not easy to derive the convergence guarantee for adaptive methods with momentum-based variance reduction. I appreciate the authors' efforts.

Weakness
1. While I acknowledge the theoretical contributions of this paper, I'm not very convinced that the proposed method is better than existing much simpler ones.
    - First of all, the comparisons in figure 2 are not fair. In FedDA, clients use local momentum based variance reduction. But for other methods, clients just perform local SGD (even without local momentum for FedAdam). So it is really unclear whether the performance improvements came from momentum-based variance reduction or the new adaptive optimization framework. Although the authors presented MIME-MVR, I think this is not enough. A more convincing way is to just remove the VR part at clients and compare it with all other methods. I found these results in Appendix and actually if we compare figure 5 option 2-1/2 with other methods in figure 2, the improvements due to a better adaptive optimization framework is very marginal and even worse than FedCM. Also, this new framework comes with additional price. Compared to FedAdam, the clients need to download 3x more parameters and upload 2x more. This additional communication overhead does not appear in the experiments. Given the marginal accuracy improvement and much more additional communication, I'm not convinced that this is a better algorithm.
    - Similar to the above point, if you want to show this is a better adaptive optimization framework, then you need to analyze it without any VR and compare it with FedAvg. Analysis of FedMVR is valuable, but it should be treated as an extension.
    - Also, the authors missed a very simple baseline [1], where clients just run any adaptive optimizer from scratch at each round, and the server averages accumulated model changes to uses it in a server-side adaptive optimizer. This simple solution does not introduce any additional communication costs and can work with any optimizer (although not theoretically). But it is worth to discuss and compare with it to see whether the proposed method is unnecessarily complicated.

[1] Wang et al. ICML 2021 workshop. "Local Adaptivity in Federated Learning: Convergence and Consistency"

**Summary Of The Paper:**

This paper studies how to apply adaptive gradient methods into federated learning. Although there are many existing works in this direction, this paper brings up a very novel view. A key problem in adaptive gradients in FL is that there are too many ways to combine the model parameters and local optimizer states at different clients. The authors mentioned that most previous works do not make sense or have obvious drawbacks if we view adaptive methods as a mirror gradient descent algorithm. Instead, in this paper, the authors proposed a more theoretically meaningful framework to aggregate primal (i.e. model parameter) and dual states (i.e. gradients) separately. Under this framework, they provide convergence analysis to an instantiation with momentum-based variance reduction. Experiments on some classification datasets validate the performance of the proposed algorithm.

**Summary Of The Review:**

I acknowledge that this paper makes some contribution in getting a better understanding on the adaptive federated optimization methods. But I am not convinced that the proposed framework is better than previous works, given the fact that without MVR, it only provides marginal improvements but costs 2-3x more communication.

---

> ### Author Response · Authors · 2022-11-15
> **Response to Reviewer 7JLR**
>
> We are glad to see the reviewer agrees that our framework brings a novel view to the adaptive federated learning optimization, this is very encouraging. The comments about empirical comparison are also constructive, we have added more experiments to show the effectiveness of our framework. The detailed responses are as follows:
>
> **Q1**: Discussion and comparison with [Wang et al.]?
>
> **A1**: This is a good reference and related to our method. We added it as a baseline (we refer it as **Local-Adapt** for convenience) in our experiments. In [Wang et. al.], the authors propose to use restarted local adaptive gradients, besides, they also propose two important corrections: local and global corrections (Section 5 of [Wang et. al.]). First, for the local correction, it actually shares some spirit as our dual averaging idea. In the local correction, client first divides the local update by $N_i$ and then sends it to the server for averaging, where $N_i$ is the sum of local precondition/adaptive matrix (see Line 10 in Algorithm 1 of [Wang et. al.]). Since the adaptive matrix defines the mirror map, the 'divide the local update by $N_i$' operation actually converts the client state from the primal space to the dual space. As stated by the authors, local correction helps remove the bias caused by local adaptive gradients. Next, the global correction operation (see Line 16 in Algorithm 1 of [Wang et. al.]) averages the local adaptive matrix to get a coefficient for the global update. This shows that [Wang et. al.] actually shares more information (i.e. adaptive matrix) other than the gradient/model state. In terms of the empirical performance, we use the Adam optimizer for both clients and the server update, and compare it with the FedAdam method (only the server uses Adam and clients use SGD for update). For FEMNIST (Figure 3), Local-Adapt [Wang et. al.] is better than FedAdam, but for CIFAR10 (Figure 2), the improvement is very marginal. Compared to the improvement of our algorithm over FedAdam, this justifies the effectiveness of keeping global momentum and adaptive matrix.
>
>
> **Q2**: Removing VR for better comparison?
>
> **A2**: In our experimental section, we consider three sets of baselines: SGD-based methods, including FedAvg; Momentum-based methods, including FedCM, STEM and MIME-MVR, where FedCM uses momentum, STEM and MIME-MVR uses momentum-based variance reduction; Adaptive methods, including FedAdam, Local-Adapt ([Wang et. al.]) and Local-AMSGrad.
>
> First, compared to FedAvg, all of our variants converges much faster. In particular, FedAvg is bad at generalization (low test accuracy) and dealing with heterogeneous data (as shown by the new heterogeneous results in Section 8.3 of Appendix). Next, the comparison between FedDA-MVR over STEM and MIME-MVR in Figure 2 and 3 shows the benefit of using adaptive gradients in our framework when variance reduction is used for gradient estimate.
>
> Furthermore, as suggested by the reviewer, we add more discussions in Section 8.2 of Appendix (revised version) to compare FedDA without VR. First, we want to correct that in the original version, the results for FedDA-2-1 (using Adam-style adaptive gradients) uses momentum coefficient 0.99, which was intended to be 0.9 for a better performance (0.9 is a common good value for momentum coefficient and other baselines also use this value).
> After this correction, we make various comparisons (under different values of local steps $I$). In Figure 7, we compare FedCM with FedDA-2-1 (Adam) and FedDA-2-2 (AdaGrad), both variants outperform FedCM. In Figure 8, we compare Local-AMSGrad with FedDA-2-1 (Adam), both methods use Adam-style adaptive gradients and have same communication-cost, but our FedDA-2-1 outperforms Local-AMSGrad. Finally, we compare FedDA-2-1 with FedAdam and Local-Adapt in Figure 9. It is true that FedAdam has lower communication cost per epoch compared to FedDA. However, FedAdam has limited ability to exploit local steps. As shown in the first row of Figure 9, the performance of FedAdam improves very litter when we increase $I$ from 5 to 20. This justifies transferring momentum and adaptive matrix to the clients in our FedDA. In fact, baselines which transfer momentum or adaptive matrix can also benefit from local steps, such as FedCM and Local-AMSGrad.

---

### Official Review · Reviewer_rANd · 2022-11-05

**Confidence:** 4
**Clarity, Quality, Novelty And Reproducibility:** The paper is clearly written.
**Correctness:** 4
**Technical Novelty And Significance:** 2
**Empirical Novelty And Significance:** 2
**Recommendation:** 5

**Strength And Weaknesses:**

*Strengths*

The paper is well written. The theoretical analysis is well done and experimental results are detailed.

*Weakness*

- My main concern is regarding the motivation. If the eventual convergence rate is similar to other wxisting works, then what is the additional benefit to consider this approach?  The experimental gain alone is also not significant enough. And that too coming at the cost of added computatinal complexity at the device to perform argmin operation.

- Also, it seems the comparison to FedAvg is also not fair because the proposed algorithm requires to share some gradient information with the server, which is not needed in FedAvg. How to incorporate that fact during the comparisons? Also, isn't that against the spirit of federated learning in general to share the gradient information with the server?

- The dual state is not defined before it's use in Line 4 in the description of Algorithm 4.

- Is it true that via $\nu$, the information of global gradient is getting used for the local updates at each client? If yes, this needs to be discussed in detail.

- In the proposed algorithm, because it is assumed that we can share the estimated gradients with the server, then what if we evaluate the average of these local gradients at the server, and then run the update at the server.  What is the additional benefit of running local gradient updates, except for the linear speedup?

-

**Summary Of The Paper:**

The authors have considered the standard problem of federated learning (FL) and proposed an adaptive gradient method to solve the problem. The paper is written and theoretical convergence analysis is provided for the proposed approach. The experimental results are also mentioned to further support the claims. My main comments are provided below.

**Summary Of The Review:**

Please refer to the weakness mentioned above.

---

> ### Author Response · Authors · 2022-11-15
> **Response to Reviewer rANd (1)**
>
> Thanks for spending time on reviewing our manuscript.
>
> **Q1:** ... If the eventual convergence rate is similar to other existing
> works,....And that too coming at the cost of added computatinal complexity at the device to perform argmin operation.
>
> **A1:** The **main contribution and novelty** of our paper is that we propose
> a novel adaptive primal-dual algorithmic framework for federated learning based on the mirror descent
> iteration. **NOTE THAT** our adaptive iterations
> (the line 9 at our Algorithm 1 and the line 5 at our Algorithm 2) can be seen as the mirror descent
> iteration. Moreover, we use the momentum techniques in local gradient estimators
> (the line 6 at our Algorithm 2, i.e., the Eq. (5) and (6) in our paper). In particular, the momentum-based
> variance reduced technique of STORM let our algorithm obtain the near-optimal
> gradient complexity $O(\epsilon^{-1.5})$ and communication complexity $O(\epsilon^{-1})$ simultaneously
> as in **[1]**. Since **[2]** shows the stochastic algorithms in solving the nonconvex stochastic problems
> has a lower bound complexity $O(\epsilon^{-1.5})$ for finding an $\epsilon$-stationary point,
> our adaptive algorithm obtain this near-optimal
> gradient complexity $O(\epsilon^{-1.5})$, and can not further improve it.
> To the best of our knowledge, our FedDA-MVR is the first adaptive federated algorithm that attains such convergence rate.
> Additionally, it is well known that adaptive gradient methods perform well in practice although with same convergence rate as non-adaptive gradient methods.
> Therefore, it is meaningful to investigate adaptive gradient methods in the federated learning setting.
>
> **NOTE THAT**  When $\mathcal{X}=\mathbb{R}^d$, i.e., the problem (1) in our paper is **unconstrained** optimization,
> the update step in line 5 of Algorithm 2 can be simplified to simple (adaptive) gradient descent step as follows:
> $x_{i+1} = x_0-\lambda H^{-1}z_{i}$ (we omit the $\tau$ and $k$ for ease of notation), there is no extra computational cost,
>  since the adaptive matrices are diagonal matrices.
> When $\mathcal{X}\subset \mathbb{R}^d$, i.e.,
> the problem is **constrained** optimization, since the adaptive matrices are diagonal matrices,
> we can also easily obtain their closed-form solutions under some specific constrained sets such as $\|x\|_1 \leq r$.
>
>
> **Q2:** Also, it seems the comparison to FedAvg is also not fair because the proposed algorithm requires
> to share some gradient information with the server,.....
>
> **A2:** The aggregated dual state $z$ shared in our algorithm is equivalent to sharing model difference in the FedAvg algorithm. Note that model difference sharing is equivalent to model sharing when the server has the model state of the last global epoch.
> Meanwhile, our algorithm uses the momentum (variance reduced) techniques to estimate
> the local stochastic gradients (the line 6 at our Algorithm 2, i.e., the Eq. (5) and (6) in our paper),
> which can be seen as local momentum. Since the goal of FL is to find the global
> solution of the global objective function, we should share the global momentum to all clients
> at each epoch (i.e., $\tau$ in Algorithm 1) by averaging the local momentum (i.e., $\nu^k$) in the server.
> **NOTE THAT** this averaging the local momentum is very common in momentum-based FL methods such as STEM **[1]**.
> By using these local and global momentums, our algorithm can obtain the near-optimal
> gradient complexity $O(\epsilon^{-1.5})$ and communication complexity $O(\epsilon^{-1})$ simultaneously
> as in **[1]**.
>
> The main contribution and novelty of our paper is that we propose
> a novel adaptive primal-dual algorithmic framework for federated learning based on the mirror descent
> iteration.  Clearly, our new adaptive
> primal-dual algorithmic framework can also use the other variance-reduced and momentum techniques.
> **Note that  our adaptive primal-dual algorithmic framework does not rely on the momentum techniques,
> we only use the momentum techniques in our algorithm as an example.**
>
>
> **[1]** Khanduri, Prashant, et al. "Stem: A stochastic two-sided momentum algorithm achieving near-optimal sample and
> communication complexities for federated learning."
> NeurIPS, 34 (2021): 6050-6061.
>
> **[2]** Arjevani, Yossi, et al. "Lower bounds for non-convex stochastic optimization." Mathematical Programming (2022): 1-50.

---

> > ### Author Response · Authors · 2022-11-16
> > **Response to Reviewer rANd (2)**
> >
> > **Q3:** The dual state is not defined before it's use in Line 4 in the description of Algorithm 4.
> >
> > **A3:** Thanks for your suggestion. The definition of dual/primal states in our manuscript follows the convention in mirror descent literature
> > (Line 5-6 of the third paragraph in the Introduction section): We denote the parameter space as the
> > primal space and the gradient space as the dual space. More specifically, $z$ is the dual state in our algorithm.
> > At each epoch, clients update the dual state $z^{k}$ by accumulating local gradient estimate $\nu^{k}$ (line 4 of algorithm 2),
> > then the server averages the local dual states (line 8 of algorithm 1) to get the global dual state $z$.
> >  Note the global dual state is used to recover the primal state $x$ (line 9 of algorithm 1)
> > and to update the adaptive matrix (line 11 of algorithm 1).
> >
> > **Q4:** Is it true that via $\nu$, the information of global gradient is getting used for the local updates at each client?
> > If yes, this needs to be discussed in detail.
> >
> > **A4:**  Yes, you are right. Our algorithm use the momentum (variance reduced) techniques to estimate
> > the local stochastic gradients (the line 6 at our Algorithm 2, i.e., the Eq. (5) and (6) in our paper),
> > which can be seen as local momentum. Since the goal of FL is to find the global
> > solution of the global objective function, we should share the global momentum to all clients
> > at each epoch (i.e., $\tau$ in Algorithm 1) by averaging the local momentum (i.e., $\nu^k$) in the server.
> > Under this case, our algorithm can obtain the convergence guarantee and also obtain near-optimal
> > gradient complexity $O(\epsilon^{-1.5})$ and communication complexity $O(\epsilon^{-1})$ simultaneously
> > as in  **[1]** .
> > **NOTE THAT** this averaging the local momentums is very common in momentum-based FL methods such as **STEM [1]**.
> >
> > **Q5:** In the proposed algorithm, because it is assumed that we can share the estimated gradients with the server,
> > .... What is the additional benefit of running local gradient updates, except for the linear speedup?
> >
> > **A5:** Thanks for your comment. In fact, our algorithm uses the momentum-based variance reduction techniques to estimate
> > the local stochastic gradients (the line 6 at our Algorithm 2, i.e., the Eq. (5) and (6) in our paper),
> > which can be seen as local momentum. Since the goal of FL is to find the global
> > solution of the global objective function, we should share the global momentum to all clients
> > at each global epoch (i.e., $\tau$ in Algorithm 1) by averaging the local momentum (i.e., $\nu^k$) in the server.
> > Under this case, our algorithm can obtain the convergence guarantee and also obtain near-optimal
> > gradient complexity $O(\epsilon^{-1.5})$ and communication complexity $O(\epsilon^{-1})$ simultaneously
> > as in **[1]** . This averaging the local momentum is very common in momentum-based FL methods such as **STEM [1]**.
> >
> > **NOTE THAT** in our algorithm, at each global epoch, the server only averages the local momentum $\nu^k$  and does not perform update to the momentum. Instead, clients perform multiple steps of momentum update (line 6 of algorithm 2) locally. In fact, updating the momentum locally is important for gaining the optimal convergence rate we mentioned above. If as suggested by the reviewer, we only update the momentum at the server (after the server averages the local momentum), the result is that we can only perform $E$ (the number of global epochs) steps of update and the convergence rate will be $O(E^{-2/3})$. Note in corollary 1, we attain convergence bound in terms of $T$ ($T = E*I$) with rate of $O(T^{-2/3})$. In other words, the acceleration effect of local steps will disappear if we choose to update the momentum at the server.
> >
> >
> > **[1]** Khanduri, Prashant, et al. "Stem: A stochastic two-sided momentum algorithm achieving near-optimal sample and
> > communication complexities for federated learning."
> > NeurIPS, 34 (2021): 6050-6061.
> >
> > **[2]** Arjevani, Yossi, et al. "Lower bounds for non-convex stochastic optimization." Mathematical Programming (2022): 1-50.

---

### Author Response · Authors · 2022-11-15
**General Response to All Reviewers (2)**



In this manuscript, we study the adaptive gradient methods in Federated Learning. Due to the empirical success of adaptive gradients, the community has also tried to apply them in FL, but several works **[1,2]** find that directly using adaptive gradients in client updates leads to bias. However, there still lacks formal understanding of why adaptive gradients is not compatible with local gradients. In our manuscript, we offer one explanation from the primal-dual's (model weight-accumulated gradient's) view. Simply speaking, using local adaptive gradients makes the dual space of clients not aligned with each other, thus direct averaging of primal states is not meaningful. We propose a solution by fixing the adaptive matrix in local updates and averaging over the dual states (aggregated gradients). To the best of our knowledge, we are the first one to use this primal-dual view for adaptive methods in FL. In **[3]** (A reference suggested by **Reviewer 7JLR**), authors perform restarted local adaptive gradients, but they propose to use 'local correction' when averaging clients' states. The local correction operation divides the local states by the sum of adaptive matrix, thus, this shares the spirit of our dual averaging idea. A more detailed discussion of [3] is in the response to Q1 of **Reviewer 7JLR**.

Furthermore, our novel primal-dual view provides a way to incorporate various momentum and adaptive gradient estimation methods (**Reviewer pitV and Reviewer 7JLR**), moreover, we can solve both unconstrained and constrained problems. In particular, when integrating with the momentum-based variance reduction (and adaptive gradients), we achieve optimal convergence rate. This is achieved for the first time to the best of our knowledge and the theoretical analysis is non-trivial (**Reviewer 7JLR**). In particular, in response to **Reviewer u8LZ** about how the variance reduction is achieved: the momentum-based variance reduction is a different variance reduction technique from SAGA, SVRG etc., and it does not rely on the finite sum structure. Besides, the main concern of **Reviewer rANd** is that why do we study the adaptive gradient methods if the convergence rate is the same as non-adaptive ones? The advantage of adaptive methods is its good empirical performance (as pointed out by the **Reviewer u8LZ**).

Finally, to further validate the empirical performance of our FedDA, we add some experiments in our revised version of the manuscript (Section 8.2 and 8.3 in the Appendix). In section 8.2, we make more comparisons between our methods and baselines and add some ablation studies (vary the number of local steps $I$); In section 8.3, we test over a manually created heterogeneous dataset based on CIFAR10. Besides, In section 8.4, we show the form of our FedDA when $I=1$, i.e. no local steps. Please see the revised version of our manuscript for more details.


**[1]** Tang, Hanlin, et al. "1-bit adam: Communication efficient large-scale training with adam’s convergence speed." International Conference on Machine Learning. PMLR, 2021.

**[2]** Chen, Xiangyi, Xiaoyun Li, and Ping Li. "Toward communication efficient adaptive gradient method." Proceedings of the 2020 ACM-IMS on Foundations of Data Science Conference. 2020.

**[3]** Wang et al. ICML 2021 workshop. "Local Adaptivity in Federated Learning: Convergence and Consistency"

---

### Author Response · Authors · 2022-11-16
**General Response to All Reviewers (1)**

Firstly, we want to thank all reviewers for spending time reviewing our manuscript and for providing insightful comments.

**Our contributions and novelties** of are:

1). We propose  **a novel primal-dual algorithmic framework for adaptive federated learning** based on the mirror descent
iteration. In fact, our adaptive iterations
(the line 9 at our Algorithm 1 and the line 5 at our Algorithm 2) can be seen as the mirror descent
iteration. Moreover, we use the momentum techniques in local gradient estimators
(the line 6 at our Algorithm 2, i.e., the Eq. (5) and (6) in our paper). In particular, the momentum-based
variance reduced technique let our algorithm obtain the near-optimal
gradient complexity $O(\epsilon^{-1.5})$ and communication complexity $O(\epsilon^{-1})$ simultaneously
as in **[1]**. Clearly, our new adaptive
primal-dual algorithmic framework can also use other variance-reduced and momentum techniques.
**Note that** our adaptive primal-dual algorithmic framework does not rely on the momentum techniques,
we only use the momentum techniques in our algorithm as an example.

2). We provide **a new convergence analysis framework** for our adaptive algorithm.
Specifically, we prove that our algorithm obtains the near-optimal
gradient complexity $O(\epsilon^{-1.5})$ and communication complexity $O(\epsilon^{-1})$ simultaneously
as in **[1]**. Since **[2]** shows the stochastic algorithms in solving the nonconvex stochastic problems
has a lower bound complexity $O(\epsilon^{-1.5})$ for finding an $\epsilon$-stationary point,
our adaptive algorithm obtains this near-optimal
gradient complexity $O(\epsilon^{-1.5})$, and can not further improve it.
To the best of our knowledge, our FedDA-MVR is the first adaptive federated algorithm that attains such convergence rate.
Additionally, it is well known that adaptive gradient methods perform well in practice although with same convergence rate as non-adaptive gradient methods.
Therefore, it is meaningful to investigate adaptive gradient methods in the federated learning setting.

**[1]** Khanduri, Prashant, et al. "Stem: A stochastic two-sided momentum algorithm achieving near-optimal sample and
communication complexities for federated learning."
NeurIPS, 34 (2021): 6050-6061.

**[2]** Arjevani, Yossi, et al. "Lower bounds for non-convex stochastic optimization." Mathematical Programming (2022): 1-50.

---

### Author Response · Authors · 2022-12-01
**We are happy to have discussion with you**

Dear Reviewers,

We want to thank you again for your reviews. If you still have any concerns, please let us know. We are very happy to discuss them, Thanks!

Best regards,

Authors

---

### Decision · Program_Chairs · 2023-01-20

**Decision:**

Reject

**Justification For Why Not Higher Score:**

Lack of technical innovation. This work is a straightforward application of pre-existing theories, and does not contribute anything unique to the federated learning problem class.

**Justification For Why Not Lower Score:**

NA

**Metareview: Summary, Strengths And Weaknesses:**

This work introduces a dual averaging and restarting scheme based on proximal iteration and Lagrange duality for incorporating adaptive step-size. Convergence rate analysis is given, and experimental results corroborate the main theoretical findings. In particular, improved adaptivity in practice is demosntrated.

The strengths are in the literature review, the first introduction of dual averaging/restarting into federated learning framework, and the experimental merits.

The weaknesses are in the fact that dual averaging and restarting are already well-known in the optimization literature, and their application to federated learning is not particularly novel. Experimental analysis demonstrating improvement relative to some baselines is insufficient, given that the proposed technique ultimately achieves comparable performance theoretically to pre-existing results.

---

> ### Author Response · Authors · 2023-02-13
> **AC misunderstand and underestimate the contribution and novelty of our work**
>
> In the final comment, AC claims that our FedDA does not contribute anything unique to the federated learning problems. We believe **this comment is wrong** and our method indeed make **unique** contributions to the FL problem.  We want to restate our **contribution** as follows:
>
> Adaptive gradients are not directly applicable to the Federated Learning setting, and directly using adaptive gradients in client updates leads to bias.  In our work, we first explain the cause of bias from the primal-dual's perspective:  using local adaptive gradients makes the dual space of clients not aligned with each other, thus direct averaging of primal states causes bias.Then we propose a solution by fixing the adaptive matrix in local updates and averaging over the dual states. **Note that the 'dual averaging' here represents the average of dual states among clients, this is completely different from the dual averaging in the non-distributed setting.**
>
> Furthermore, we propose a general framework to incorporate various momentum and adaptive gradient estimation methods on top of our dual averaging technique. In particular, when integrating with the momentum-based variance reduction (and adaptive gradients), we achieve optimal convergence rate. This is achieved for the first time to the best of our knowledge and the theoretical analysis is non-trivial. **Besides, note that although adaptive gradient methods achieves the same convergence rate as the non-adaptive ones, it leads to good empirical performance, therefore, it is meaningful to study adaptive gradient methods.**